# Appropriate Balance of Diversification and Intensification Improves Performance and Efficiency of Adversarial Attacks

**Keiichiro Yamamura** *keiichiro.yamamura@kyudai.jp*
*Graduate School of Mathematics*
*Kyushu University*

**Issa Oe**[*] *issa-oe@kyudai.jp*
*Graduate School of Mathematics*
*Kyushu University*

**Nozomi Hata**[†] *n.hata@kyudai.jp*
*Institute of Mathematics for Industry*
*Kyushu University*

**Hiroki Ishikura** *ishikura.h.aa@m.titech.ac.jp*
*Institute of Innovative Research*
*Tokyo Institute of Technology*

**Katsuki Fujisawa** *fujisawa.k.aa@m.titech.ac.jp*
*Institute of Innovative Research*
*Tokyo Institute of Technology*

**Reviewed on OpenReview:** *https://openreview.net/forum?id=mK6TwmInTg*

## Abstract

Recently, adversarial attacks that generate adversarial examples by optimizing a multimodal function with many local optimums have attracted considerable research attention. Quick convergence to a nearby local optimum (intensification) and fast enumeration of multiple different local optima (diversification) are important to construct strong attacks. Most existing white-box attacks that use the model's gradient enumerate multiple local optima based on multi-restart; however, our experiments suggest that the ability of diversification based on multi-restart is limited. To tackle this problem, we propose the multi-directions/objectives (MDO) strategy, which uses multiple search directions and objective functions for diversification. Efficient Diversified Attack, a combination of MDO and multi-target strategies, showed further diversification performance, resulting in better performance than recently proposed attacks against around 88% of 41 CNN-based robust models and 100% of 10 more advanced models, including transformer-based architecture. These results suggest a relationship between attack performances and a balance of diversification and intensification, which is beneficial to constructing more potent attacks.

## 1 Introduction

Deep neural networks (DNNs) have demonstrated excellent performance in several applications. However, DNNs are known to misclassify adversarial examples generated by tiny perturbing inputs that are imperceptible to humans (Szegedy et al., 2014). Vulnerabilities caused by adversarial examples can have fatal

---

[*]Present affiliation is NTT Computer & Data Science Laboratories, NTT Corporation.
[†]Present affiliation is NTT Communication Science Laboratories, NTT Corporation.

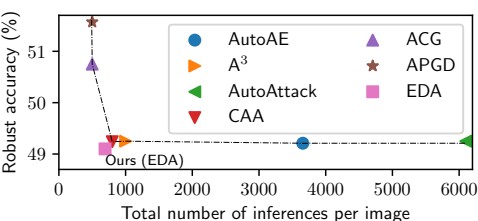

Figure 1: EDA vs. recently proposed attacks on adversarially trained CIFAR-10 model Engstrom et al. (2019).

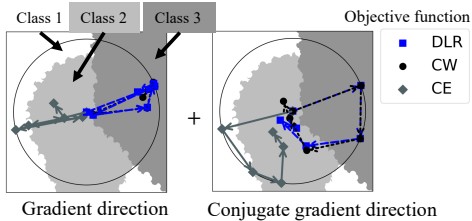

Figure 2: Illustration of the MDO strategy with two search directions and three objective functions on a toy model. The different search directions/objective functions search for different regions.

consequences, especially in safety-critical applications such as automated driving (Gupta et al., 2021), facial recognition (Adjabi et al., 2020), and cybersecurity (Liu et al., 2022b). Adversarial training (Madry et al., 2018) is one of the most effective defenses, which uses adversarial examples during training. Generating many adversarial examples faster is beneficial for the security of DNNs because adversarial training requires many adversarial examples. Our ultimate goal is to find many adversarial examples quickly, with proper control of both extensive and intensive searches.

Adversarial attacks optimize a challenging nonconvex function to find adversarial examples. We focus on white-box attacks that use gradient-based optimization algorithms, assuming access to the outputs and gradients of the DNN. Higher objective function values increase misclassification chances, creating adversarial example candidates out of local optima. The objective function of this problem is multimodal because its maximization involves a complex DNN. Because a multimodal function has a myriad of local optima, quick convergence to a nearby local optimum and fast enumeration of multiple different local optima are important. These are referred to as intensification and diversification, respectively (Glover & Samorani, 2019).

Many existing gradient-based attacks are considered to achieve some degree of intensification because the objective value can be improved by moving in the gradient direction within a neighborhood. Whereas many existing attacks diversify the search based on the multi-restart (Dong et al., 2013; Madry et al., 2018; Croce & Hein, 2020b; Liu et al., 2022c), few studies have examined other diversification strategies. Therefore, further research on diversification is needed.

Diversification and intensification involve trade-offs, particularly under a strict computational budget. Thus, we have to balance both of them appropriately. However, an appropriate control method for this balance has yet to be developed. This study is the first to systematically explore and control this balance by coordinating multiple search directions and objective functions based on diversification indices. Using the proposed methodology, we can discover adversarial examples that are elusive even to current state-of-the-art methods. Our experiments using diversification indices in Section 3.1 suggest that the attacks using a multi-start strategy are likely to converge in the same cluster, which does not yield high diversification performance. However, the experiments in Section 3.1 also suggest that different local solutions may be enumerated using different search directions and objective functions. Inspired by this observation, we propose the multi-directions/objectives (MDO) strategy that uses multiple search directions and objective functions. Figure 2 is a toy example of an attack using the MDO strategy.

To implement this strategy effectively, we propose the Automated Diversified Selection (ADS) and a search framework called the MDO framework in Section 3.2 and Section 3.3, respectively. The MDO strategy needs to utilize an appropriate combination of search direction and objective function for effective diversification. This is because some combinations may explore similar areas. ADS selects search directions and objective functions that are likely to search different areas based on the Diversity Index (DI) (Yamamura et al., 2022) to enhance diversification. Unlike existing approaches for attack selection (Mao et al., 2021; Liu et al., 2023b), which rely on direct indicators like loss value and attack success rate, ADS employs an indirect indicator such as DI.

The MDO framework explicitly considers the diversification and intensification phases through ensemble and composite of attacks to balance them appropriately. The diversification phase finds promising starting points via extensive search, and the intensification phase enhances attack performance through intensive search in the vicinity of these points.

We evaluated the proposed methods with robust accuracy, computation cost, and DI because the attack with an appropriate balance of diversification and intensification is expected to be strong and fast. The experimental results in Sections 4.3 and 4.4 suggest that ADS contributes to search diversification and successfully finds the appropriate combination in both diversification and intensification. The MDO framework also realizes an appropriate balance of diversification and intensification, resulting in a promising attack performance. As shown in Section 4.2, the MDO strategy found adversarial examples for some inputs in less time than the multi-target (MT) strategy (Gowal et al., 2019), one of the promising attacks. These results imply that the combination of MDO and MT strategies can realize stronger and faster attacks.

Motivated by these results, we experimentally investigated the attack performance of the combination of MDO and MT strategies, called Efficient Diversified Attack (EDA). Experimental results in Section 4.1 show that EDA exhibits higher diversification performance than the attack using the MDO strategy alone, resulting in better attack performance in less computation time than the recently proposed attacks under the standard setting of perturbation bound ($\varepsilon$) in the RobustBench leaderboard Croce et al. (2021). Given the difference in robust accuracy among recently proposed attacks, EDA exhibits sufficiently large improvements in robust accuracy and runtime. The above experimental results suggest that appropriately enhancing diversification leads to higher attack performance. The major contributions of this study are summarized below.

1. **Multi-directions/objectives (MDO)** strategy and its implementation: A novel search strategy using multiple search directions and objective functions, realized by Automated Diversified Selection (ADS) and the MDO framework. ADS chooses the gradient-based search direction and the objective function that improves the likelihood of misclassification so that the DI-based index is maximized, thereby achieving efficient diversification and avoiding getting stuck during intensification. The MDO framework finds promising starting points in the diversification phase and enhances the objective value during the intensification phase.
2. **Efficient Diversified Attack (EDA)**: A faster and stronger attack using MDO and MT strategies. Under the standard setting of perturbation bound in RobustBench leaderboard, EDA showed the best attack performance among recent attacks against around 88% of 41 CNN-based robust models trained on three representative datasets and 100% of 10 more advanced models, including transformer-based architecture, trained on ImageNet (Russakovsky et al., 2015).

## 2 Preliminaries

### 2.1 Problem settings

Let $g : D \to \mathbb{R}^C$ be a locally differentiable $C$-classifier, $\boldsymbol{x}_{\mathrm{org}} \in D$ be a point with $c$ as the correct label, and $d : D \times D \to \mathbb{R}$ be a distance function. Given $\varepsilon > 0$, the feasible region $\mathcal{S}$ is defined as the set of points $\boldsymbol{x} \in D$ that are within a distance of $\varepsilon$ from $\boldsymbol{x}_{\mathrm{org}}$, i.e., $\mathcal{S} := \{\boldsymbol{x} \in D \mid d(\boldsymbol{x}, \boldsymbol{x}_{\mathrm{org}}) \leq \varepsilon\}$. Then, we define an adversarial example as $\boldsymbol{x}_{\mathrm{adv}} \in \mathcal{S}$ satisfying $\arg\max_{i=1,\dots,C} g_i(\boldsymbol{x}_{\mathrm{adv}}) \neq c$. The following expression is a formulation of one type of adversarial attack, the untargeted attack, where the attacker does not specify the misclassification target.

$$\text{find } \boldsymbol{x} \in \mathcal{S} \text{ s.t. } \max_{i=1,\dots C} g_i(\boldsymbol{x}) - g_c(\boldsymbol{x}) > 0 \tag{1}$$

Problem 1 is solved through maximization of the objective function $L(g(\boldsymbol{x}), c)$ within the feasible region $\mathcal{S}$. This maximization aims to reduce the probability that $\boldsymbol{x}$ is classified in class $c$ by $g$. Therefore, $\boldsymbol{x}$ with a high objective value $L(g(\boldsymbol{x}), c)$ is more likely to be misclassified by $g$. When $d(\boldsymbol{x}_{\mathrm{adv}}, \boldsymbol{x}_{\mathrm{org}})$ is small, the norm of the adversarial perturbation is also small. The targeted attack aims at maximizing the probability that $\boldsymbol{x}_{\mathrm{adv}}$ is classified in a particular class $t \neq c$ by solving $\max_{\boldsymbol{x} \in \mathcal{S}} L(g(\boldsymbol{x}), c, t)$. For adversarial attacks on image classifiers, $D = [0, 1]^n$ and $d(\boldsymbol{v}, \boldsymbol{w}) := \|\boldsymbol{v} - \boldsymbol{w}\|_p, (p = 2, \infty)$ is typically used. This study focuses on the untargeted attack on image classifiers using $d(\boldsymbol{v}, \boldsymbol{w}) := \|\boldsymbol{v} - \boldsymbol{w}\|_\infty$, referred to as $\ell_\infty$ attacks.

## 2.2 Related work

**General white-box attacks.** In the white-box attack, the initial point sampling x determines $\boldsymbol{x}^{(0)}$ first. Then, the step size update rule $\eta$ and the search direction $\boldsymbol{\delta} = \boldsymbol{\delta}(L)$ updates the step size $\eta^{(k)}$ and the moving direction $\boldsymbol{\delta}^{(k)}$, respectively. Subsequently, the search point $\boldsymbol{x}^{(k+1)}$ is updated by the following formula.

$$\boldsymbol{x}^{(0)} \leftarrow \text{sampled by x}, \ \boldsymbol{x}^{(k+1)} \leftarrow P_{\mathcal{S}}\left(\boldsymbol{x}^{(k)} + \eta^{(k)}\boldsymbol{\delta}^{(k)}\right), \tag{2}$$

where $k$ is the iteration, and $P_{\mathcal{S}}$ is a projection onto $\mathcal{S}$. The moving direction $\boldsymbol{\delta}^{(k)}$ is usually computed based on the gradient $\nabla L(g(\boldsymbol{x}^{(k)}), c)$. According to equation 2, attack methods are characterized by the initial point sampling x, step size update rule $\eta$, search direction $\boldsymbol{\delta}$, and objective function $L$. The tuple $a = (\mathrm{x}, \eta, \boldsymbol{\delta}, L)$ is referred to as an attack $a$. Algorithm 1 shows the procedure of the general white-box attacks $a$ without multi-restart. We assume that the white-box attack $a$ returns a set of best search points $X^a$ and a set of classification labels with the highest probability except for the correct classification labels during the search $\Pi^a$. The basic white-box attacks iteratively update the search point $\boldsymbol{x}^{(k)}$ as lines 5 and 6 in Algorithm 1 to search for adversarial examples. At each iterations $k$, the best search point $\boldsymbol{x}^*$ is updated if $L_{\mathrm{CW}}(g(\boldsymbol{x}^*), c) \leq L_{\mathrm{CW}}(g(\boldsymbol{x}^{(k)}), c)$. $L_{\mathrm{CW}}$ denotes CW loss proposed by Carlini & Wagner (2017). After the $N$ iterations of updates, the attack procedure is finished. For the attacks with a multi-restart strategy,

---

**Algorithm 1** General white-box attacks without multi-restart (Attack)

---

**Require:** $a = (\mathrm{x}, \eta, \boldsymbol{\delta}, L)$: an attack, $N$: maximum iteration, $P_{\mathcal{S}}$: a projection function, $g$: DNN, $\mathcal{I}$: test samples

**Ensure:** $X^a$: a set of best search points, $\Pi^a$: a set of classification labels with the highest probabilities except for the correct classification class

1: **for** $i = 1, \ldots, |\mathcal{I}|$ **do**
2:     $\boldsymbol{x}_{\mathrm{org}} \leftarrow \boldsymbol{x}_i \in \mathcal{I}, c_i \in Y \leftarrow$ correct classification label corresponding to $\boldsymbol{x}_i$
3:     $\boldsymbol{x}^{(0)} \leftarrow$ initialize by x, $\boldsymbol{x}^* \leftarrow \boldsymbol{x}^{(0)}, f_i^{best} \leftarrow L_{\mathrm{CW}}(g(\boldsymbol{x}^{(0)}), c_i)$
4:     **for** $k = 0, \ldots, N - 1$ **do**
5:         Update $\eta^{(k)}$ and $\boldsymbol{\delta}^{(k)}$ by update rule $\eta$ and search direction $\boldsymbol{\delta}$.
6:         $\boldsymbol{x}^{(k+1)} \leftarrow P_{\mathcal{S}}\left(\boldsymbol{x}^{(k)} + \eta^{(k)} \cdot \boldsymbol{\delta}^{(k)}\right)$
7:         Update $\boldsymbol{x}^*$ and $f_i^{best}$
8:         $\Pi_i^a \leftarrow \Pi_i^a \cup \{\arg\max_{j \neq c_i} g_j(\boldsymbol{x}^{(k+1)})\}$
9:     **end for**
10:    $X_i^a \leftarrow X_i^a \cup \{\boldsymbol{x}^*\}$
11: **end for**

---

Algorithm 1 is repeated from different initial points. The returned $X_i^a$ should contains $R$ best search points if the number of restarts is $R$.

**Commonly used attack components.** Projected Gradient Descent (PGD) (Madry et al., 2018) is a fundamental white-box attack. PGD uses a fixed step size ($\eta_{\mathrm{fix}}$) and moves to the normalized gradient direction ($\boldsymbol{\delta}_{\mathrm{PGD}}$). Auto-PGD (APGD) (Croce & Hein, 2020b) is a variant of PGD using a heuristic ($\eta_{\mathrm{APGD}}$) for updating step size and moving to the momentum direction ($\boldsymbol{\delta}_{\mathrm{APGD}}$). In addition, some studies use cosine annealing ($\eta_{\cos}$) (Loshchilov & Hutter, 2017) for updating step size. Gradual reduction of step size, such as $\eta_{\mathrm{APGD}}$ and $\eta_{\cos}$, showed better results than fixed step size. Auto-Conjugate Gradient attack (ACG) (Yamamura et al., 2022) uses $\eta_{\mathrm{APGD}}$ and moves to the normalized conjugate gradient direction ($\boldsymbol{\delta}_{\mathrm{ACG}}$). Whereas the sort of steepest directions, such as $\boldsymbol{\delta}_{\mathrm{PGD}}$ and $\boldsymbol{\delta}_{\mathrm{APGD}}$, are suitable for intensification, the conjugate gradient-based direction is suitable for diversification. For the initial point, uniform sampling from $\mathcal{S}$ or input points ($\mathrm{x}_{\mathrm{org}}$) is usually used. Assuming the many restarts, Output Diversified Sampling (ODS, $\mathrm{x}_{\mathrm{ODS}}$) and its variant, which considers the output diversity of the threat model (Tashiro et al., 2020; Liu et al., 2022c), outperformed naive random sampling. For the objective functions, cross-entropy (CE) loss (Goodfellow et al., 2015) ($L_{\mathrm{CE}}$) and margin-based losses such as CW loss ($L_{\mathrm{CW}}$), a variation of CW loss scaled by the softmax function ($L_{\mathrm{SCW}}$), and Difference of Logits Ratio (DLR) loss (Croce & Hein, 2020b) ($L_{\mathrm{DLR}}$) are often used. We denote the targeted version of these losses as $L^T$, e.g., $L_{\mathrm{CE}}^T$ for targeted CE loss. See Appendices A.1 and A.2 for mathematical formulas of attack components.

**Robustness evaluation.** The robustness of DNNs is usually evaluated using AutoAttack (AA) Croce & Hein (2020b), combining four different attacks, including the MultiTargeted (MT) Gowal et al. (2019) strategy. MT strategy runs targeted attacks on each of multiple target classes and thus requires higher computation costs. The high computational cost of AA has motivated the research community to pursue faster attacks for adversarial training and robustness evaluation (Gao et al., 2022; Xu et al., 2022). Composite Adversarial Attack (CAA) (Mao et al., 2021) and AutoAE (Liu et al., 2023b) combine multiple attacks by solving an additional optimization problem, requiring the pre-execution of candidate methods on relatively large samples. Consequently, they are still computationally expensive. Adaptive Auto Attack ($A^3$) Liu et al. (2022c) demonstrated state-of-the-art (SOTA) attack performance in less computation time by improving the initial point sampling and discarding the hard-to-attack images. $A^3$ is based on PGD with CW loss, which uses a single search direction and objective function. Some studies have investigated how to switch either the search direction or the objective function based on case studies to improve attack performance (Yamamura et al., 2022; Antoniou et al., 2022). However, to the best of our knowledge, this is the first study that investigates the combination of multiple search directions and objective functions based on search diversification. See Appendix A.3 for more information.

**Quantification of search diversity.** Yamamura et al. (2022) proposed the *Diversity Index* (DI) to quantify the degree of diversification during the attacks. DI is defined as

$$\mathrm{DI}(X, M) := \frac{1}{M} \int_0^M h(\theta; X) \ d\theta, \tag{3}$$

where $M = \sup\{\|\boldsymbol{x} - \boldsymbol{y}\|_2 \mid \boldsymbol{x}, \boldsymbol{y} \in \mathcal{S}\}$ is the size of the feasible region, $X$ is the set of search points, and $h(\theta; X)$ is the function of $\theta$ based on the global clustering coefficient of the graph $G(X, \theta) = (X, E(\theta)) = (X, \{(\boldsymbol{u}, \boldsymbol{v}) \mid \boldsymbol{u}, \boldsymbol{v} \in X, \|\boldsymbol{u} - \boldsymbol{v}\|_2 \leq \theta\})$. By definition, $0 \leq h(\theta; X) \leq 1$ holds. Thus, $0 \leq DI(X, M) \leq 1$ also holds. DI quantifies the degree of density of any point set as a value between 0 and 1 based on the global clustering coefficient of a graph. DI tends to be small when the point set forms clusters. Because of this feature, DI is suitable to evaluate how clustered the point set is. We use the weighted average of DI to choose pairs of search directions and objective functions that are likely to search for different areas. This study uses the same value of $M$ as in Yamamura et al. (2022). For simplicity, $DI(X, M)$ is denoted by $DI(X)$ in this paper.

## 3 Multi-directions/objectives strategy

### 3.1 Motivation

We hypothesize that diversification contributes to attack performance. This section empirically demonstrates that attacks with multiple search directions ($\boldsymbol{\delta}$) and objective functions ($L$) can achieve more efficient diversification than attacks with a single $\boldsymbol{\delta}$ and $L$. Four search directions and seven objective functions are used in this experiment. $N_{\max}$ iterations of attacks are performed from $R$ initial points for each pair of search direction and objective function. The search point with the highest CW loss ($L_{\mathrm{CW}}$) value is collected at the end of the attack, which starts at each initial point. The obtained set of these search points is denoted as $X_i^a$. We focus on this set of search points to analyze the characteristics of the attack using a single $\boldsymbol{\delta}$ and $L$. For this experiment, we used 10,000 images from CIFAR-10 (Krizhevsky et al., 2009) as test samples and attacked the robust model proposed by Sehwag et al. (2022). See Appendix C for more information and results.

**Notation.** Let $\mathcal{L} = \{L_{\mathrm{CE}}, L_{\mathrm{CW}}, L_{\mathrm{SCW}}, L_{\mathrm{DLR}}\} \cup \{L_{\mathrm{G\text{-}DLR},q} \mid q = 4, 5, 6\}$ be a set of objective functions, and $\mathcal{D} = \{\boldsymbol{\delta}_{\mathrm{PGD}}, \boldsymbol{\delta}_{\mathrm{APGD}}, \boldsymbol{\delta}_{\mathrm{ACG}}, \boldsymbol{\delta}_{\mathrm{Nes}}\}$ be a set of search directions. We have proposed $\boldsymbol{\delta}_{\mathrm{Nes}}$ and $L_{\mathrm{G\text{-}DLR},q}$ in this study. $\boldsymbol{\delta}_{\mathrm{Nes}}$ is the search direction of Nesterov's accelerated gradient (NAG) (Nesterov, 2004) normalized by the sign function to accommodate $\ell_\infty$ attacks. We refer to $L_{\mathrm{G\text{-}DLR},q}$ as generalized-DLR (G-DLR) loss with the denominator of DLR loss extended from $g_{\pi_1}(\boldsymbol{x}) - g_{\pi_3}(\boldsymbol{x})$ to $g_{\pi_1}(\boldsymbol{x}) - g_{\pi_q}(\boldsymbol{x})$; $\pi_q \in Y$ denotes the class label that has the $q$-th largest value of $g(\boldsymbol{x})$. Mathematical expressions of $\boldsymbol{\delta}_{\mathrm{Nes}}$ and $L_{\mathrm{G\text{-}DLR},q}$ can be found in Appendix B. Given an initial point sampling $\mathbf{x}$ and a step size update rule $\eta$, we define a set of attacks

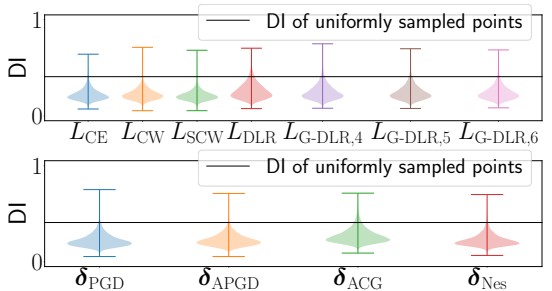

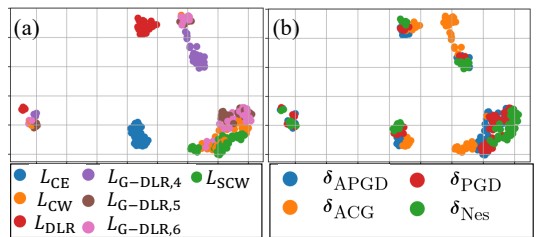

Figure 3: Violin plot of $DI(X_i^a)$. The upper/lower figure represents the attack with the same objective function/search direction, respectively.

Figure 4: 2D visualization of $X_i^a$ using UMAP. Points of the same color in (a)/(b) represent points obtained using the same objective function/search direction, respectively.

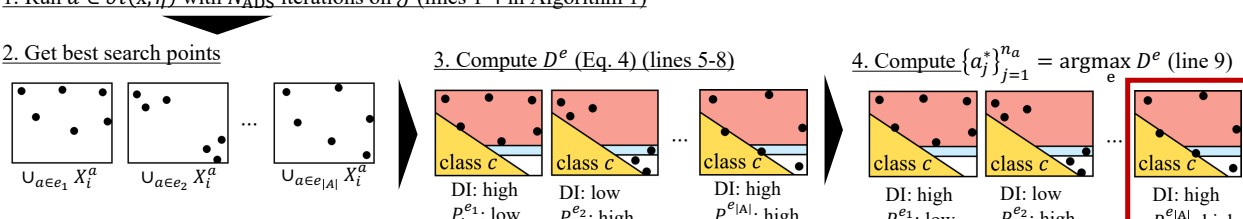

Figure 5: The procedure of ADS. The black circle represents a single search point, and DI tends to be smaller when the search points form clusters. Regions of different colors represent different classification classes, and the more search points are distributed around multiple regions, the higher the degree of diversification in the output space, i.e., the value of $P_i^e$.

as $\mathcal{A}(\mathbf{x}, \eta) = \{a = (\mathbf{x}, \eta, \boldsymbol{\delta}, L) \mid \boldsymbol{\delta} \in \mathcal{D}, L \in \mathcal{L}\}$. In the following paragraphs, we analyze the characteristics of attacks $a \in \mathcal{A}(\mathbf{x}_{\mathrm{ODS}}, \eta_{\cos})$, which use single $\boldsymbol{\delta}$ and $L$, based on the experimental results with $N_{\max} = 30$ and $R = 10$.

**Limited diversification ability of attacks using single $\boldsymbol{\delta}$ and $L$.** We quantify the diversity of $X_i^a$ using DI to reveal the diversification ability of the attack $a$, which uses a single $\boldsymbol{\delta}$ and $L$. Figure 3 shows the violin plot of $DI(X_i^a)$ for 10,000 images and all attacks $a$. The mean and standard deviation of the first, second, and third quartiles were $0.190 \pm 0.019$, $0.223 \pm 0.023$, and $0.269 \pm 0.033$, respectively. Figure 3 and these DI values suggest that the diversity of the best point set $X_i^a$ is relatively low. Thus, the diversification ability of $a$ seems to be limited.

**Attacks using multiple $\boldsymbol{\delta}$ and $L$ can lead to efficient diversification.** Figure 4 shows the best point set $X_i^a$ embedded in a two-dimensional space using Uniform Manifold Approximation and Projection (UMAP) (McInnes et al., 2018). Dimensionality reduction methods, such as UMAP, preserve the maximum possible distance information in high-dimensional spaces as possible. Figure 4 (a) depicts the points obtained by the attack using the same $L$ in the same color. Figure 4 (a) shows that sets of best search points obtained from searches with different $L$ tend to form different clusters. Figure 4 (b) depicts the points obtained by the attack using the same $\boldsymbol{\delta}$ in the same color. Similarly, sets of best search points obtained by the attack using different $\boldsymbol{\delta}$ also tend to form different clusters. Based on these observations, it is possible to efficiently search for different local solutions using different $\boldsymbol{\delta}$ and $L$, or an appropriate combination of both.

**Algorithm 2** Automated Diversified Selection (ADS)

**Require:** $\mathcal{J}$: small samples, $\mathcal{A}(\mathbf{x}, \eta)$: candidate attacks, $N_{\text{ADS}}$: maximum iterations, $P_{\mathcal{S}}$: projection, $g$: DNN, $M$: the size of feasible region, $n_a$: number of attacks to select

**Ensure:** $\{(\boldsymbol{\delta}_{a_j^*}, L_{a_j^*})\}_{j=1}^{n_a}$: The pairs of search direction and objective function

1: **for** $a = (\mathbf{x}, \eta, \boldsymbol{\delta}, L) \in \mathcal{A}(\mathbf{x}, \eta)$ **do**
2:    $X^a, \Pi^a \leftarrow \text{Attack}(a, N_{\text{ADS}}, P_{\mathcal{S}}, g, \mathcal{J})$ /*Algorithm 1*/
3: **end for**
4: $A \leftarrow \{e \subset \mathcal{A}(\mathbf{x}, \eta) \mid |e| = n_a \wedge |\{L_{a_j}\}_{j=1}^{n_a}| = n_a\}$
5: **for** $e \in A$ **do**
6:    $P_i^e \leftarrow |\cup_{a \in e} \Pi_i^a|, \forall i = 1, \ldots, |\mathcal{J}|$
7:    Compute $D^e$ by equation 4
8: **end for**
9: Get $\{(\boldsymbol{\delta}_{a_j^*}, L_{a_j^*})\}_{j=1}^{n_a}$ from $\{a_j^*\}_{j=1}^{n_a} = \arg\max_{e \in A} D^e$.

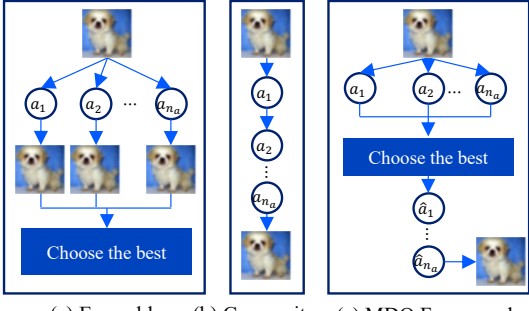

(a) Ensemble    (b) Composite    (c) MDO Framework

Figure 6: Illustration of the ensemble, composite, and the MDO framework.

## 3.2 Automated Diversified Selection

The analyses in Section 3.1 suggest that the MDO strategy can efficiently search for different local solutions. However, this strategy is ineffective unless the combinations of $\boldsymbol{\delta}$ and $L$ are properly determined because different attacks may search similar regions. To address this issue, we propose Automated Diversified Selection (ADS), which selects the combinations of search directions and objective functions based on the degree of diversification in input and output spaces. Algorithm 2 and Figure 5 show the ADS procedure. Ideally, we aim to identify the combination of $\boldsymbol{\delta}$ and $L$ that facilitates the most extensive exploration through numerous iterations of candidate attacks on the entire dataset. In practice, however, computation time should be minimized. Therefore, we attempt to approximate the combination of $\boldsymbol{\delta}$ and $L$ capable of exploring extensive regions by assessing the diversity of the best solutions derived from a limited iteration of attacks on small samples. The ADS procedure is outlined below.

First, $N_{\text{ADS}}$ iterations of attack candidates $a \in \mathcal{A}(\mathbf{x}, \eta)$ are executed on the image set $\mathcal{J} \subset D$ (lines 1-3 in Algorithm 2). In this study, $\mathcal{J}$ is 1% of the images uniformly sampled from all the test samples. Subsequently, the set of best search points $X_i^a$ and class labels $\Pi_i^a = \{\arg\max_{q \neq c_i} g_q(\boldsymbol{x}^{(k)}) \in Y \mid k = 1, \ldots, N_{\text{ADS}}\}$ are obtained. $X_i^a$ and $\Pi_i^a$ are constructed as described in lines 10 and 8 of Algorithm 1, respectively. The observations in Section 3.1 suggest that the degree of diversification may be greatly reduced when the selected attacks employ the same objective function. Therefore, a set of the candidate combinations of $n_a$ attacks is defined as $A = \{\{a_j\}_{j=1}^{n_a} \subset \mathcal{A}(\mathbf{x}, \eta) \mid |\{L_{a_j}\}_{j=1}^{n_a}| = n_a\}$ so that each attack uses a different $L$ (line 4 in Algorithm 2). The weighted average of the DI is calculated for all $e \in A$ to quantify the diversity of the best point set $\cup_{a \in e} X_i^a$ as follows:

$$D^e = \frac{1}{|\mathcal{J}|} \sum_{i=1}^{|\mathcal{J}|} P_i^e \cdot DI(\cup_{a \in e} X_i^a), \tag{4}$$

where $P_i^e = |\cup_{a \in e} \Pi_i^a|$ is the number of types of classification labels with the highest prediction probability excluding the correct label (lines 5-8 in Algorithm 2). A high DI indicates a high diversity of $\cup_{a \in e} X_i^a$, and a high $P_i^e$ indicates a high output diversity. Finally, ADS outputs $\{(\boldsymbol{\delta}_{a_j^*}, L_{a_j^*})\}_{j=1}^{n_a}$ as the appropriate combinations of $\boldsymbol{\delta}$ and $L$, where $\{a_j^*\}_{j=1}^{n_a} = \arg\max_{e \in A(\mathbf{x}, \eta)} D^e$ (line 9 in Algorithm 2). The algorithm is designed on the assumption that the best search points found through appropriate diversification are unlikely to form clusters. Therefore, ADS selects combinations based on DI, which can directly quantify whether a point set contains clusters or not. Because there is no general definition of appropriate diversification, a selection method based on indices other than DI is also possible. The experimental results suggest that selecting combinations based on DI is a reasonable approach.

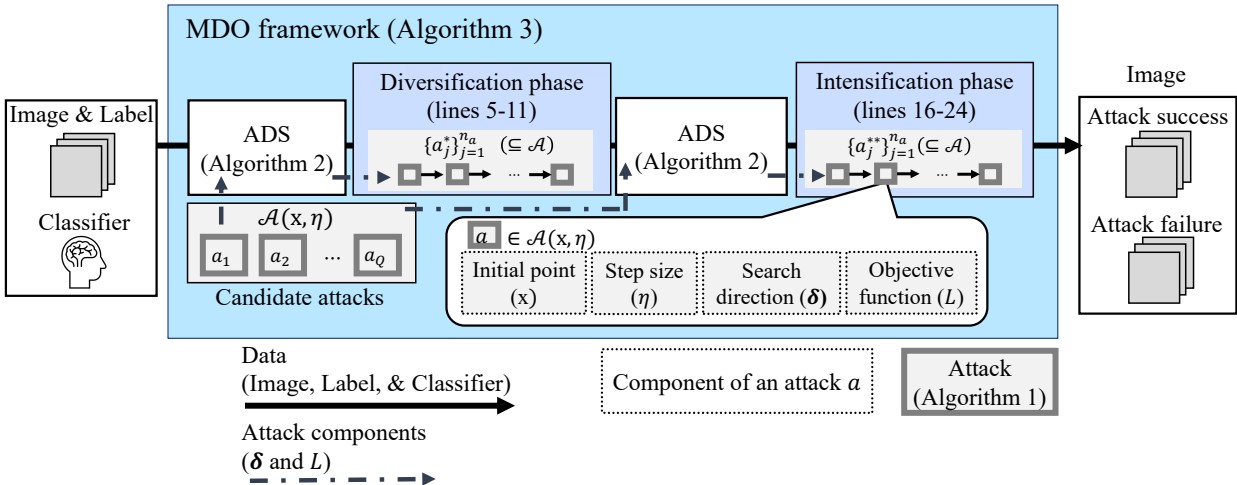

Figure 7: Flowchart of MDO framework.

## 3.3 Search framework for the MDO strategy

Considering the difference in diversification/intensification performance between PGD and CG-based search directions reported by Yamamura et al. (2022), we propose a search framework called the MDO framework consisting of diversification and intensification phases. MDO framework is a combination of ensemble and composite. Figure 6 illustrates the procedure of the ensemble, composite, and MDO framework. An ensemble realizes the diversification phase, and the intensification phase is based on a composite. The diversification phase aims to find appropriate starting points via extensive search within the feasible region, and the intensification phase aims to improve the objective value through intensive searches of nearby areas around the best point obtained during the diversification phase. As demonstrated in Figure 12, DI of best search points obtained by an ensemble/a composite tends to be relatively high/low. The high/low DI value indicates that the corresponding search procedure is extensive/intensive. Thus, an ensemble is suitable for diversification, and a composite is suitable for intensification. The pseudocode of the MDO framework is described in Algorithm 3. The overview of the MDO framework is also illustrated in Figure 7. Assume that the total number of the MDO framework iterations is $N \times n_a$. Inspired by the APGD step size control, the diversification phase takes $\lceil 0.41N \rceil \times n_a$ iterations, and the intensification phase uses the remaining iterations.

**Diversification phase.** The diversification phase uses large step sizes to search a broader area. First, ADS is executed with a fixed step size of $2\varepsilon$ and initial point sampling x to determine the pairs $\{(\boldsymbol{\delta}_{a_j^*}, L_{a_j^*})\}_{j=1}^{n_a}$, where $(a_1^*, a_2^*, \ldots, a_{n_a}^*) \subset \mathcal{A}(\mathbf{x}_{\text{init}}, \eta_{\text{fix}})$ (line 4 in Algorithm 3). For each pair of $\boldsymbol{\delta}_{a_j^*}$ and $L_{a_j^*}$, $N_1 = \lceil 0.41N \rceil$ iterations of the attack are executed with an initial step size of $2\varepsilon$ starting at an initial point selected by $\mathbf{x}_{\text{init}}$ (lines 6-11 in Algorithm 3). The images whose adversarial example is found are excluded from the attack target (line 10 in Algorithm 3) because we assume the robustness evaluation. The step size is updated by $\eta_{\text{APGD}}$. When the total iteration is $N_{iter}$, APGD step size control allocates at least $\lceil 0.22 N_{iter} \rceil$ iterations with a step size of $2\varepsilon$ and $\lceil (0.41 - 0.22)N_{iter} \rceil = \lceil 0.19 N_{iter} \rceil$ iterations with a step size of $2\varepsilon$ or $\varepsilon$. Inspired by this, the diversification phase uses the total iterations of $N_1$ for each attack and the checkpoints of $W = \{\lceil 0.22 \times N \rceil\}$ for step size control $\eta_{\text{APGD}}$ to achieve diversification with large step sizes. The diversification phase aims to identify promising starting points through extensive exploration. Therefore, we need to thoroughly explore the neighborhood of these identified points to enhance the objective value further. To achieve this, we introduce the intensification phase subsequently.

**Intensification phase.** The intensification phase takes the solution with the highest $L_{\text{CW}}$ value found by the diversification phase as the initial point and searches for different local solutions within a range not far from the initial point. In APGD step size control, the step size is set to $\varepsilon/2$ after the search with a step size

---

**Algorithm 3** MDO framework

---

**Require:** $\mathcal{I}$: test samples, $g$: DNN, $\varepsilon$: allowed perturbation size, $n_a$: number of pairs to select, $N_{\text{ADS}}$: search iterations in ADS, $M$: the size of the feasible region, $N$: search iterations per attack

**Ensure:** $X^{\text{advs}}$: adversarial examples

1: $X^{\text{advs}} \leftarrow \emptyset$
2: /* Diversification phase */
3: $\mathcal{J} \leftarrow$ uniformly sampled 1% images of $\mathcal{I}$.
4: $\{(\boldsymbol{\delta}_{a_j^*}, L_{a_j^*})\}_{j=1}^{n_a} \leftarrow \text{ADS}\left(\mathcal{J}, \mathcal{A}(\mathbf{x}_{\text{init}}, \eta_{\text{fix}}), N_{\text{ADS}}, P_{\mathcal{S}}, g, M, n_a\right)$
5: $N_1 \leftarrow \lceil 0.41N \rceil, N_2 \leftarrow N - N_1$
6: **for** $j = 1, \ldots, n_a$ **do**
7: $\quad a' \leftarrow (\mathbf{x}_{\text{init}}, \eta_{\text{APGD}}, \boldsymbol{\delta}_{a_j^*}, L_{a_j^*})$
8: $\quad X^{a'}, \Pi^{a'} \leftarrow \text{Attack}(a', N_1, P_{\mathcal{S}}, g, \mathcal{I})$ /*Algorithm 1*/
9: $\quad$ Update $X^{\text{advs}}$ by $X^{a'}$
10: $\quad$ Update $\mathcal{I}$ by excluding images that succeeded in the attack.
11: **end for**
12: /* Intensification phase */
13: $\mathcal{I}_{sub} \leftarrow \mathcal{I}$
14: $\mathcal{J}' \leftarrow$ uniformly sampled 1% images of $\mathcal{I}$.
15: $\{(\boldsymbol{\delta}_{\hat{a}_j}, L_{\hat{a}_j})\}_{j=1}^{n_a} \leftarrow \text{ADS}\left(\mathcal{J}', \mathcal{A}(\mathbf{x}_{\text{best}}, \eta_{\text{fix}}), N_{\text{ADS}}, P_{\mathcal{S}}, g, M, n_a\right)$
16: **for** $j = 1, \ldots, n_a$ **do**
17: $\quad a^l \leftarrow (\mathbf{x}_{\text{best}}, \eta_{\text{cos}}, \boldsymbol{\delta}_{\hat{a}_j}, L_{\hat{a}_j})\{$Use $\eta_{\text{APGD}}$ instead of $\eta_{\text{cos}}$ when $\boldsymbol{\delta}_{\hat{a}_j} = \boldsymbol{\delta}_{\text{ACG}}\}$
18: $\quad X^{a^l}, \Pi^{a^l} \leftarrow \text{Attack}(a^l, N_2, P_{\mathcal{S}}, g, \mathcal{I}_{sub})$ /*Algorithm 1*/
19: $\quad$ Update $X^{\text{advs}}$ by $X^{a^l}$.
20: $\quad$ /* Extract images whose highest CW loss value is close to 0 for further intensification. */
21: $\quad \mathcal{I}_{sub} \leftarrow \mathcal{I}_{sub} \cap \left\{ \boldsymbol{x}_i \in \mathcal{I} \mid -0.05 \leq \max_{\boldsymbol{x}^* \in X_i^{a^l}} L_{\text{CW}}(g(\boldsymbol{x}^*), c_i) \leq 0 \right\}$
22: $\quad X^{a^l}, \Pi^{a^l} \leftarrow \text{Attack}(a^l, N_2, P_{\mathcal{S}}, g, \mathcal{I}_{sub})$ /*Algorithm 1*/
23: $\quad$ Update $X^{\text{advs}}$ by $X^{a^l}$.
24: **end for**

---

of $\varepsilon$. Based on this, the initial step size is set to $\varepsilon/2$. First, ADS is executed with a fixed step size of $\eta = \varepsilon/2$ to determine the pairs $\{(\boldsymbol{\delta}_{\hat{a}_j}, L_{\hat{a}_j})\}_{j=1}^{n_a}$ (line 15 in Algorithm 3), where $(\hat{a}_1, \hat{a}_2, \ldots, \hat{a}_{n_a}) \subset \mathcal{A}(\mathbf{x}_{\text{best}}, \eta_{\text{fix}})$, and $\mathbf{x}_{\text{best}}$ denotes the initial point sampling that uses the solution with the highest $L_{\text{CW}}$ value as the initial point. The experimental results in Table 6 demonstrate the effectiveness of the ADS execution in this phase (line 15 in Algorithm 3). For each pair of $(\boldsymbol{\delta}_{\hat{a}_j}, L_{\hat{a}_j})$, $N_2 = N - N_1$ iterative searches are performed with the initial point determined by $\mathbf{x}_{\text{best}}$, and the initial step size of $\varepsilon/2$, using step size update rule $\eta_{\text{cos}}$ (lines 16-19 in Algorithm 3). $\mathbf{x}_{\text{best}}$ enables composite execution of the attack sequence. The step size update rule $\eta_{\text{cos}}$ leads to a larger step size than $\eta_{\text{APGD}}$ in nature. Thus, $\eta_{\text{cos}}$ is expected to avoid getting stuck during intensification employing the steepest descent-based search directions. On the contrary, $\eta_{\text{cos}}$ may result in insufficient intensification using a diversified search direction such as $\boldsymbol{\delta}_{\text{ACG}}$. For this reason, $\eta_{\text{APGD}}$ is used to update the step size when the search direction $\boldsymbol{\delta}_{\hat{a}_j}$ is equal to $\boldsymbol{\delta}_{\text{ACG}}$, aiming for a better intensification performance (line 17 in Algorithm 3). Subsequently, the same searches are performed on the images whose highest $L_{\text{CW}}$ values are greater than or equal to $-0.05$ to accelerate the intensification (lines 20-24 in Algorithm 3). Lines 20-24 are introduced to avoid insufficient optimization, one cause of attack failure pointed out by Pintor et al. (2022). Same as in the diversification phase, successfully perturbed images, i.e. $\boldsymbol{x}_i \in \mathcal{I}_{sub}$ s.t. $\max_{\boldsymbol{x}^* \in X_i^{a^l}} L_{\text{CW}}(g(\boldsymbol{x}^*), c_i) > 0$, are excluded (line 23 in Algorithm 3).

### 3.4 Efficient Diversified Attack

The comparisons of the MDO framework and $\text{MT}_{\text{cos}}$ in Section 4.2 suggest that combining MDO and MT strategies would lead to a faster and stronger attack. This result motivates us to propose Efficient Diversified Attack (EDA), a combination of MDO and MT strategies. EDA is implemented as an ensemble of the MDO framework and a targeted attack $a^T = (\mathbf{x}_{\text{init}}, \eta_{\text{APGD}}, \boldsymbol{\delta}_{\text{GD}}, L_{\text{CW}}^T)$ with $N$ iterations. We selected this targeted

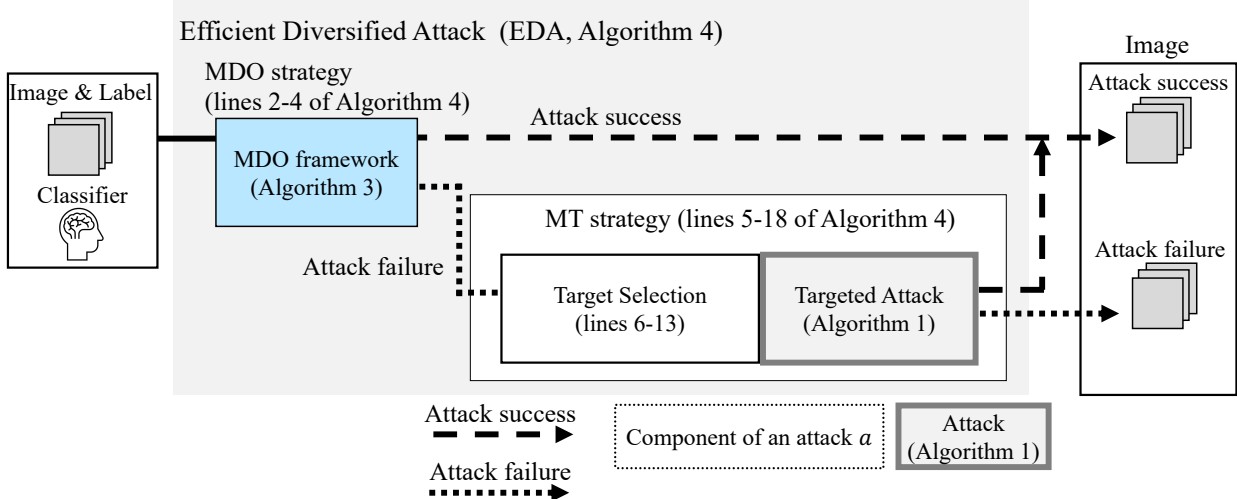

Figure 8: Flowchart of EDA.

attack configuration based on small experiments in Appendix G.1. Considering computational efficiency, we estimate the most likely to cause misclassification target class $T_i$ for the targeted attack through a few iterations of the multi-targeted attack. This procedure is referred to as a small-scale search (lines 6-13 in Algorithm 4). See Appendix F for validation of the target selection scheme based on the small-scale search. The pseudocode of EDA is described in Algorithm 4. Figure 8 provides an overview of EDA. EDA executes the MDO framework first (lines 1-4 in Algorithm 4). Same as in the MDO framework, successfully perturbed images are excluded because we assume the robustness evaluation (lines 4 in Algorithm 4). Subsequently, the target class $T_i$ is selected based on the small-scale search from top-$K$ classes (lines 9-13 in Algorithm 4). Existing researches support the effectiveness of the multi-targeted attack with top-$K$ classes Croce & Hein (2020b); Gao et al. (2022). Finally, the targeted attack $a^T$ with selected target $T_i$ is executed (lines 14-18 in Algorithm 4). Lines 1-4 are the MDO strategy, and lines 5-18 represent the MT strategy. MT strategy is independent of the MDO strategy. Thus, the execution order of MDO and MT strategies is not so important.

## 4    Experiments

The efficacy of the proposed methods was examined through a series of experiments involving an $\ell_\infty$ attack against $\ell_\infty$ defense models listed in RobustBench (Croce et al., 2021).

**Dataset and models.**    We used 41 models and 21 different defenses[1], including 25 models trained on CIFAR-10, 11 on CIFAR-100 (Krizhevsky et al., 2009), and five on ImageNet (Russakovsky et al., 2015). We also used 10 additional models trained on ImageNet with three different defenses to test the EDA's performance against more advanced models, including transformer-based architectures. We performed $\ell_\infty$ attacks on 10,000 images with $\varepsilon = 8/255$ for CIFAR-10 and CIFAR-100 models and on 5,000 images with $\varepsilon = 4/255$ for ImageNet models. These $\varepsilon$ values are used in the RobustBench leaderboard. The test images were sampled in the same way as in RobustBench. The text presents results against nine representative models for CNN-based models. Complete results for CNN-based models are described in Appendix D.

**Computer specification.**    The experiments were conducted with two types of CPUs and a single type of GPU. The CPUs used in the experiments were Intel(R) Xeon(R) Gold 6240R CPU @ 2.40GHz and Intel(R) Xeon(R) Silver 4216 CPU @ 2.10GHz. The GPU used in the experiments is NVIDIA GeForce RTX 3090.

---

[1]The used models are publicly available as of robustbench v1.1.

---

**Algorithm 4** Efficient Diversified Attack (EDA)

---

**Require:** $\mathcal{I}$: test set, $g$: DNN, $\varepsilon$: allowed perturbation size, $n_a$: number of pairs to select, $K$: number of target classes, $N_{\mathrm{ADS}}$: search iterations in ADS, $M$: the size of the feasible region, $N$: search iterations per attack $a$, $N_s$: number of iterations in small-scale search

**Ensure:** $X^{\mathrm{advs}}$: Adversarial examples

1: $X^{\mathrm{advs}} \leftarrow \emptyset$
2: /* Run MDO framework (Algorithm 3) */
3: $X^{\mathrm{advs}} \leftarrow$ MDO framework$(\mathcal{I}, g, \varepsilon, n_a, N_{\mathrm{ADS}}, M, N)$
4: Update $\mathcal{I}$ by excluding images that succeeded in the attack.
5: /* Begin multi-targeted strategy */
6: /* Select target class based on small-scale search */
7: $T_i \leftarrow \pi_1, (\forall i = 1, \ldots, |\mathcal{I}|)$
8: $a^t \leftarrow (\mathbf{x}_{\mathrm{init}}, \eta_{\cos}, \boldsymbol{\delta}_{\mathrm{GD}}, L_{\mathrm{CW}}^T)$
9: **for** $t = 1, \ldots K$ **do**
10:     /* Run targeted attack with the target class $\pi_t$ */
11:     $X^{a^t}, \Pi^{a^t} \leftarrow$ Attack$(a^t, N_s, P_{\mathcal{S}}, g, \mathcal{I})$ /*Algorithm 1*/
12:     Update target class $T_i$ for images $\boldsymbol{x}_i$ that record the highest loss value
13: **end for**
14: $a^T \leftarrow (\mathbf{x}_{\mathrm{init}}, \eta_{\mathrm{APGD}}, \boldsymbol{\delta}_{\mathrm{GD}}, L_{\mathrm{CW}}^T)$
15: /* Run targeted attack with the target $T_i$ */
16: $X^{a^T}, \Pi^{a^T} \leftarrow$ Attack$(a^T, N, P_{\mathcal{S}}, g, \mathcal{I})$ /*Algorithm 1*/
17: /* End multi-targeted strategy */
18: Update $X^{\mathrm{advs}}$ by $X^{a^T}$.

---

Table 1: Comparison in robust accuracy. The lower the robust accuracy, the higher the attack performance. The lowest robust accuracy is in **bold**. The "#query" row shows the number of queries in the worst-case per image. The robust accuracy of AA is the value reported by RobustBench. MDO represents the MDO framework. OOM (Out of Memory) indicates that execution could not be completed due to a memory error.

| Dataset | No. | Defense | clean | AA | ACG | $\mathrm{MT}_{\cos}$ | CAA | AutoAE | $\mathrm{A}^3$ | MDO | EDA |
|---|---|---|---|---|---|---|---|---|---|---|---|
| #query | | | | 6.1k | 500 | 504 | 800 | 3.6k | 1k | 502.24 | 802.24 |
| CIFAR-10 | 1 | (Sehwag et al., 2022) | 84.59 | 55.54 | 56.19 | 55.54 | 55.52 | 55.51 | 55.53 | 55.58 | **55.49** |
| | 2 | (Carmon et al., 2019) | 89.69 | 59.53 | 60.10 | 59.54 | 59.50 | 59.47 | 59.44 | 59.46 | **59.40** |
| ($\varepsilon = 8/255$) | 3 | (Rebuffi et al., 2021) | 88.54 | 64.25 | 64.80 | 64.28 | 64.23 | **64.19** | 64.24 | 64.32 | 64.20 |
| CIFAR-100 | 4 | (Rice et al., 2020) | 53.83 | 18.95 | 19.48 | 18.99 | 18.96 | 18.91 | 18.89 | 18.97 | **18.88** |
| | 5 | (Sitawarin et al., 2021) | 62.82 | 24.57 | 25.69 | 24.55 | 24.56 | 24.52 | 24.56 | 24.65 | **24.50** |
| ($\varepsilon = 8/255$) | 6 | (Gowal et al., 2020) | 69.15 | 36.88 | 37.84 | 36.95 | 36.90 | 36.86 | 36.87 | 36.96 | **36.81** |
| ImageNet | 7 | (Salman et al., 2020) | 52.92 | 25.32 | 26.40 | 25.24 | 25.27 | OOM | 25.22 | 25.22 | **25.11** |
| | 8 | (Engstrom et al., 2019) | 62.56 | 29.22 | 31.54 | 29.34 | 29.41 | OOM | 29.32 | 29.20 | **29.01** |
| ($\varepsilon = 4/255$) | 9 | (Wong et al., 2020) | 55.62 | 26.24 | 28.46 | 26.40 | 26.56 | OOM | 26.42 | 26.22 | **26.12** |
| Summary (# **bold** / 41 models) | | | | 0 | 0 | 0 | 0 | 2 | 3 | 0 | 36 |

When we compared the performance of each attack on model A and dataset B, all compared attacks were run on the same device. The runtime comparison is thus fair.

**Hyperparameters.** The parameters of the ADS are $n_a = 5$ and $N_{\mathrm{ADS}} = 4$, which are the number of pairs of $\boldsymbol{\delta}$ and $L$ and the number of iterations, respectively. These parameters were determined based on small-scale experiments in Appendix D.1. The parameter of the MDO framework is $N = 100$. The total number of the MDO framework iterations is $n_a \times N = 500$. EDA has parameters $N_s = 10$ and $K$ in addition to ADS and the MDO framework parameters. The experiments used $K = 9, 14$, and $20$ for CIFAR-10, CIFAR-100, and ImageNet, respectively. Unless otherwise noted, the initial point sampling ($\mathbf{x}_{\mathrm{init}}$) in ADS, the diversification phase, and the targeted attack was Prediction Aware Sampling ($\mathbf{x}_{\mathrm{PAS}}$, PAS), a variant of ODS described in Appendix E.

Table 2: The comparison in the robust accuracy against robust models trained on ImageNet, including transformer-based architectures. The robust accuracy of AA is the value reported in RobustBench. The lowest robust accuracy is in bold. "sec (ratio)" columns show the runtime in seconds and the ratio of runtime to EDA.

| Defense | Model | clean | AA | CAA | | $A^3$ | | EDA | |
|---------|-------|-------|-----|-----|------|-----|------|-----|------|
| | | acc | acc | acc | sec (ratio) | acc | sec (ratio) | acc | sec (ratio) |
| (Liu et al., 2023a) | ConvNeXt-B | 76.02 | 55.82 | 55.86 | 20,244 (1.4) | 55.84 | 37,437 (2.7) | **55.78** | 14,073 (1.0) |
| (Liu et al., 2023a) | Swin-B | 76.16 | 56.16 | 56.14 | 25,913 (1.9) | 56.06 | 20,620 (1.5) | **56.04** | 13,988 (1.0) |
| (Debenedetti et al., 2023) | XCiT-L12 | 73.76 | 47.60 | 47.70 | 25,196 (2.0) | 47.54 | 20,044 (1.6) | **47.44** | 12,705 (1.0) |
| (Debenedetti et al., 2023) | XCiT-M12 | 74.04 | 45.24 | 45.44 | 13,187 (1.7) | 45.22 | 11,067 (1.4) | **45.14** | 7,884 (1.0) |
| (Debenedetti et al., 2023) | XCiT-S12 | 72.34 | 41.78 | 41.88 | 8,690 (1.6) | 41.74 | 6,560 (1.2) | **41.60** | 5,548 (1.0) |
| (Singh et al., 2023) | ConvNeXt-B+ConvStem | 75.90 | 56.14 | 56.12 | 21,650 (1.5) | 56.14 | 35,422 (2.5) | **56.02** | 14,399 (1.0) |
| (Singh et al., 2023) | ConvNeXt-S+ConvStem | 74.10 | 52.42 | 52.44 | 14,184 (1.3) | 52.38 | 24,512 (2.3) | **52.30** | 10,601 (1.0) |
| (Singh et al., 2023) | ConvNeXt-T+ConvStem | 72.72 | 49.46 | 49.42 | 8,444 (1.2) | 49.44 | 19,391 (2.7) | **49.38** | 7,311 (1.0) |
| (Singh et al., 2023) | ViT-B+ConvStem | 76.30 | 54.66 | 54.72 | 22,262 (1.9) | 55.02 | 17,141 (1.4) | **54.62** | 11,962 (1.0) |
| (Singh et al., 2023) | ViT-S+ConvStem | 72.56 | 48.08 | 48.18 | 8,912 (1.7) | 48.08 | 5,928 (1.1) | **48.02** | 5,355 (1.0) |

## 4.1 Comparison with recent attacks

The comparative experiments of EDA with the standard version of AA, CAA, AutoAE, and $A^3$ were conducted to investigate the performance of EDA. The parameters of the existing methods were the default values in their official codes. In addition, $MT_{cos}$, a step size variant of MT-PGD (Gowal et al., 2019), was compared with the MDO framework to analyze the characteristics of the MDO strategy. In our notation, $MT_{cos}$ is expressed as $(x_{org}, \eta_{cos}, \delta_{GD}, L_{CW}^T)$. The parameters of $MT_{cos}$ were the number of target classes, $K = 9$, and iterations per target class, $N_T = 56$. The MDO framework and $MT_{cos}$ thus spent almost the same number of queries. To investigate the stability of the compared methods including $A^3$ and EDA, we ran five times with different random seeds for the experiments on 41 CNN-based models. The results suggest a stable performance. Table 1 shows the mean of five runs. See Table 11 in the Appendix for the experimental results including standard deviations. The remaining experiments were conducted with a single fixed random seed.

**EDA showed SOTA performance in less runtime under the standard $\varepsilon$ setting in RobustBench**. The summary in Table 1 shows that the attack performance of EDA exceeds that of recently proposed methods for around 88% of 41 CNN-based robust models. We could not finish AutoAE execution against ImageNet models due to the out-of-memory error. As shown in Figure 1, EDA showed lower robust accuracy with fewer numbers of worst-case queries than AA, CAA, AutoAE, and $A^3$. In addition, Figure 9 shows that EDA is 1.07-2.28, 1.46-2.87, and 18.74-38.61 times faster than $A^3$, CAA, and AutoAE on average. $A^3$ stops the search before the given query budget, depending on the models. Because of this, $A^3$ spent less time than EDA for some models, specifically for CIFAR-10 models. The reasons for better performance and less runtime of EDA are the complementarity of MDO and multi-targeted strategies (Section 4.2) and a higher degree of diversification of EDA (Figure 12), which lead to efficient diversification and intensification. The differences in robust accuracy among CAA, AutoAE, and $A^3$ are approximately 0.05%. Considering these advances in robust accuracy by recent attacks, EDA's performance improvements, around 2 to 30 times speed-up and approximately 0.01 to 0.21% improvement in robust accuracy, are sufficiently large. As shown in Table 2, EDA also demonstrated better performance against 100% of 10 advanced models, including transformer-based architectures. We discuss the complexity of the compared methods in the number of queries in Appendix D.2 in detail. In the following analyses of EDA, we mainly focus on the comparison with $A^3$, which is sufficiently fast and strong among baseline methods.

**EDA showed higher transferability.** We chose three recent models trained on CIFAR-10, including Semi (Carmon et al., 2019), MMA (Ding et al., 2020), and LBGAT (Cui et al., 2021). Table 3 shows that EDA tends to exhibit the highest performance among baseline methods in the transfer scenarios from more robust models to less robust models. EDA showed the second-best performance in the other scenarios.

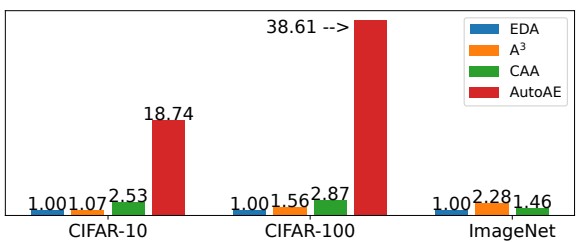

Figure 9: The average ratio of computation time to EDA.

Table 3: Comparison in robust accuracy for transfer setting. "A→B" indicates the transfer attack from model A to model B.

| Transfer | clean | AA | CAA | AutoAE | A³ | EDA |
|---|---|---|---|---|---|---|
| Semi→MMA | 84.36 | 71.82 | 73.15 | 73.24 | 70.46 | **68.14** |
| Semi→LBGAT | 88.22 | 71.90 | 74.76 | 74.66 | 73.46 | **70.29** |
| MMA→Semi | 89.69 | **83.23** | 85.01 | 84.85 | 85.16 | 83.68 |
| MMA→LBGAT | 88.22 | **79.77** | 82.16 | 81.83 | 82.42 | 80.32 |
| LBGAT→Semi | 89.69 | **76.17** | 80.24 | 80.32 | 79.94 | 77.24 |
| LBGAT→MMA | 84.36 | 70.16 | 72.07 | 72.09 | 71.28 | **69.43** |

| | | LBGAT | MMA | Semi |
|---|---|---|---|---|
| Best known robust accuracy in RobustBench leaderboard | | 52.86 | 41.44 | 59.53 |

Figure 10: Visualization of adversarial examples ($\varepsilon = 4/255$ on ImageNet). "2-norm" denotes the perturbation size in terms of Euclidean distance. EDA and CAA succeeded in the attack, but A³ failed.

**Visualization of the adversarial examples.** Figure 10 shows the visualization of the generated adversarial examples. EDA and CAA, which induced misclassification, perturbed the original image more extensively than A³ in terms of Euclidean distance. This indicates that EDA explored a broader area than A³, leading to a successful attack. The DI-based analysis in Section 4.1 also suggests a higher diversification performance of EDA than A³. Additionally, EDA perturbed the image to a lesser extent than CAA, yet both methods led to misclassification. This suggests that EDA is more likely to generate adversarial examples that are closer to the original image than CAA. This study does not focus on the perturbation size in the Euclidean norm, but it represents an advantageous property for the imperceptibility of perturbations.

**Performance evaluation with different perturbation bounds $\varepsilon$.** As shown in Figure 11, for all datasets, the performance difference between the compared attacks is very small when epsilon is small. Regarding the runtime, EDA spent slightly less time than A³. When epsilon is large, A³ tends to have

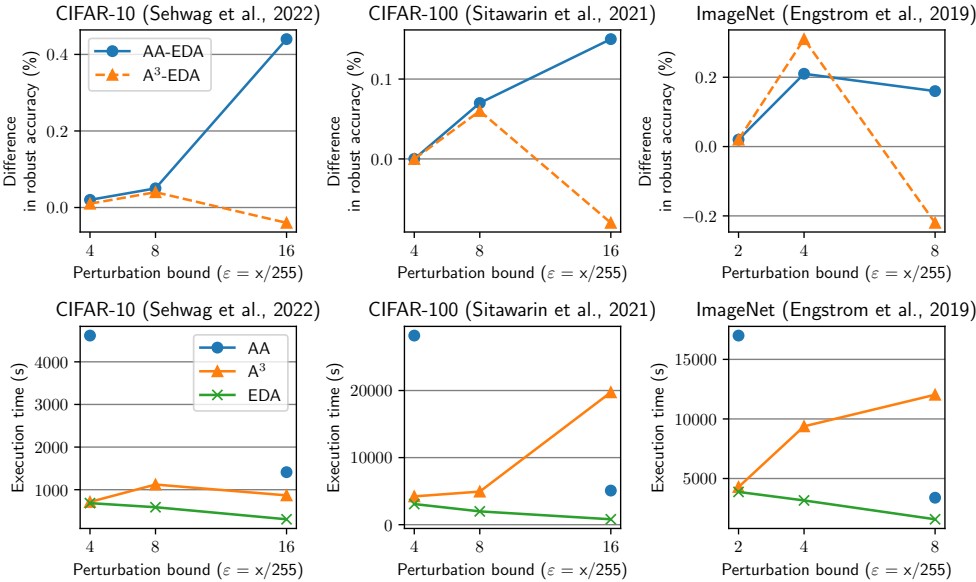

Figure 11: Difference in robust accuracy and runtime with different perturbation bounds $\varepsilon$. The upper part shows the difference in robust accuracy, and the lower part shows the runtime in seconds. The positive value of the difference in robust accuracy indicates the EDA's performance is higher than the compared method. For the execution time, the lower value is better.

the highest attack performance. However, regarding runtime, AA and EDA tend to decrease as epsilon increases, while $A^3$ tends to increase. EDA showed better performance than AA for all perturbation bounds. In terms of overall runtime and performance, EDA is considered to achieve a good balance between the two. Appendix G.4 provides further results as a table. To further analyze the attack performance of EDA for large perturbation bounds, we compare the robust accuracy of $A^3$ and EDA with a sufficiently large value of parameter $N$. $N$ is set so that the runtime of EDA does not exceed the runtime of $A^3$. We report the representative results corresponding to Figure 11. For the model proposed by Sehwag et al. (2022) and trained on CIFAR-10, EDA achieved 19.91% robust accuracy in 718 seconds with $N = 300$, and $A^3$ achieved 20.07% in 868 seconds. For the model proposed by Sitawarin et al. (2021) and trained on CIFAR-100, EDA showed 6.14% in 11,024 seconds with $N = 2000$, and $A^3$ showed 6.14% in 19,993 seconds. For the model proposed by Engstrom et al. (2019) and trained on ImageNet, EDA showed 7.38% robust accuracy in 6,752 seconds with $N = 1000$, and $A^3$ showed 7.44% in 11,717 seconds. These results suggest that EDA's performance may be competitive to or slightly better than $A^3$ if the computation budget in the sense of runtime is almost the same. Thus, the lower attack performance of EDA than $A^3$ in the experimental results presented in Figure 11 can be attributed to the slower convergence of EDA's search rather than its earlier convergence. Given that the runtime of $A^3$ increased as the perturbation size increased and that the attack performance of EDA with increased runtime was equal to or better than $A^3$, the number of queries required for attack convergence may increase as the perturbation size increases.

**The appropriate balance of diversification and intensification is one of the reasons for EDA's high performance.** According to Figure 12, EDA and MDO framework showed higher DI values than $A^3$. The point set with a higher value of DI is less likely to form clusters. This suggests that EDA searches a larger area than $A^3$ with almost the same or shorter runtime, resulting in EDA's high attack performance. A comparison of the MDO framework with ADS (ADS) and EDA in Figure 12 reveals that EDA exhibits a greater minimum DI than ADS. The distinction between ADS and EDA lies in the presence of MT strategy. Consequently, the results presented in Figure 12 suggest that the MT strategy may facilitate the diversification of images that are not adequately diversified by the MDO strategy. Whereas $A^3$ uses a single $\boldsymbol{\delta}$ and $L$, EDA and MDO framework use multiple $\boldsymbol{\delta}$ and $L$. Thus, this result suggests that the attack

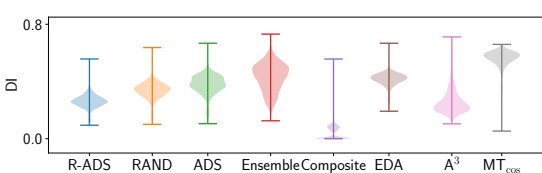

Figure 12: Violin plot of DI. The model is (Carmon et al., 2019)

Table 4: Ablation study of ADS.

| No. | MDO framework | | |
|---|---|---|---|
| | ADS | R-ADS | RAND |
| 1 | **55.58** | **55.58** | 55.61 |
| 2 | **59.46** | 59.53 | 59.56 |
| 3 | 64.32 | 64.54 | **64.23** |
| 4 | **18.97** | 18.99 | 18.98 |
| 5 | **24.65** | 24.68 | 24.71 |
| 6 | **36.96** | 37.53 | 37.05 |
| 7 | **25.22** | 25.44 | **25.22** |
| 8 | **29.20** | 29.26 | 29.56 |
| 9 | **26.22** | 26.36 | 26.26 |

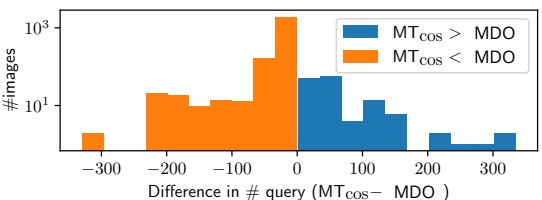

Figure 13: The difference between $MT_{cos}$ and the MDO framework in #queries to find adversarial examples. The model is (Carmon et al., 2019)

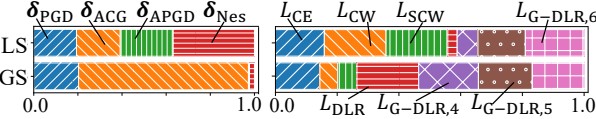

Figure 14: The ratio of $\delta$ and $L$ selected by ADS. LS and GS refer to the intensification and diversification phases, respectively.

with multiple $\delta$ and $L$ has a higher diversification ability than the attack with a single $\delta$ and $L$. Although gradient-based attacks can achieve some degree of intensification by adjusting step size, the diversification and intensification are trade-offs given the computation budget. Figure 12 shows that $MT_{cos}$ showed higher DI than EDA. However, $MT_{cos}$ showed lower attack performance than EDA, as shown in Table 1. This result suggests that $MT_{cos}$ performs an insufficient intensification, and EDA has an appropriate balance between diversification and intensification.

## 4.2 Analyses of the MDO strategy

**The combination of MDO and MT strategies would lead to an efficient attack.** Figure 13 shows the difference between the number of queries spent by $MT_{cos}$ and the MDO framework for images that both succeeded in attack. The images that $MT_{cos}$ and the MDO framework spent the same number of queries were excluded. The positive value means $MT_{cos}$ spent more queries than the MDO framework. Figure 13 indicates the presence of images that can be successfully attacked with fewer queries by $MT_{cos}$/the MDO framework but require more queries by the MDO framework/$MT_{cos}$.

**The MDO strategy mainly contributed to the attack performance of EDA.** We examined the ratio of adversarial examples generated only by the MDO framework, the targeted attack $a^T$, and both methods to entire images that EDA succeeded but $A^3$ failed. In the case of CIFAR-10, the percentages of adversarial examples generated only by the MDO framework, $a^T$, and both methods are 58.98%, 6.71%, and 34.31%, respectively. These values are averages over the 25 models trained on CIFAR-10. Similarly, the percentages are 35.02%, 8.28%, and 56.70% for CIFAR-100 and 25.90%, 15.51%, and 58.59% for ImageNet. Here, the percentages in CIFAR-100 are averages over 11 models and those in ImageNet over five. This analysis suggests that the MDO strategy contributes to the attack performance more than the MT strategy, and it can find adversarial examples that are difficult to find for existing attacks. In addition, EDA showed larger improvements in robust accuracy in half the runtime of $A^3$ for models trained on ImageNet, and the MDO framework showed promising results for the same models. These results indicate that the MDO strategy may have advantages in the attack for models trained on ImageNet, which is more practical regarding image size and the number of classification classes.

### 4.3 Analyses of ADS

**The combination of $\delta$ and $L$ selected by ADS brings a higher degree of diversification.** We compared the attack performance of the MDO framework with three selection algorithms, including ADS, Reverse ADS (R-ADS), which finds the pairs minimizing equation 4, and uniform sampling (RAND). ADS in Figure 12 and Table 4 represents the MDO framework with the selection algorithm ADS, and the same is applied to R-ADS and uniform sampling. Figure 12 shows that the DI of the best search points obtained by the MDO framework with ADS tends to be higher than that obtained by the MDO framework with R-ADS and uniform sampling, indicating ADS may select the pairs that enhance the diversification. According to Table 4, the MDO framework with ADS showed lower robust accuracy than that with R-ADS and uniform sampling. In addition, the MDO framework with R-ADS showed significantly higher robust accuracy on several models. Although the performance difference between ADS and uniform sampling is smaller than between ADS and R-ADS, ADS can select the appropriate pair more stably than uniform sampling. These results indicate that ADS selected the pairs that diversify the search more, leading to stronger attacks.

**The percentage of $\delta$ and $L$ selected by ADS.** As shown in Figure 14, during the diversification phase, ADS tended to choose DLR loss and the search direction of ACG. In the intensification phase, CW loss and directions other than ACG were more likely to be selected. Yamamura et al. (2022) reported that ACG showed superior diversification ability compared to APGD. In addition, ACG showed better attack performance with DLR loss, whereas APGD showed better performance with CW loss. These experimental results suggest that ADS selected appropriate pairs in both the diversification and intensification phases. Similar trends were observed regardless of the dataset.

**The role of $D^e$ in the intensification phase.** The correlation between $D^e$ and the highest CW loss value was checked to investigate the influence of ADS on the intensification phase. The results show that $D^e$ and the highest CW loss value are positively correlated, with correlation coefficients of 0.57 on average. As discussed in Section 4.4, attack performance is significantly degraded if ADS is not performed in the intensification phase. These experimental results suggest that ADS can select appropriate pairs for diversification and intensification. We further discuss the individual influence of the two terms, $P^e$ and DI, that appear in $D^e$ calculation on the overall ADS search performance in Appendix G.3.

**Influence of restricting candidate combinations.** To investigate the impact of restricting the candidate combinations in ADS (line 4 in Algorithm 2), we compare robust accuracy and runtime with and without this restriction. Table 5 shows that restriction on the candidate combinations does not necessarily have a positive impact on attack performance. However, in terms of runtime, this restriction provides a significant advantage. This advantage is due to the difference in the number of $D^e$ calculations. $D^e$ calculation requires the Euclidean distance between search points whose computation cost depends on the size of the images. Therefore, the runtime becomes more significant without this restriction for ImageNet, where the image size is larger.

### 4.4 Ablation study of the MDO framework

**Comparison with an ensemble and composite.** We executed $n_a$ attacks selected by ADS using the MDO framework, ensemble, and composite. The ensemble and composite executed the attacks $a_j = (\mathbf{x}_{\mathrm{PAS}}, \eta_{\cos}, \boldsymbol{\delta}_{a_j^*}, L_{a_j^*})$ for $j = 1, \ldots, n_a$. The initial step size and number of iterations of each $a_j$ were set to $2\varepsilon$ and $N = 100$, respectively. Table 6 shows that the attack performance of the MDO framework is higher than that of the ensemble and composite in most cases. Table 1 also shows that the attack performance of the MDO framework is higher than AA for several models in fewer queries. The violin plot for DI in Figure 12 shows that the degree of diversification is higher for the ensemble, the MDO framework, and the composite, in descending order. These experimental results indicate that the MDO framework is one of the effective implementations of the MDO strategy because it can achieve an appropriate balance of diversification and intensification than ensemble and composite.

Table 5: Influence of restricting candidate combinations.

| No. | w/o restriction | | w/ restriction | | Δ |
|---|---|---|---|---|---|
| | acc | sec | acc | sec | (sec) |
| 1 | 55.62 | 672 | 55.58 | 386 | 285 |
| 2 | 59.46 | 2,308 | 59.46 | 2,112 | 195 |
| 3 | 64.23 | 15,300 | 64.32 | 15,187 | 113 |
| 4 | 18.98 | 487 | 18.97 | 299 | 187 |
| 5 | 24.61 | 1,420 | 24.65 | 1,186 | 233 |
| 6 | 36.92 | 9,229 | 36.96 | 9,175 | 53 |
| 7 | 25.20 | 6,190 | 25.22 | 1,430 | 4,759 |
| 8 | 29.28 | 6,492 | 29.20 | 1,973 | 4,518 |
| 9 | 26.40 | 7,765 | 26.22 | 2,930 | 4,834 |

Table 6: Ablation study of the MDO framework.

| No. | Ensemble | Composite | MDO framework | MDO framework w/o ADS in the intensification phase |
|---|---|---|---|---|
| 1 | 55.68 | 55.90 | **55.58** | 62.80 |
| 2 | 59.70 | 59.59 | **59.46** | 69.25 |
| 3 | 64.53 | 64.43 | **64.32** | 70.71 |
| 4 | 19.08 | 19.12 | **18.97** | 24.40 |
| 5 | 24.74 | 24.83 | **24.65** | 32.51 |
| 6 | 37.19 | 37.17 | **36.96** | 43.87 |
| 7 | 25.46 | 25.48 | **25.22** | 29.76 |
| 8 | 29.64 | 29.68 | **29.20** | 36.74 |
| 9 | 26.84 | 26.78 | **26.22** | 30.66 |

Table 7: Influence of further intensification (lines 21-23 of Algorithm 3).

| | w/o lines 21-23 | | w/o lines 22-23 | | w/ lines 21-23 | |
|---|---|---|---|---|---|---|
| Attack on subset (line 21) | - | | ✓ | | ✓ | |
| Extra attack iteration (lines 22-23) | - | | - | | ✓ | |
| No. | acc | sec | acc | sec | acc | sec |
| 1 | 55.58 | 574 | 55.63 | 340 | 55.58 | 387 |
| 2 | 59.46 | 3,675 | 59.45 | 1,914 | 59.46 | 2,112 |
| 3 | 64.30 | 26,709 | 64.30 | 13,993 | 64.32 | 15,187 |
| 4 | 18.94 | 380 | 19.00 | 262 | 18.97 | 299 |
| 5 | 24.66 | 1,749 | 24.67 | 1,078 | 24.65 | 1,187 |
| 6 | 36.93 | 14,385 | 36.93 | 8,335 | 36.96 | 9,175 |
| 7 | 25.18 | 1,376 | 25.18 | 1,182 | 25.22 | 1,431 |
| 8 | 29.16 | 2,688 | 29.36 | 1,929 | 29.20 | 1,974 |
| 9 | 26.36 | 3,795 | 26.42 | 2,838 | 26.22 | 2,931 |

**Influence of ADS in the intensification phase.** Table 6 shows the robust accuracy of the MDO framework with and without ADS execution prior to the intensification phase. The results show that the attack performance significantly drops without ADS prior to the intensification phase. This suggests that the pairs suitable for diversification and intensification differ from each other. The EDA's performance may be further improved by considering the execution order of the attacks with selected $\delta$ and $L$. However, determining the execution order requires that $(n_a!)^2$ permutations be considered, which is computationally expensive. The experimental results suggest that EDA achieves a satisfactory trade-off between the computation cost and the attack performance.

**Influence of further intensification (lines 21-23 of Algorithm 3).** The intensification phase of the MDO framework is conducted on a subset of test samples to reduce the computational cost. This section validates the efficacy of this heuristic for determining the subset of test samples and further intensification on this subset (lines 21-23 of Algorithm 3). Table 7 indicates that limiting the attack target images during the intensification phase results in a notable reduction in computational cost compared to a scenario where no such limitations are imposed. Nevertheless, simply limiting the target images may reduce attack performance by up to 0.2%. According to the comparison between "w/o lines 21-23" and "w/ lines 21-23", the performance degradation is up to 0.04% when more attack queries are allocated to the subset of test samples. The required computational overhead is not particularly significant compared to "w/o lines 22-23". Furthermore, the attack performance of "w/ lines 21-23" is the highest for some models.

### 4.5 Additional results and ablations

Appendix D.5 describes the analysis of EDA for the model of Ding et al. (2020), which showed different trends. The analysis using the Euclid distance-based measure in Appendix D.4 showed similar trends to DI-based analysis. Appendix G shows the result of the ablation studies, including the hyperparameter sensitivity of EDA and performance evaluation with different perturbation bounds. EDA is expected to be robust to hyperparameter settings to some extent. As described in Appendix H, EDA also showed better performance than AA and $A^3$ for randomized defenses.

## 5 Limitations and assumptions

This study has the following assumptions and limitations. EDA assumes to be able to access the outputs and gradients of the attacked models. Similar to the Adaptive Auto Attack ($A^3$), EDA assumes to attack the set of images. These assumptions are reasonable for robustness evaluation and adversarial training. The number of combinations of search directions and objective functions grows exponentially. Thus, the number of search directions and objective functions to be selected should be manageable. Unlike $A^3$, AA and EDA do not alter the number of attack queries during the attack process. Although we empirically find that EDA is robust to hyperparameter settings, users should specify enough number of attack queries, especially for the different perturbation bounds.

**Expected situations in EDA work well** This study experimentally verifies the performance of EDA, focusing on image classifiers using deterministic defenses such as adversarial training. Although EDA was tested with images of different resolutions, domains, and models of different architectures, EDA worked in most settings. Therefore, it can be assumed that EDA has some generalization performance for image classifiers using deterministic defenses. Furthermore, EDA showed lower robust accuracy in less runtime than AA and $A^3$ on several randomized defenses. Similar to AA, EDA is applicable to the models whose gradients are available. However, the performance of EDA against models out of image classifiers needs further investigation.

**Expected situations in EDA do not work well** EDA contains gradient-based attacks as some components. Therefore, EDA's performance may be degraded for models which cause incorrect gradient calculations. As reported by Croce & Hein (2020b), existing techniques like expectation over transformation might help improve the attack performance on these models. Also, the performance of EDA may be degraded when the MDO and MT strategies do not work well.

## 6 Conclusion

This study empirically confirmed that different local solutions can be efficiently enumerated using various search directions and objective functions. Based on this observation, we have proposed the MDO strategy and its implementation, including ADS and the MDO framework. The experiments on robust models, including 41 CNN-based and 10 more advanced architectures, have demonstrated that the MDO strategy realized by the MDO framework has a higher diversification ability. In addition, EDA, a combination of MDO and MT strategy, showed higher attack performance than recently proposed attacks. Though more appropriate indices may exist, these results suggest that the attack designed based on the DI shows an appropriate balance of diversification and intensification, resulting in a stronger attack. We are interested in exploring simpler algorithms as future work that perform as well or better than the proposed method.

The experimental results in Section 3.1 provide future research directions for further understanding of optimization-based adversarial attacks or adversarial training. The first direction is a systematic classification of the optimization-based attacks. Although many optimization-based attacks have been proposed, the appropriate method differs depending on the purpose, including training, white-box evaluation, and transfer-based attacks. Systematic classification may help in selecting appropriate methods. The second direction is adversarial training using multiple attacks. The experimental results suggest that different attacks

find different clusters of adversarial examples. This slight difference may affect the stability or efficiency of adversarial training. Adversarial training using EDA is also an interesting topic.

**Broader Impact Statement**

Deep neural networks (DNNs) are known to be vulnerable to adversarial examples. A promising defense mechanism to address this vulnerability is adversarial training, where training is performed using adversarial examples. Many adversarial examples generated by strong attack methods are required to produce robust models through adversarial training. Therefore, developing strong and fast adversarial attacks helps improve the robustness of DNNs. The EDA, which is one of the main proposals of this research, can generate a larger number of adversarial examples in a shorter time and can be used for both the robustness evaluation of defense methods and data generation in adversarial training. Thus, this research significantly contributes to the security of DNNs. The positive impact of this research is twofold. First, we can make DNNs more robust through adversarial training using the data generated by the strong attack method EDA. Second, we can more accurately evaluate the robustness of the models. The potential negative impact of this research includes possible attacks by malicious users on systems containing DNNs. However, EDA is a white-box attack that assumes the accessibility of model gradients. In addition, it is difficult to access the model gradients involved in a real system. EDA is thus unsuitable for attacking a real system. As described above, research benefits are more significant than the potential negative effects. This study helps to improve the robustness of DNNs, allowing them to be more safely applied to a broader range of applications, including safety-critical applications.

**Acknowledgments**

This research project was supported by the Japan Science and Technology Agency (JST), the Core Research of Evolutionary Science and Technology (CREST), the Center of Innovation Science and Technology based Radical Innovation and Entrepreneurship Program (COI Program), JSPS KAKENHI Grant Number JP16H01707 and JP21H04599, Japan.

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

## Appendix

This supplementary material provides additional information as follows.

1. The summary of abbreviations and mathematical notations defined in the main text (Tables 8 and 9).

2. More information about related work (Appendix A).

3. The proposed search direction and objective function (Appendix B).

4. Additional results of the analysis in Section 3.1 (Appendix C).

5. Complete results of the experiments in Section 4 (Appendix D).

6. The details of Prediction Aware Sampling (Appendix E).

7. The details of the targeted attack used in EDA (Appendix F).

8. Ablation study of EDA (Appendix G).

9. Experiments on randomized defenses (Appendix H).

Table 8: Summary of abbreviations

| | Existing attack techniques |
|---|---|
| PGD | Projected Gradient Descent |
| MT-PGD | MultiTargeted-PGD |
| APGD | Auto-PGD |
| ACG | Auto Conjugate Gradient attack |
| AA | AutoAttack |
| CAA | Composite Adversarial Attack |
| $A^3$ | Adaptive Auto Attack |
| CE | Cross-entropy |
| DLR | Difference of Logit Ratio |
| ODS | Output Diversified Sampling |
| DI | Diversity Index |
| | Proposed methods |
| G-DLR | Generalized-DLR |
| NAG | Nesterov's accelerated gradient |
| PAS | Prediction Aware Sampling |
| MDO | Multi-directions/objectives |
| ADS | Automated Diversified Sampling |
| MDO framework | Search framework for the MDO strategy |
| EDA | Efficient Diversified Attack |
| | Others |
| DNN | Deep neural network |
| SOTA | State-of-the-art |
| MT | Multi-target |
| UMAP | Uniform Manifold Approximation and Projection |

# A More information about related work

## A.1 Search directions

**Projected Gradient Descent**   Projected Gradient Descent (PGD) (Madry et al., 2018) is the most fundamental adversarial attack based on the steepest gradient descent. The search direction of PGD is computed as follows:

$$\boldsymbol{\delta}_{\text{PGD}}^{(k)} = \text{sign}\left(\nabla L(g(\boldsymbol{x}^{(k)}), c)\right) \tag{5}$$

Also, Fast Gradient Sign Method (FGSM) (Goodfellow et al., 2015) updates towards the same direction.

**Auto-PGD**   Auto-PGD (APGD) (Croce & Hein, 2020b) is a PGD variant that adjusts the step size and updates towards momentum direction in addition to the PGD's search direction. The search direction of APGD is defined as follows.

$$\boldsymbol{z}^{(k)} = P_{\mathcal{S}}\left(\boldsymbol{x}^{(k)} + \eta^{(k)}\boldsymbol{\delta}_{\text{PGD}}^{(k)}\right), \tag{6}$$

$$\boldsymbol{\delta}_{\text{APGD}}^{(k)} = \alpha(\boldsymbol{z}^{(k)} - \boldsymbol{x}^{(k)}) + (1 - \alpha)(\boldsymbol{x}^{(k)} - \boldsymbol{x}^{(k-1)}), \tag{7}$$

where $\alpha$ is a coefficient of momentum term. APGD uses $\alpha = 1$ for the first iteration and $\alpha = 0.75$ for the remaining iterations. The step size $\eta^{(k)}$ is halved if the following conditions are satisfied at the $w_j \in W$ iteration, with the initial value $\eta^{(0)} = 2\varepsilon$.

Table 9: Summary of mathematical notations

| Defined in Section 2.1 | |
| --- | --- |
| $g : D \to \mathbb{R}^C$ | Locally differentiable $C$-classifier. |
| $\boldsymbol{x}_{\mathrm{org}} \in D$ | Original input. |
| $c \in Y = \{1, 2, \ldots, C\}$ | The correct classification label of $\boldsymbol{x}_{\mathrm{org}}$. |
| $d : D \times D \to \mathbb{R}$ | A distance function. |
| $\mathcal{S}$ | The feasible region. |
| $L : \mathbb{R}^C \times Y \to \mathbb{R}$ | Objective function (untargeted attack). |
| $L^T : \mathbb{R}^C \times Y \times Y \to \mathbb{R}$ | Objective function (targeted attack). |

| Defined in Section 2.2 | |
| --- | --- |
| $P_{\mathcal{S}} : D \to \mathcal{S}$ | Projection onto the feasible region $\mathcal{S}$. |
| $\mathtt{x}$ | Initial point sampling. |
| $\eta$ | Step size update rule. |
| $\boldsymbol{\delta}$ | Search direction/Update formula. |
| $\boldsymbol{x}^{(k)} \in D$ | Search point at iteration $k$. |
| $\eta^{(k)} \in \mathbb{R}$ | Step size at iteration $k$. |
| $\boldsymbol{\delta}^{(k)} \in \mathbb{R}^n$ | Search direction at iteration $k$. |
| $a = (\mathtt{x}, \eta, \boldsymbol{\delta}, L)$ | An attack. |
| $X^a$ | A set of best search points. |
| $X_i^a$ | A set of best search points corresponds to image $\boldsymbol{x}_i$. |
| $\Pi^a$ | A set of classification labels with the highest probabilities except for the correct classification class. |

| Defined in Section 3.1 | |
| --- | --- |
| $\pi_q \in Y$ | The class label that has the $q$-th largest value of $g(\boldsymbol{x})$. |
| $\mathcal{D}$ | A set of update formulas. |
| $\mathcal{L}$ | A set of objective functions. |
| $\mathcal{A}(\mathtt{x}, \eta)$ | A set of attacks with initial point sampling $\mathtt{x}$ and step size update rule $\eta$. |
| $DI(X) = DI(X, M)$ | Diversity Index of a point set $X$. $M$ is the size of feasible region. |

| Defined in Section 3.2 | |
| --- | --- |
| $e = \{a_j\}_{j=1}^{n_a}$ | A combination of attacks. |

1. $\displaystyle\sum_{i=w_{j-1}}^{w_j - 1} \mathbf{1}_{L\left(g(\boldsymbol{x}^{(i+1)}), c\right) > L\left(g(\boldsymbol{x}^{(i)}), c\right)} < \rho \cdot (w_j - w_{j-1})$

2. $L_{\max}\left(g(\boldsymbol{x}^{(w_{j-1})}), c\right) = L_{\max}\left(g(\boldsymbol{x}^{(w_j)}), c\right)$ and $\eta^{(w_{j-1})} = \eta^{(w_j)}$

The sequence of checkpoints $W$ is computed based on the following gradual equation depending on the total number of iterations $N_{iter}$. $p_0 = 0, p_1 = 0.22, p_{j+1} = p_j + \max\{p_j - p_{j-1} - 0.03, 0.06\}, w_j = \lceil p_j N_{iter} \rceil$. In our notation, $\eta_{\mathrm{APGD}}$ denotes this step size updating rule.

**Auto-Conjugate Gradient attack** Auto-Conjugate Gradient (ACG) attack (Yamamura et al., 2022) is inspired by the Conjugate Gradient method for nonlinear optimization problems. ACG performs a more

diversified search than the attacks based on the steepest descent. The search direction of ACG is as follows.

$$\boldsymbol{y}^{(k)} = \nabla L(g(\boldsymbol{x}^{(k-1)}), c) - \nabla L(g(\boldsymbol{x}^{(k)}), c) \tag{8}$$

$$\beta^{(k)} = -\frac{\nabla L(\boldsymbol{x}^{(k)}, c)^T \boldsymbol{y}^{(k)}}{(\boldsymbol{y}^{(k)})^T \boldsymbol{\delta}_{\text{ACG}}^{(k-1)}} \tag{9}$$

$$\boldsymbol{\delta}_{\text{ACG}}^{(k)} = \nabla L(\boldsymbol{x}^{(k)}, c) + \beta^{(k)} \boldsymbol{\delta}_{\text{ACG}}^{(k-1)} \tag{10}$$

## A.2 Objective functions

**Cross-entropy loss** The untargeted version of cross-entropy (CE) loss is defined as follows.

$$L_{\text{CE}}(g(\boldsymbol{x}), c) = -g_c(\boldsymbol{x}) + \log\left(\sum_{j \neq c} \exp\left(g_j(\boldsymbol{x})\right)\right) \tag{11}$$

Also, the targeted version of CE loss is defined as follows.

$$L_{\text{CE}}^T(g(\boldsymbol{x}), c, t) = g_t(\boldsymbol{x}) - \log\left(\sum_{j \neq t} \exp\left(g_j(\boldsymbol{x})\right)\right), \tag{12}$$

where $t$ denotes the target label of misclassification. CE loss is known to be sensitive to the scaling of the logit, i.e., the attack performance significantly varies depending on the scaling of the logit (Carlini & Wagner, 2017; Croce & Hein, 2020b).

**CW loss** The untargeted version of CW loss is defined as follows.

$$L_{\text{CW}}(g(\boldsymbol{x}), c) = \max_{j \neq c} g_j(\boldsymbol{x}) - g_c(\boldsymbol{x}) \tag{13}$$

Also, the targeted version of CW loss is defined as follows.

$$L_{\text{CW}}^T(g(\boldsymbol{x}), c, t) = g_t(\boldsymbol{x}) - g_c(\boldsymbol{x}), \tag{14}$$

where $t$ denotes the target label of misclassification.

**Difference of Logit Ratio loss** The untargeted version of the Difference of Logit Ratio (DLR) loss is defined as follows.

$$L_{\text{DLR}}(g(\boldsymbol{x}), c) = \frac{\max_{j \neq c} g_j(\boldsymbol{x}) - g_c(\boldsymbol{x})}{g_{\pi_1} - g_{\pi_3}}, \tag{15}$$

where $\pi_q$ denotes the classification label with $q$-th highest value in $g(\boldsymbol{x})$. Also, the targeted version of DLR loss is defined as follows.

$$L_{\text{DLR}}^T(g(\boldsymbol{x}), c, t) = \frac{g_t(\boldsymbol{x}) - g_c(\boldsymbol{x})}{g_{\pi_1} - (g_{\pi_3}(\boldsymbol{x}) + g_{\pi_4}(\boldsymbol{x}))/2}, \tag{16}$$

where $t$ denotes the target label of misclassification.

## A.3 Comparison of existing attacks and EDA

Table 10 summarizes the characteristics of PGD-like attacks, Auto Attack (AA) (Croce & Hein, 2020b), Composite Adversarial Attack (CAA) Mao et al. (2021), AutoAE Liu et al. (2023b), Adaptive Auto Attack ($A^3$) (Liu et al., 2022c), and EDA in terms of diversification, intensification, and computational cost. The attacks in Table 10 perform well in intensification because they include gradient-based attacks with appropriate step size management. Although PGD-like attacks have several variations, this section describes the representative one in Table 10. PGD-like attacks and $A^3$ use multi-restart for diversification, and both attacks spend a relatively short computational time. However, $A^3$ outperforms PGD-like attacks because $A^3$ uses better initial point sampling. AA considers multiple objective functions and multi-target attacks

Table 10: Characteristics of PGD-like attacks, AA, A$^3$, and EDA. multi-$L$ denotes the diversification using multiple objective functions, and multi-$\boldsymbol{\delta}$ denotes the diversification using multiple search directions.

| Attacks | Diversification | | | | Intensification | Runtime |
|---|---|---|---|---|---|---|
| | multi-$L$ | multi-$\boldsymbol{\delta}$ | multi-restart | multi-target | | |
| PGD-like | - | - | ✓ | - | ✓ | short |
| AA | ✓ | - | ✓ | ✓ | ✓ | long |
| CAA | ✓ | - | ✓ | - | ✓ | long |
| AutoAE | ✓ | - | ✓ | - | ✓ | long |
| A$^3$ | - | - | ✓ | - | ✓ | short |
| EDA | ✓ | ✓ | ✓ | ✓ | ✓ | short |

for diversification in addition to multi-restart. Although AA achieves a high attack success rate, AA is computationally expensive because AA consists of four attacks, including APGD with untargeted CE loss, APGD with targeted DLR loss, FAB attack (Croce & Hein, 2020a), and square attack (Andriushchenko et al., 2020). CAA and AutoAE also combine multiple attacks by solving an additional optimization problem. CAA executes MultiTargeted (MT) attack Gowal et al. (2019) and CW attack Carlini & Wagner (2017) as a composite of attacks. AutoAE runs APGD with CE loss, APGD with DLR loss, FAB, and MT attack as an ensemble. In the sense that AA, CAA, and AutoAE use different types of attacks, we can consider that they employ the diversification strategy based on multi-$\boldsymbol{\delta}$. However, these methods use only a single search direction to solve (2) in the white-box setting. Therefore, we do not consider AA an attack with a diversification strategy based on multi-$\boldsymbol{\delta}$. In contrast, EDA, the proposed attack, uses all diversification strategies listed in Table 10 and achieves a higher attack success rate in a short computation time.

## B The proposed search direction and objective function

### B.1 Search direction inspired by Nesterov's Accelerated Gradient

Although some attacks were inspired by Nesterov's accelerated gradient (Nesterov, 2004), most of them apply constant value to the coefficient of momentum (Lin et al., 2020; Liu et al., 2022a). However, the original Nesterov's accelerated gradient method determines the coefficient of momentum term by solving the quadratic equations. So then we try to adopt Nesterov's accelerated gradient to $\ell_\infty$ attacks. Mathematically, $\boldsymbol{\delta}_{\text{Nes}}$ is computed by the following equations.

$$\rho^{(k)} \text{ is a positive solution of } (\rho^{(k)})^2 = (1 - \rho^{(k)})(\rho^{(k-1)})^2 \tag{17}$$

$$\gamma^{(k)} \leftarrow \frac{\rho^{(k-1)}\left(\rho^{(k-1)} - 1\right)}{\rho^{(k)} + (\rho^{(k-1)})^2} \tag{18}$$

$$\tilde{\boldsymbol{x}}^{(k)} \leftarrow \boldsymbol{x}^{(k)} + \gamma^{(k)}\left(\boldsymbol{x}^{(k)} - \boldsymbol{x}^{(k-1)}\right) \tag{19}$$

$$\boldsymbol{\delta}_{\text{Nes}}^{(k+1)} \leftarrow \text{sign}\left(\nabla L(g(\tilde{\boldsymbol{x}}^{(k)}), c)\right) \tag{20}$$

### B.2 Generalized-DLR loss

We generalize DLR loss by extending the denominator of DLR loss from $g_{\pi_1}(\boldsymbol{x}) - g_{\pi_3}(\boldsymbol{x})$ to $g_{\pi_1}(\boldsymbol{x}) - g_{\pi_q}(\boldsymbol{x})$. $\pi_q \in Y$ denotes the class label that has the $q$-th largest value of $g(\boldsymbol{x})$. More precisely, $L_{\text{G-DLR},q}$ is defined as

$$L_{\text{G-DLR},q}(g(\boldsymbol{x}), c) = -\frac{g_c(\boldsymbol{x}) - g_{\pi_2}(\boldsymbol{x})}{g_{\pi_1}(\boldsymbol{x}) - g_{\pi_q}(\boldsymbol{x})}. \tag{21}$$

The motivation for proposing generalized-DLR loss is to increase diversity in the output space. For the search points with larger values of G-DLR loss, the denominator/numerator will be smaller/larger value, respectively. The small value of denominator $g_{\pi_1}(\boldsymbol{x}) - g_{\pi_q}(\boldsymbol{x})$ means that the values of $g_{\pi_1}(\boldsymbol{x}), g_{\pi_2}(\boldsymbol{x}), \ldots g_{\pi_q}(\boldsymbol{x})$ are

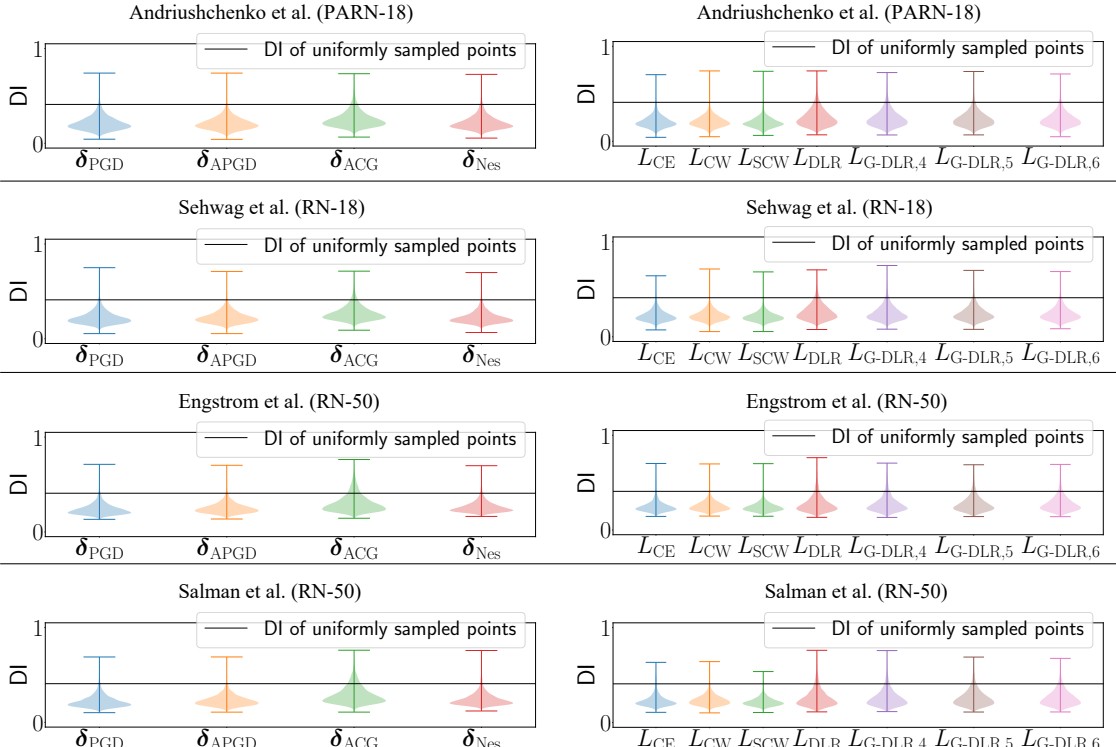

Figure 15: Violin plot of DI obtained by attacking the models proposed by (Sehwag et al., 2022; Andriushchenko & Flammarion, 2020) for CIFAR-10 and (Engstrom et al., 2019; Salman et al., 2020) for ImageNet.

close. Therefore, the large and small relationships between the predicted probability of classification classes are expected to be easier to variate than in the case of DLR loss.

## C   Additional results of the analysis in section 3.1

### C.1   Diversity index for the set of best search points

The main paper only includes the results for the model proposed by Sehwag et al. (2022). This section describes the violin plot of DI for several models (Andriushchenko & Flammarion, 2020; Sehwag et al., 2022; Engstrom et al., 2019; Salman et al., 2020). Figure 15 shows the violin plot of DI obtained by attacking the robust models. According to Figure 15, the best point sets obtained by attacks with a single search direction and objective function have similar DI value trends.

### C.2   Visualization of the best search points via UMAP

This section describes the 2D visualization of the best search points using UMAP for several models (Andriushchenko & Flammarion, 2020; Sehwag et al., 2022; Engstrom et al., 2019; Salman et al., 2020). Figure 16 shows the 2D visualization of the best search points obtained by attacking the robust models trained on CIFAR-10. Figure 17 shows the 2D visualization of the best search points obtained by attacking the robust models trained on ImageNet. According to Figures 16 and 17, the best point sets obtained by attacks with different search directions and objective functions tend to form different clusters. The points determined to belong to the same cluster due to clustering using the X-means (Pelleg & Moore, 2000) are also plotted close together in the visualization using UMAP. These results suggest that 2D visualizations using UMAP are expected to reflect the actual distribution of search points.

Andriushchenko et al. (PARN-18)

Sehwag et al. (RN-18)

Figure 16: 2D visualization of the best search points obtained by attacking the models proposed by (Sehwag et al., 2022; Andriushchenko & Flammarion, 2020). The dataset is CIFAR-10. The same color in the left/center figure represents points obtained using the same objective function/search direction, respectively. The same color in the right figure shows the points determined by X-means to belong to the same cluster.

### C.3 The reason why we used UMAP

The objective of the qualitative evaluation using UMAP is to know how the best points obtained by attacks using different objective functions/search directions are distributed and form different clusters. In order to achieve this goal, it is necessary to consider the distance between any two points and the distance between clusters. We have tried quantitative evaluation. However, we finally chose qualitative evaluation using UMAP because quantitative evaluations based on indicators such as objective values or DI are difficult to achieve our objective for the reasons described below. First, in adversarial attacks, distant points may show the same objective value or close points may show very different objective values because the adversarial attack is a maximization problem with many local optimums. Therefore, quantitative evaluation using objective values is considered difficult. In addition, DI cannot consider the distance between clusters because DI shows low values when a point set forms one or more clusters. Another possible evaluation method is clustering, such as k-means, but this is a qualitative evaluation as with UMAP. UMAP is a dimensionality reduction method that preserves the distance information in the original space as much as possible so it can reflect important information, such as the distance between any two points or clusters. Therefore, we think that the qualitative evaluation by UMAP provides convincing results.

## D Complete results of the experiments

Table 11 shows the complete results of the experiments in Section 4.1, as described in Table 1. Table 12 shows the runtime of CAA, AutoAE, $A^3$, and EDA. Table 13 is the complete results of the experiments in Sections 4.3 and 4.4, as described in Table 4. To investigate the stability of the proposed methods, we report the mean and standard deviation from five runs with different random seeds against 41 CNN-based models in Section 4.1 and Appendix G.2. The results in Section 4.1 and Appendix G.2 suggest a stable performance of the proposed methods. Owing to the computation cost, the remaining experiments were conducted with a single fixed random seed. Table 14 shows the quantified degree of diversification of $A^3$, the MDO framework with ADS, the MDO framework with R-ADS, the MDO framework with uniform sampling, and EDA. Table 15 provides the complete results for the transferability evaluation of EDA described in Table 3. Figure 18 shows the difference between $MT_{cos}$ and the MDO framework in #queries to find adversarial examples for some models. Figure 19 shows the violin plot of DI for several models.

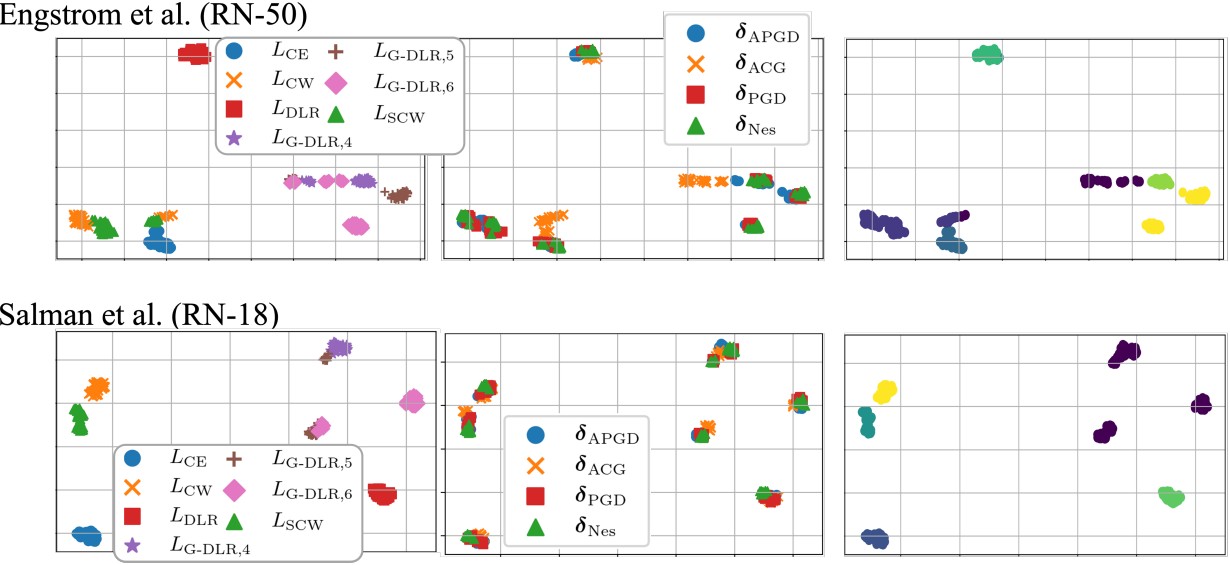

Figure 17: 2D visualization of the best search points obtained by attacking the models proposed by (Engstrom et al., 2019; Salman et al., 2020). The dataset is ImageNet. The same color in the left/center figure represents points obtained using the same objective function/search direction, respectively. The same color in the right figure shows the points determined by X-means to belong to the same cluster.

## D.1 Hyperparameter determination

The values of $n_a, N_{\text{ADS}}$, and $N_s$ are determined based on the following preliminary experiments, as they should be as small as possible regarding computational cost. The initial step size is determined based on the step size rules of APGD, a powerful heuristic. The step size of $\eta = 2\varepsilon$ allows the initial search to move from one end of the feasible region to the other, thus allowing a broader search. The parameters 0.22 and 0.41, which determine the allocation of the number of iterations for the diversification phase and the intensification phase, are inspired by the checkpoints in the APGD's step size update. $N_1$ is the number of iterations to be searched with a step size of $2\varepsilon$ to $\varepsilon$ in APGD. The experiments in Yamamura et al. (2022) suggest that the CG diversification performance is well achieved by moving in the CG direction according to this iteration allocation and step size assignment. Therefore, we chose these values for step sizes, $N_1$, and $N_2$.

**Preliminary experiments to determine hyperparameters of ADS** We conducted preliminary experiments on the following five models to determine the hyperparameters of ADS. 1. ResNet-18 (Sehwag et al., 2022), 2. WideResNet-28-10 (Gowal et al., 2020), 3. PreActResNet-18 (Rice et al., 2020), 4. WideResNet-34-10 (Sitawarin et al., 2021), 5. ResNet-50 (Engstrom et al., 2019). These numbers correspond to the "No." column in Table 16.

## D.2 Comparison in computation cost based on the number of queries

We compare the number of queries because the bottleneck in an adversarial attack is forward/backward (queries). In attack selection, CAA requires $KNt\times\#$ samples $= 60t\times$ #samples $\geq 60\times$ #samples, where $K = 20$ is the population size, $N = 3$ is the policy length, $t \geq 1$ is the number of iterations for the candidate attacks, and #samples $= 4,000$ for CIFAR-10 and 1000 for ImageNet. In summary, attack selection in CAA requires more than 240,000 queries for CIFAR-10 and 60,000 for ImageNet. For attack selection by ADS, $2|A|N_{ADS}\times$ #samples $= 112\times$ #samples queries are required, where $|A| = 28$ is the number of candidates, $N_{ADS} = 4$ is the number of iterations for candidates, and #samples$=100$ for CIFAR-10/100 and 50 for ImageNet. ADS thus requires 11,200 queries for CIFAR-10 and 5,600 queries for ImageNet.

Table 11: Average robust accuracy over five runs. The lowest accuracies are in bold. RN: ResNet, WRN: WideResNet, PARN: PreActResNet, $\Delta$: $A^3$-EDA. The "EDA/$A^3$" column is the same as the "ratio" column in Table 1.

| CIFAR-10 ($\varepsilon = 8/255$) | model | clean | AA | ACG | $MT_{cos}$ | CAA | AutoAE | $A^3$ | EDA |
|---|---|---|---|---|---|---|---|---|---|
| Andriushchenko & Flammarion (2020) | PARN-18 | 79.84 | 43.93 | 45.16 | 43.96 | 43.93 | 43.93 | 43.96±0.00 | 43.85±0.02 |
| Addepalli et al. (2022) | RN-18 | 85.71 | 52.48 | 52.87 | 52.48 | 52.48 | 52.46 | 52.46±0.02 | 52.43±0.03 |
| Sehwag et al. (2022) | RN-18 | 84.59 | 55.54 | 56.19 | 55.54 | 55.53 | 55.51 | 55.53±0.01 | 55.49±0.01 |
| Engstrom et al. (2019) | RN-50 | 87.03 | 49.25 | 50.88 | 49.21 | 49.24 | 49.21 | 49.25±0.02 | 49.10±0.03 |
| Carmon et al. (2019) | WRN-28-10 | 89.69 | 59.53 | 60.10 | 59.54 | 59.54 | 59.46 | 59.44±0.01 | 59.40±0.01 |
| Gowal et al. (2020) | WRN-28-10 | 89.48 | 62.80 | 63.18 | 62.86 | 62.78 | 62.80 | 62.77±0.01 | 62.75±0.02 |
| Hendrycks et al. (2019) | WRN-28-10 | 87.11 | 54.92 | 55.71 | 54.90 | 54.88 | 54.81 | 54.85±0.01 | 54.77±0.02 |
| Rebuffi et al. (2021) | WRN-28-10 | 87.33 | 60.75 | 61.23 | 60.80 | 60.74 | 60.71 | 60.72±0.01 | 60.64±0.01 |
| Sehwag et al. (2020) | WRN-28-10 | 88.98 | 57.14 | 57.68 | 57.18 | 57.16 | 57.10 | 57.14±0.02 | 57.03±0.01 |
| Sridhar et al. (2022) | WRN-28-10 | 89.46 | 59.66 | 60.22 | 59.67 | 59.60 | 59.60 | 59.56±0.01 | 59.46±0.02 |
| Wang et al. (2020) | WRN-28-10 | 87.50 | 56.29 | 57.45 | 56.38 | 56.29 | 56.23 | 56.28±0.01 | 56.15±0.02 |
| Wu et al. (2020) | WRN-28-10 | 88.25 | 60.04 | 60.35 | 60.06 | 60.00 | 59.97 | 60.02±0.01 | 59.94±0.01 |
| Ding et al. (2020) | WRN-28-4 | 84.36 | 41.44 | 44.36 | 41.99 | 41.67 | 41.48 | 41.24±0.06 | 41.74±0.06 |
| Addepalli et al. (2022) | WRN-34-10 | 88.71 | 57.81 | 58.43 | 57.77 | 57.77 | 57.74 | 57.73±0.01 | 57.69±0.02 |
| Sehwag et al. (2022) | WRN-34-10 | 86.68 | 60.27 | 61.09 | 60.31 | 60.26 | 60.26 | 60.22±0.01 | 60.18±0.02 |
| Sitawarin et al. (2021) | WRN-34-10 | 86.84 | 50.72 | 51.94 | 50.75 | 50.70 | 50.64 | 50.69±0.02 | 50.59±0.01 |
| Zhang et al. (2019) | WRN-34-10 | 87.20 | 44.83 | 45.76 | 44.69 | 44.65 | 44.57 | 44.63±0.03 | 44.51±0.02 |
| Zhang et al. (2020) | WRN-34-10 | 84.52 | 53.51 | 54.00 | 53.52 | 53.48 | 53.48 | 53.46±0.01 | 53.42±0.02 |
| Sridhar et al. (2022) | WRN-34-15 | 86.53 | 60.41 | 60.93 | 60.43 | 60.39 | 60.35 | 60.38±0.01 | 60.32±0.01 |
| Gowal et al. (2020) | WRN-34-20 | 85.64 | 56.86 | 57.15 | 56.83 | 56.82 | 56.85 | 56.81±0.01 | 56.79±0.03 |
| Pang et al. (2020) | WRN-34-20 | 85.14 | 53.74 | 54.66 | 53.71 | 53.72 | 53.67 | 53.69±0.01 | 53.66±0.01 |
| Rice et al. (2020) | WRN-34-20 | 85.34 | 53.42 | 54.33 | 53.39 | 53.36 | 53.35 | 53.38±0.01 | 53.34±0.01 |
| Gowal et al. (2020) | WRN-70-16 | 85.29 | 57.20 | 57.60 | 57.15 | 57.16 | 57.19 | 57.11±0.01 | 57.12±0.01 |
| Gowal et al. (2020) | WRN-70-16 | 91.10 | 65.88 | 66.29 | 65.96 | 65.87 | 65.80 | 65.85±0.01 | 65.83±0.01 |
| Rebuffi et al. (2021) | WRN-70-16 | 88.54 | 64.25 | 64.80 | 64.28 | 64.23 | 64.19 | 64.24±0.01 | 64.20±0.03 |
| CIFAR-100 ($\varepsilon = 8/255$) | model | clean | AA | ACG | $MT_{cos}$ | CAA | AutoAE | $A^3$ | EDA |
| Rice et al. (2020) | PARN-18 | 53.83 | 18.95 | 19.48 | 18.99 | 18.97 | 18.91 | 18.89±0.00 | 18.88±0.01 |
| Hendrycks et al. (2019) | WRN-28-10 | 59.23 | 28.42 | 29.51 | 28.43 | 28.44 | 28.35 | 28.32±0.02 | 28.27±0.02 |
| Rebuffi et al. (2021) | WRN-28-10 | 62.41 | 32.06 | 32.73 | 32.07 | 32.05 | 32.02 | 32.00±0.02 | 31.94±0.03 |
| Addepalli et al. (2022) | WRN-34-10 | 68.75 | 31.85 | 32.38 | 31.80 | 31.80 | 31.83 | 31.81±0.02 | 31.78±0.01 |
| Cui et al. (2021) | WRN-34-10 | 60.64 | 29.33 | 29.65 | 28.99 | 28.99 | 28.86 | 28.84±0.02 | 28.83±0.02 |
| Sitawarin et al. (2021) | WRN-34-10 | 62.82 | 24.57 | 25.69 | 24.55 | 24.57 | 24.52 | 24.56±0.03 | 24.50±0.01 |
| Wu et al. (2020) | WRN-34-10 | 60.38 | 28.86 | 29.90 | 28.86 | 28.82 | 28.79 | 28.79±0.02 | 28.76±0.01 |
| Cui et al. (2021) | WRN-34-20 | 62.55 | 30.20 | 30.83 | 30.03 | 29.99 | 29.86 | 29.84±0.01 | 29.85±0.01 |
| Gowal et al. (2020) | WRN-70-16 | 60.86 | 30.03 | 30.54 | 30.00 | 30.00 | 29.99 | 29.97±0.01 | 29.96±0.01 |
| Gowal et al. (2020) | WRN-70-16 | 69.15 | 36.88 | 37.84 | 36.95 | 36.90 | 36.86 | 36.87±0.02 | 36.81±0.01 |
| Rebuffi et al. (2021) | WRN-70-16 | 63.56 | 34.64 | 35.30 | 34.68 | 34.63 | 34.62 | 34.62±0.01 | 34.55±0.01 |
| ImageNet ($\varepsilon = 4/255$) | model | clean | AA | ACG | $MT_{cos}$ | CAA | AutoAE | $A^3$ | EDA |
| Salman et al. (2020) | RN-18 | 52.92 | 25.32 | 26.40 | 25.24 | 25.28 | OOM | 25.22±0.03 | 25.11±0.02 |
| Engstrom et al. (2019) | RN-50 | 62.56 | 29.22 | 31.54 | 29.34 | 29.41 | OOM | 29.32±0.05 | 29.01±0.01 |
| Salman et al. (2020) | RN-50 | 64.02 | 34.96 | 36.26 | 34.68 | 34.77 | OOM | 34.75±0.04 | 34.52±0.02 |
| Wong et al. (2020) | RN-50 | 55.62 | 26.24 | 28.46 | 26.40 | 26.57 | OOM | 26.42±0.04 | 26.12±0.10 |
| Salman et al. (2020) | WRN-50-2 | 68.46 | 38.14 | 40.24 | 38.22 | 38.23 | OOM | 38.26±0.02 | 38.03±0.02 |

For the entire attack procedure, the standard AA requires queries of $6100 \times$ #images. CAA requires at least $60t \times$ #samples queries. AutoAE runs 32 iterations of APGD with CE loss, 63 iterations of APGD with DLR loss, 160 iterations of FAB, and 378 iterations of MultiTargeted attack with nine target classes. According to the official implementation of AutoAE, the MultiTargeted attack runs 378 iterations for each target class. Therefore, AutoAE requires $(32+63+160+378\times9) \times$ #images $= 3,657 \times$ #images queries for the adversarial attack. EDA requires $n_a \times (N_1 + N_2 + N_3) \times$ # images$= 5 \times 100 \times$ #images queries for the MDO framework, and $K \times N_s + N_4 \times$ # images$= 190\text{-}300 \times$ #images queries for the targeted attack $a^t$. Therefore, EDA requires 692.24 to $802.24 \times$ # images queries in total. We compared the runtime of EDA with that of $A^3$ because $A^3$ automatically terminates its search before the query limit.

Table 12: The computation time in seconds and ratio of runtime to EDA.

| CIFAR-10 ($\varepsilon = 8/255$) | model | clean acc | EDA sec | ratio | A³ sec | ratio | CAA sec | ratio | AutoAE sec | ratio |
|---|---|---|---|---|---|---|---|---|---|---|
| Andriushchenko & Flammarion (2020) | PARN-18 | 79.84 | 513±32 | 1.0 | 382±2 | 0.74 | 999±4 | 1.95 | 7,782 | 15.17 |
| Addepalli et al. (2022) | RN-18 | 85.71 | 625±25 | 1.0 | 434±2 | 0.69 | 1,107±3 | 1.77 | 8,503 | 13.60 |
| Sehwag et al. (2022) | RN-18 | 84.59 | 589±36 | 1.0 | 1,121±68 | 1.90 | 1,104±2 | 1.87 | 8,183 | 13.89 |
| Engstrom et al. (2019) | RN-50 | 87.03 | 1,485±78 | 1.0 | 1,572±11 | 1.06 | 3,672±5 | 2.47 | 28,263 | 19.03 |
| Carmon et al. (2019) | WRN-28-10 | 89.69 | 3,316±65 | 1.0 | 4,223±4 | 1.27 | 8,957±5 | 2.70 | 64,382 | 19.42 |
| Gowal et al. (2020) | WRN-28-10 | 89.48 | 4,557±112 | 1.0 | 3,841±13 | 0.84 | 11,358±8 | 2.49 | 80,517 | 17.67 |
| Hendrycks et al. (2019) | WRN-28-10 | 87.11 | 3,121±45 | 1.0 | 2,719±50 | 0.87 | 8,455±7 | 2.71 | 62,232 | 19.94 |
| Rebuffi et al. (2021) | WRN-28-10 | 87.33 | 4,459±74 | 1.0 | 3,928±30 | 0.88 | 11,225±11 | 2.52 | 79,293 | 17.78 |
| Sehwag et al. (2020) | WRN-28-10 | 88.98 | 3,255±64 | 1.0 | 2,662±50 | 0.82 | 8,761±7 | 2.69 | 64,273 | 19.75 |
| Sridhar et al. (2022) | WRN-28-10 | 89.46 | 3,355±55 | 1.0 | 3,245±119 | 0.97 | 8,768±8 | 2.61 | 64,021 | 19.08 |
| Wang et al. (2020) | WRN-28-10 | 87.50 | 3,285±78 | 1.0 | 2,732±4 | 0.83 | 8,688±7 | 2.64 | 63,662 | 19.38 |
| Wu et al. (2020) | WRN-28-10 | 88.25 | 3,502±39 | 1.0 | 3,273±8 | 0.93 | 8,920±8 | 2.55 | 64,568 | 18.44 |
| Ding et al. (2020) | WRN-28-4 | 84.36 | 695±29 | 1.0 | 2,017±90 | 2.90 | 1,591±2 | 2.29 | 12,828 | 18.46 |
| Addepalli et al. (2022) | WRN-34-10 | 88.71 | 4,225±109 | 1.0 | 3,926±7 | 0.93 | 11,183±8 | 2.65 | 80,337 | 19.01 |
| Sehwag et al. (2022) | WRN-34-10 | 86.68 | 4,172±18 | 1.0 | 3,858±6 | 0.92 | 11,067±14 | 2.65 | 79,526 | 19.06 |
| Sitawarin et al. (2021) | WRN-34-10 | 86.84 | 3,591±103 | 1.0 | 3,845±22 | 1.07 | 10,675±5 | 2.97 | 79,519 | 22.14 |
| Zhang et al. (2019) | WRN-34-10 | 87.20 | 3,219±32 | 1.0 | 3,500±12 | 1.09 | 10,395±8 | 3.23 | 79,064 | 24.56 |
| Zhang et al. (2020) | WRN-34-10 | 84.52 | 3,936±134 | 1.0 | 3,912±16 | 0.99 | 10,593±7 | 2.69 | 76,724 | 19.49 |
| Sridhar et al. (2022) | WRN-34-15 | 86.53 | 7,785±116 | 1.0 | 6,805±14 | 0.87 | 20,170±9 | 2.59 | 14,9077 | 19.15 |
| Gowal et al. (2020) | WRN-34-20 | 85.64 | 14,693±284 | 1.0 | 13,463±38 | 0.92 | 40,072±21 | 2.73 | 301,256 | 20.50 |
| Pang et al. (2020) | WRN-34-20 | 85.14 | 18,775±241 | 1.0 | 12,436±21 | 0.66 | 31,712±25 | 1.69 | 242,493 | 12.92 |
| Rice et al. (2020) | WRN-34-20 | 85.34 | 11,255±377 | 1.0 | 12,290±5 | 1.09 | 32,478±55 | 2.89 | 244,587 | 21.73 |
| Gowal et al. (2020) | WRN-70-16 | 85.29 | 21,790±430 | 1.0 | 26,587±1,032 | 1.22 | 60,431±71 | 2.77 | 451,566 | 20.72 |
| Gowal et al. (2020) | WRN-70-16 | 91.10 | 24,885±371 | 1.0 | 29,544±252 | 1.19 | 63,761±54 | 2.56 | 468,883 | 18.84 |
| Rebuffi et al. (2021) | WRN-70-16 | 88.54 | 24,652±479 | 1.0 | 29,075±887 | 1.18 | 63,433±35 | 2.57 | 459,409 | 18.64 |
| CIFAR-100 ($\varepsilon = 8/255$) | model | clean acc | EDA sec | ratio | A³ sec | ratio | CAA sec | ratio | AutoAE sec | ratio |
| Rice et al. (2020) | PARN-18 | 53.83 | 497±74 | 1.0 | 1,531±924 | 3.08 | 1,062±130 | 2.14 | 14,409 | 28.99 |
| Hendrycks et al. (2019) | WRN-28-10 | 59.23 | 1,981±38 | 1.0 | 2,684±10 | 1.35 | 5,508±57 | 2.78 | 75,410 | 38.07 |
| Rebuffi et al. (2021) | WRN-28-10 | 62.41 | 2,701±86 | 1.0 | 3,044±8 | 1.13 | 7,494±60 | 2.77 | 98,909 | 36.62 |
| Addepalli et al. (2022) | WRN-34-10 | 68.75 | 2,792±95 | 1.0 | 3,046±17 | 1.09 | 8,045±83 | 2.88 | 109,690 | 39.29 |
| Cui et al. (2021) | WRN-34-10 | 60.64 | 3,075±27 | 1.0 | 3,002±7 | 0.98 | 7,121±76 | 2.32 | 95,323 | 31.00 |
| Sitawarin et al. (2021) | WRN-34-10 | 62.82 | 1,985±40 | 1.0 | 4,935±137 | 2.49 | 7,081±86 | 3.57 | 99,096 | 49.92 |
| Wu et al. (2020) | WRN-34-10 | 60.38 | 2,360±31 | 1.0 | 3,258±58 | 1.38 | 7,073±66 | 3.00 | 96,679 | 40.97 |
| Cui et al. (2021) | WRN-34-20 | 62.55 | 8,027±197 | 1.0 | 9,798±8 | 1.22 | 22,894±265 | 2.85 | 313,505 | 39.06 |
| Gowal et al. (2020) | WRN-70-16 | 60.86 | 13,060±348 | 1.0 | 21,452±701 | 1.64 | 42,325±393 | 3.24 | 563,747 | 43.17 |
| Gowal et al. (2020) | WRN-70-16 | 69.15 | 15,641±467 | 1.0 | 22,423±1,969 | 1.43 | 49,029±476 | 3.13 | 633,166 | 40.48 |
| Rebuffi et al. (2021) | WRN-70-16 | 63.56 | 15,474±404 | 1.0 | 21,546±190 | 1.39 | 45,074±428 | 2.91 | 575,450 | 37.19 |
| ImageNet ($\varepsilon = 4/255$) | model | clean acc | EDA sec | ratio | A³ sec | ratio | CAA sec | ratio | AutoAE sec | ratio |
| Salman et al. (2020) | RN-18 | 52.92 | 1,667±119 | 1.0 | 2,937±10 | 1.76 | 1,511±13 | 0.91 | OOM | OOM |
| Engstrom et al. (2019) | RN-50 | 62.56 | 3,159±136 | 1.0 | 9,380±188 | 2.97 | 4,776±16 | 1.51 | OOM | OOM |
| Salman et al. (2020) | RN-50 | 64.02 | 3,525±302 | 1.0 | 9,989±234 | 2.83 | 4,997±27 | 1.42 | OOM | OOM |
| Wong et al. (2020) | RN-50 | 55.62 | 4,459±110 | 1.0 | 8,472±194 | 1.90 | 7,320±31 | 1.64 | OOM | OOM |
| Salman et al. (2020) | WRN-50-2 | 68.46 | 5,102±119 | 1.0 | 9,886±120 | 1.94 | 9,267±79 | 1.82 | OOM | OOM |

## D.3 Mathematical definition of the best point sets of A³ and EDA

Mathematically, the best point sets of $A^3$ and EDA are defined as follows. First, the best point set of $A^3$ is defined as $X_i^{a_{A^3}}$, where $a_{A^3} = (\mathbf{x}_{\mathrm{ADI}}, \eta_{\cos}, \boldsymbol{\delta}_{\mathrm{GD}}, L_{\mathrm{CW}})$ and $\mathbf{x}_{\mathrm{ADI}}$ is Adaptive Direction Initialization (ADI) proposed by Liu et al. (2022c). Subsequently, the best point set of EDA is defined as $(\cup_{a^* \in e^*} X_i^{a^*}) \cup X_i^{a^t}$, where $e^* = \left\{ (\mathbf{x}, \eta, \boldsymbol{\delta}_a, L_a) \mid a \in \{a_1^* \ldots, a_{n_a}^*, \hat{a}_1, \ldots, \hat{a}_{n_a}\} \right\}$ and $a^t = (\mathbf{x}_{\mathrm{PAS}}, \eta_{\mathrm{APGD}}, \boldsymbol{\delta}_{\mathrm{GD}}, L_{\mathrm{CW}}^T)$.

## D.4 Analysis of EDA using an index based on Euclid distance

DI takes small values when the point set forms a cluster, even if the Euclidean distance between any two points is large. Therefore, quantification by DI and quantification based on the Euclidean distance between points in the point set may have different characteristics. Therefore, in this section, to compare the diversification performance from a different perspective than DI, we consider quantifying the degree of diversification of the best point set based on the average value of the Euclidean distance between the centroid of the point set $X$ and all points in the point set $X$. Mathematically, the average Euclidean distance between all points in

Table 13: Comparisons with MDO framework(ADS), MDO framework(R-ADS), and MDO framework(RAND) in robust accuracy to validate ADS. The abbreviations are the same as those used in the main text. The lowest robust accuracies are in bold.

| | | MDO framework | | | Ensemble | Composite |
|---|---|---|---|---|---|---|
| CIFAR-10 ($\varepsilon = 8/255$) | model | RAND | R-ADS | ADS | ADS | ADS |
| (Andriushchenko & Flammarion, 2020) | PARN-18 | 44.09 | 44.02 | **43.95** | 44.12 | 43.95 |
| (Addepalli et al., 2022) | RN-18 | **52.45** | 52.83 | 52.50 | 52.63 | 52.65 |
| (Sehwag et al., 2022) | RN-18 | 55.61 | **55.58** | **55.58** | 55.68 | 55.90 |
| (Engstrom et al., 2019) | RN-50 | 49.60 | **49.40** | 49.52 | 49.79 | 50.04 |
| (Carmon et al., 2019) | WRN-28-10 | 59.56 | 59.53 | **59.46** | 59.70 | 59.59 |
| (Gowal et al., 2020) | WRN-28-10 | **62.82** | 62.99 | 62.83 | 62.92 | 62.86 |
| (Hendrycks et al., 2019) | WRN-28-10 | 54.94 | 55.02 | **54.87** | 55.09 | 55.14 |
| (Rebuffi et al., 2021) | WRN-28-10 | **60.67** | 60.84 | 60.77 | 60.82 | 60.82 |
| (Sehwag et al., 2020) | WRN-28-10 | 57.16 | 57.38 | **57.14** | 57.43 | 57.34 |
| (Sridhar et al., 2022) | WRN-28-10 | 59.63 | 59.76 | **59.59** | 59.89 | 59.85 |
| (Wang et al., 2020) | WRN-28-10 | **56.32** | 56.50 | 56.33 | 56.66 | 56.87 |
| (Wu et al., 2020) | WRN-28-10 | 60.02 | 60.13 | **59.99** | 60.14 | 60.13 |
| (Ding et al., 2020) | WRN-28-4 | 43.54 | 44.88 | **43.24** | 43.49 | 44.11 |
| (Addepalli et al., 2022) | WRN-34-10 | 57.75 | 57.87 | **57.72** | 57.92 | 57.90 |
| (Sehwag et al., 2022) | WRN-34-10 | 60.26 | 60.38 | **60.21** | 60.52 | 60.44 |
| (Sitawarin et al., 2021) | WRN-34-10 | 50.83 | 50.78 | **50.70** | 50.98 | 51.14 |
| (Zhang et al., 2019) | WRN-34-10 | 44.78 | 44.70 | **44.62** | 44.87 | 44.79 |
| (Zhang et al., 2020) | WRN-34-10 | 53.51 | **53.50** | 53.55 | 53.60 | 53.52 |
| (Sridhar et al., 2022) | WRN-34-15 | 60.40 | 60.46 | **60.39** | 60.46 | 60.48 |
| (Gowal et al., 2020) | WRN-34-20 | 56.86 | **56.85** | 56.88 | 56.90 | 56.84 |
| (Pang et al., 2020) | WRN-34-20 | **53.77** | 53.85 | 53.81 | 54.00 | 53.96 |
| (Rice et al., 2020) | WRN-34-20 | 53.47 | 53.48 | **53.42** | 53.52 | 53.50 |
| (Gowal et al., 2020) | WRN-70-16 | 57.23 | 57.21 | **57.18** | 57.27 | 57.22 |
| (Gowal et al., 2020) | WRN-70-16 | 65.86 | 66.02 | **65.85** | 66.04 | 65.99 |
| (Rebuffi et al., 2021) | WRN-70-16 | **64.23** | 64.54 | 64.32 | 64.53 | 64.43 |
| CIFAR-100 ($\varepsilon = 8/255$) | | | | | | |
| (Rice et al., 2020) | PARN-18 | 18.98 | 18.99 | **18.97** | 19.08 | 19.12 |
| (Hendrycks et al., 2019) | WRN-28-10 | 28.56 | 28.83 | **28.44** | 28.61 | 28.45 |
| (Rebuffi et al., 2021) | WRN-28-10 | **32.08** | 32.13 | **32.08** | 32.22 | 32.18 |
| (Addepalli et al., 2022) | WRN-34-10 | 31.91 | 32.23 | **31.86** | 31.91 | 31.97 |
| (Cui et al., 2021) | WRN-34-10 | **28.97** | 29.20 | 28.99 | 29.20 | 29.13 |
| (Sitawarin et al., 2021) | WRN-34-10 | 24.71 | 24.68 | **24.65** | 24.74 | 24.83 |
| (Wu et al., 2020) | WRN-34-10 | 28.93 | 29.46 | **28.88** | 28.97 | 28.97 |
| (Cui et al., 2021) | WRN-34-20 | 30.07 | 30.35 | **30.01** | 30.15 | 30.29 |
| (Gowal et al., 2020) | WRN-70-16 | 30.06 | 30.42 | **30.05** | 30.11 | 30.18 |
| (Gowal et al., 2020) | WRN-70-16 | 37.05 | 37.53 | **36.96** | 37.19 | 37.17 |
| (Rebuffi et al., 2021) | WRN-70-16 | **34.61** | 34.97 | 34.65 | 34.88 | 34.65 |
| ImageNet ($\varepsilon = 4/255$) | | | | | | |
| (Salman et al., 2020) | RN-18 | **25.22** | 25.44 | **25.22** | 25.46 | 25.48 |
| (Engstrom et al., 2019) | RN-50 | 29.56 | 29.26 | **29.20** | 29.64 | 29.68 |
| (Salman et al., 2020) | RN-50 | **34.68** | **34.68** | 34.84 | 35.00 | 34.92 |
| (Wong et al., 2020) | RN-50 | 26.26 | 26.36 | **26.22** | 26.84 | 26.78 |
| (Salman et al., 2020) | WRN-50-2 | 38.38 | 38.54 | **38.28** | 38.52 | 38.58 |

a point set $X$ is defined as

$$E(X) = \frac{1}{|X|} \sum_{x \in X} \|x - \bar{x}\|_2, \tag{22}$$

where $\bar{x}$ is the centroid of the point set $X$, defined as $\bar{x} = \frac{1}{|X|} \sum_{x \in X} x$. As shown in Figure 20, the value of

Table 14: The quantified degree of diversification. DI denotes the Diversity Index, and E denotes the metric defined by equation 22. RAND, R-ADS, and ADS represent MDO framework(RAND), MDO framework(R-ADS), and MDO framework(ADS), respectively.

| CIFAR-10 ($\varepsilon = 8/255$) | Models | $A^3$ | | RAND | | R-ADS | | ADS | | EDA | |
|---|---|---|---|---|---|---|---|---|---|---|---|
| | | DI | E | DI | E | DI | E | DI | E | DI | E |
| Andriushchenko & Flammarion (2020) | PARN-18 | 0.26±0.09 | 0.66±0.27 | 0.30±0.06 | 0.95±0.20 | 0.25±0.05 | 0.73±0.14 | 0.33±0.07 | 0.98±0.18 | 0.36±0.05 | 1.11±0.15 |
| Addepalli et al. (2022) | RN-18 | 0.27±0.10 | 0.69±0.28 | 0.31±0.07 | 0.94±0.20 | 0.23±0.05 | 0.78±0.11 | 0.36±0.08 | 1.02±0.19 | 0.40±0.06 | 1.18±0.17 |
| Sehwag et al. (2022) | RN-18 | 0.27±0.09 | 0.72±0.28 | 0.31±0.06 | 0.93±0.20 | 0.26±0.05 | 0.79±0.15 | 0.34±0.06 | 0.99±0.19 | 0.38±0.05 | 1.15±0.15 |
| Engstrom et al. (2019) | RN-50 | 0.28±0.09 | 0.72±0.27 | 0.33±0.06 | 0.99±0.19 | 0.25±0.05 | 0.76±0.15 | 0.37±0.07 | 1.04±0.19 | 0.39±0.05 | 1.17±0.16 |
| Carmon et al. (2019) | WRN-28-10 | 0.26±0.08 | 0.65±0.25 | 0.33±0.06 | 1.00±0.18 | 0.26±0.05 | 0.84±0.14 | 0.38±0.07 | 1.06±0.19 | 0.41±0.05 | 1.22±0.17 |
| Gowal et al. (2020) | WRN-28-10 | 0.22±0.10 | 0.58±0.30 | 0.30±0.07 | 0.94±0.21 | 0.20±0.05 | 0.71±0.11 | 0.37±0.08 | 1.04±0.22 | 0.41±0.06 | 1.20±0.18 |
| Hendrycks et al. (2019) | WRN-28-10 | 0.25±0.09 | 0.64±0.27 | 0.30±0.06 | 0.94±0.20 | 0.22±0.04 | 0.71±0.12 | 0.34±0.07 | 1.01±0.19 | 0.37±0.05 | 1.14±0.15 |
| Rebuffi et al. (2021) | WRN-28-10 | 0.24±0.10 | 0.63±0.30 | 0.30±0.06 | 0.89±0.21 | 0.24±0.05 | 0.75±0.12 | 0.34±0.07 | 0.97±0.18 | 0.37±0.05 | 1.09±0.16 |
| Sehwag et al. (2020) | WRN-28-10 | 0.25±0.08 | 0.64±0.26 | 0.33±0.06 | 0.99±0.19 | 0.23±0.04 | 0.75±0.11 | 0.38±0.07 | 1.09±0.20 | 0.43±0.05 | 1.24±0.19 |
| Sridhar et al. (2022) | WRN-28-10 | 0.25±0.09 | 0.64±0.26 | 0.33±0.06 | 1.00±0.18 | 0.24±0.04 | 0.76±0.10 | 0.37±0.07 | 1.05±0.18 | 0.41±0.05 | 1.24±0.16 |
| Wang et al. (2020) | WRN-28-10 | 0.28±0.08 | 0.68±0.24 | 0.31±0.06 | 0.93±0.18 | 0.25±0.06 | 0.79±0.12 | 0.35±0.07 | 0.99±0.18 | 0.35±0.06 | 1.04±0.15 |
| Wu et al. (2020) | WRN-28-10 | 0.25±0.09 | 0.64±0.27 | 0.32±0.06 | 0.98±0.19 | 0.23±0.04 | 0.76±0.10 | 0.38±0.07 | 1.08±0.19 | 0.43±0.06 | 1.25±0.16 |
| Ding et al. (2020) | WRN-28-4 | 0.24±0.10 | 0.91±0.34 | 0.33±0.07 | 0.98±0.22 | 0.29±0.07 | 0.91±0.23 | 0.36±0.08 | 1.02±0.22 | 0.38±0.07 | 1.19±0.16 |
| Addepalli et al. (2022) | WRN-34-10 | 0.27±0.09 | 0.67±0.28 | 0.31±0.06 | 0.94±0.20 | 0.24±0.05 | 0.83±0.13 | 0.37±0.08 | 1.04±0.20 | 0.41±0.06 | 1.21±0.17 |
| Sehwag et al. (2022) | WRN-34-10 | 0.25±0.09 | 0.65±0.26 | 0.32±0.06 | 0.96±0.18 | 0.24±0.04 | 0.74±0.12 | 0.37±0.07 | 1.02±0.18 | 0.38±0.05 | 1.15±0.14 |
| Sitawarin et al. (2021) | WRN-34-10 | 0.27±0.10 | 0.72±0.30 | 0.32±0.06 | 0.99±0.20 | 0.24±0.05 | 0.74±0.17 | 0.33±0.06 | 1.03±0.20 | 0.38±0.05 | 1.18±0.16 |
| Zhang et al. (2019) | WRN-34-10 | 0.27±0.11 | 0.73±0.31 | 0.30±0.06 | 0.96±0.21 | 0.23±0.05 | 0.70±0.16 | 0.32±0.07 | 1.00±0.21 | 0.37±0.05 | 1.18±0.17 |
| Zhang et al. (2020) | WRN-34-10 | 0.25±0.09 | 0.62±0.26 | 0.31±0.06 | 0.95±0.20 | 0.25±0.05 | 0.82±0.17 | 0.33±0.07 | 0.99±0.20 | 0.39±0.06 | 1.21±0.16 |
| Sridhar et al. (2022) | WRN-34-15 | 0.23±0.08 | 0.57±0.24 | 0.31±0.07 | 0.96±0.19 | 0.24±0.05 | 0.78±0.11 | 0.37±0.08 | 1.06±0.19 | 0.41±0.06 | 1.21±0.16 |
| Gowal et al. (2020) | WRN-34-20 | 0.21±0.11 | 0.55±0.32 | 0.29±0.07 | 0.95±0.22 | 0.21±0.05 | 0.73±0.15 | 0.34±0.07 | 1.02±0.21 | 0.38±0.05 | 1.19±0.18 |
| Pang et al. (2020) | WRN-34-20 | 0.24±0.09 | 0.66±0.30 | 0.22±0.08 | 0.80±0.23 | 0.16±0.07 | 0.64±0.25 | 0.25±0.10 | 0.87±0.21 | 0.31±0.08 | 1.06±0.19 |
| Rice et al. (2020) | WRN-34-20 | 0.24±0.11 | 0.64±0.32 | 0.28±0.06 | 0.91±0.22 | 0.22±0.05 | 0.71±0.17 | 0.33±0.07 | 0.98±0.21 | 0.37±0.05 | 1.14±0.17 |
| Gowal et al. (2020) | WRN-70-16 | 0.22±0.10 | 0.56±0.30 | 0.30±0.06 | 0.94±0.20 | 0.21±0.05 | 0.71±0.12 | 0.38±0.08 | 1.04±0.20 | 0.39±0.05 | 1.18±0.17 |
| Gowal et al. (2020) | WRN-70-16 | 0.19±0.10 | 0.52±0.32 | 0.28±0.06 | 0.94±0.22 | 0.22±0.05 | 0.76±0.15 | 0.31±0.07 | 1.01±0.21 | 0.38±0.05 | 1.19±0.16 |
| Rebuffi et al. (2021) | WRN-70-16 | 0.23±0.09 | 0.59±0.29 | 0.30±0.06 | 0.88±0.20 | 0.23±0.05 | 0.74±0.12 | 0.34±0.07 | 0.96±0.17 | 0.37±0.05 | 1.09±0.16 |

| CIFAR-100 ($\varepsilon = 8/255$) | Models | $A^3$ | | RAND | | R-ADS | | ADS | | EDA | |
|---|---|---|---|---|---|---|---|---|---|---|---|
| | | DI | E | DI | E | DI | E | DI | E | DI | E |
| Rice et al. (2020) | PARN-18 | 0.34±0.13 | 0.83±0.31 | 0.32±0.06 | 0.97±0.20 | 0.25±0.06 | 0.78±0.18 | 0.35±0.07 | 1.02±0.21 | 0.39±0.06 | 1.15±0.20 |
| Hendrycks et al. (2019) | WRN-28-10 | 0.27±0.11 | 0.71±0.30 | 0.29±0.07 | 0.94±0.22 | 0.22±0.06 | 0.75±0.16 | 0.31±0.08 | 0.97±0.21 | 0.36±0.06 | 1.13±0.18 |
| Rebuffi et al. (2021) | WRN-28-10 | 0.32±0.15 | 0.81±0.36 | 0.29±0.07 | 0.94±0.23 | 0.25±0.06 | 0.82±0.17 | 0.35±0.08 | 1.02±0.22 | 0.41±0.06 | 1.20±0.20 |
| Addepalli et al. (2022) | WRN-34-10 | 0.34±0.13 | 0.84±0.30 | 0.31±0.07 | 0.97±0.21 | 0.24±0.06 | 0.86±0.13 | 0.36±0.09 | 1.04±0.21 | 0.42±0.06 | 1.21±0.18 |
| Cui et al. (2021) | WRN-34-10 | 0.27±0.10 | 0.67±0.27 | 0.27±0.08 | 0.89±0.22 | 0.23±0.08 | 0.82±0.14 | 0.30±0.09 | 0.97±0.19 | 0.34±0.07 | 1.04±0.17 |
| Sitawarin et al. (2021) | WRN-34-10 | 0.31±0.12 | 0.79±0.30 | 0.32±0.06 | 0.98±0.20 | 0.24±0.05 | 0.75±0.17 | 0.35±0.07 | 1.02±0.21 | 0.38±0.06 | 1.14±0.20 |
| Wu et al. (2020) | WRN-34-10 | 0.28±0.12 | 0.73±0.32 | 0.29±0.07 | 0.95±0.22 | 0.24±0.05 | 0.79±0.15 | 0.32±0.08 | 0.99±0.21 | 0.37±0.06 | 1.16±0.18 |
| Cui et al. (2021) | WRN-34-20 | 0.27±0.09 | 0.66±0.27 | 0.29±0.08 | 0.92±0.21 | 0.22±0.06 | 0.75±0.13 | 0.32±0.09 | 0.99±0.21 | 0.35±0.07 | 1.07±0.18 |
| Gowal et al. (2020) | WRN-70-16 | 0.27±0.13 | 0.69±0.34 | 0.28±0.07 | 0.93±0.22 | 0.19±0.05 | 0.66±0.13 | 0.33±0.08 | 0.98±0.23 | 0.37±0.06 | 1.19±0.17 |
| Gowal et al. (2020) | WRN-70-16 | 0.31±0.14 | 0.79±0.33 | 0.31±0.07 | 0.94±0.22 | 0.23±0.06 | 0.73±0.14 | 0.33±0.08 | 1.01±0.21 | 0.40±0.06 | 1.21±0.17 |
| Rebuffi et al. (2021) | WRN-70-16 | 0.29±0.14 | 0.76±0.36 | 0.30±0.07 | 0.93±0.23 | 0.21±0.06 | 0.74±0.14 | 0.34±0.08 | 0.99±0.20 | 0.40±0.05 | 1.19±0.19 |

| ImageNet ($\varepsilon = 4/255$) | Models | $A^3$ | | RAND | | R-ADS | | ADS | | EDA | |
|---|---|---|---|---|---|---|---|---|---|---|---|
| | | DI | E | DI | E | DI | E | DI | E | DI | E |
| Salman et al. (2020) | RN-18 | 0.26±0.07 | 2.60±0.91 | 0.34±0.07 | 3.52±0.69 | 0.28±0.05 | 2.83±0.42 | 0.38±0.08 | 3.66±0.69 | 0.43±0.04 | 4.37±0.66 |
| Engstrom et al. (2019) | RN-50 | 0.29±0.06 | 2.79±0.83 | 0.38±0.05 | 3.69±0.58 | 0.31±0.06 | 3.18±0.51 | 0.40±0.06 | 3.88±0.61 | 0.44±0.04 | 4.30±0.64 |
| Salman et al. (2020) | RN-50 | 0.26±0.06 | 2.61±0.90 | 0.35±0.06 | 3.61±0.65 | 0.30±0.06 | 3.03±0.47 | 0.37±0.07 | 3.70±0.63 | 0.42±0.04 | 4.43±0.53 |
| Wong et al. (2020) | RN-50 | 0.27±0.07 | 3.45±1.12 | 0.36±0.06 | 4.59±0.82 | 0.30±0.06 | 3.76±0.58 | 0.38±0.07 | 4.81±0.80 | 0.43±0.05 | 5.43±0.89 |
| Salman et al. (2020) | WRN-50-2 | 0.27±0.06 | 2.63±0.82 | 0.37±0.06 | 3.69±0.60 | 0.34±0.06 | 3.53±0.58 | 0.38±0.05 | 3.80±0.61 | 0.43±0.04 | 4.22±0.65 |

equation 22 tends to be larger for EDA than for $A^3$ in most models where EDA has higher attack performance than $A^3$. This difference is more pronounced than the difference in DI. While EDA shows a similar trend for all models, $A^3$ shows a different trend in the value of equation 22 for some models. For example, as shown in Figure 20, the value of equation 22 for $A^3$ tends to be larger for the model proposed by Ding et al. (2020) than for the other models. Given the high attack performance of $A^3$ against these models, this suggests that the $A^3$ diversification strategy may be more effective for these models.

## D.5 Analysis of EDA for the model proposed by Ding et al.

The attack performance of EDA is significantly lower for the model proposed by Ding et al. (2020) compared to $A^3$. This section discusses the reasons for this regarding diversification performance and computation time. As described in the main text, the value of DI for the best point set tends to be higher for EDA and lower for $A^3$, similar to the results for other models. On the other hand, the analysis in the previous section shows that for the model proposed by Ding et al. (2020), the value of equation 22 for the best point set obtained by $A^3$ tends to take larger values than the results for the other models. In addition, a comparison of the computation time for EDA and $A^3$ shows that $A^3$ takes more than three times longer than EDA. The above comparison suggests that the $A^3$ can perform better diversification for the model than for other models. In summary, setting a longer computation time and increasing the number of multi-restart are

Table 15: Comparison in robust accuracy for transfer setting. "source→target" indicates that the transfer attack from the source to the target model. AutoAE was executed only with CIFAR10 because of its long execution time.

CIFAR-10

| source→target | clean acc (target model) | robust acc (in RobustBench) | AA | CAA | AutoAE | A³ | EDA |
|---|---|---|---|---|---|---|---|
| (Carmon et al., 2019)→(Ding et al., 2020) | 84.36 | 41.44 | 71.82 | 73.15 | 73.24 | 70.46 | **68.14** |
| (Carmon et al., 2019)→(Cui et al., 2021) | 88.22 | 52.86 | 71.90 | 74.76 | 74.66 | 73.46 | **70.29** |
| (Ding et al., 2020)→(Carmon et al., 2019) | 89.69 | 59.53 | **83.23** | 85.01 | 84.85 | 85.16 | 83.68 |
| (Ding et al., 2020)→(Cui et al., 2021) | 88.22 | 52.86 | **79.77** | 82.16 | 81.83 | 82.42 | 80.32 |
| (Cui et al., 2021)→(Carmon et al., 2019) | 89.69 | 59.53 | **76.17** | 80.24 | 80.32 | 79.94 | 77.24 |
| (Cui et al., 2021)→(Ding et al., 2020) | 84.36 | 41.44 | 70.16 | 72.07 | 72.09 | 71.28 | **69.43** |

CIFAR-100

| source→target | clean acc (target model) | robust acc (in RobustBench) | AA | CAA | AutoAE | A³ | EDA |
|---|---|---|---|---|---|---|---|
| Rice et al. (2020)→Cui et al. (2021) | 62.55 | 30.20 | **56.42** | 58.72 | - | 58.36 | 58.10 |
| Rice et al. (2020)→Rebuffi et al. (2021) | 63.56 | 34.64 | **58.28** | 60.45 | - | 59.74 | 59.55 |
| Cui et al. (2021)→Rice et al. (2020) | 53.83 | 18.95 | 44.51 | 45.87 | - | 44.50 | **42.86** |
| Cui et al. (2021)→Rebuffi et al. (2021) | 63.56 | 34.64 | **53.58** | 56.24 | - | 55.35 | 54.02 |
| Rebuffi et al. (2021)→Rice et al. (2020) | 53.83 | 18.95 | 44.44 | 45.64 | - | 43.51 | **40.84** |
| Rebuffi et al. (2021)→Cui et al. (2021) | 62.55 | 30.20 | 50.15 | 51.98 | - | 51.26 | **48.45** |

ImageNet

| source→target | clean acc (target model) | robust acc (in RobustBench) | AA | CAA | AutoAE | A³ | EDA |
|---|---|---|---|---|---|---|---|
| Salman et al. (2020)→Engstrom et al. (2019) | 62.52 | 29.22 | 53.74 | 55.88 | - | 55.20 | **52.76** |
| Salman et al. (2020)→Wong et al. (2020) | 55.64 | 26.24 | 47.22 | 47.90 | - | 46.40 | **45.12** |
| Engstrom et al. (2019)→Salman et al. (2020) | 68.46 | 38.14 | **63.62** | 65.34 | - | 64.68 | 64.46 |
| Engstrom et al. (2019)→Wong et al. (2020) | 55.64 | 26.24 | **44.64** | 46.34 | - | 46.06 | 44.92 |
| Wong et al. (2020)→Salman et al. (2020) | 68.46 | 38.14 | **65.36** | 66.08 | - | 65.56 | 65.42 |
| Wong et al. (2020)→Engstrom et al. (2019) | 62.52 | 29.22 | **56.40** | 58.02 | - | 57.80 | 56.72 |

Table 16: Results of the preliminary experiments to determine the hyperparameters of ADS. The robust accuracy obtained by the MDO framework is described. The default parameters are in bold.

| Dataset | No. | $N_{\mathrm{ADS}}$ | | | | $n_a$ | | | |
|---|---|---|---|---|---|---|---|---|---|
| | | 3 | **4** | 5 | 10 | 3 | 4 | **5** | 6 |
| CIFAR-10 | 1 | 55.56 | 55.58 | 55.65 | 55.67 | 55.63 | 55.66 | 55.58 | 55.57 |
| CIFAR-10 | 2 | 56.83 | 56.80 | 56.84 | 56.81 | 56.90 | 56.83 | 56.80 | 56.78 |
| CIFAR-100 | 3 | 19.01 | 18.87 | 19.03 | 19.01 | 19.16 | 19.10 | 18.87 | 18.90 |
| CIFAR-100 | 4 | 24.66 | 24.56 | 24.61 | 24.59 | 24.61 | 24.78 | 24.56 | 24.56 |
| ImageNet | 5 | 29.40 | 29.24 | 29.26 | 29.30 | 29.28 | 29.34 | 29.24 | 29.14 |

considered particularly effective in improving the attack performance for the model proposed by Ding et al. (2020).

### D.6 Trends of search directions and objective functions selected by ADS

Figure 21 is a bar chart displaying the ratio of times each search direction and objective function used by EDA. Figure 21 shows that $\boldsymbol{\delta}_{\mathrm{ACG}}$ and $L_{\mathrm{G\text{-}DLR},q}$ are frequently used in the diversification phase, and $\boldsymbol{\delta}_{\mathrm{Nes}}$ is rarely used. In the intensification phase, $L_{\mathrm{CW}}, L_{\mathrm{SCW}}$, and $\boldsymbol{\delta}_{\mathrm{Nes}}$ were more likely to be selected. This trend may reflect ACG's high diversification performance and NAG's high intensification performance. The potential reasons for these trends are: 1. $P_i^e$ and DI play different roles from each other, 2. the ACG's search direction may be similar to the steepest for small step sizes, and 3. the difference between Nesterov's acceleration gradient direction and gradient direction.

**The role of $P_i^e$ and DI.** The $P_i^e$ measures the degree of diversification in the output space during the search. A pair with the largest $P_i^e$ is expected to show a high diversity in the output space. In addition, from Yamamura et al. (2022), it can be assumed that the ACG direction increases $P_i^e$, while the steepest-like

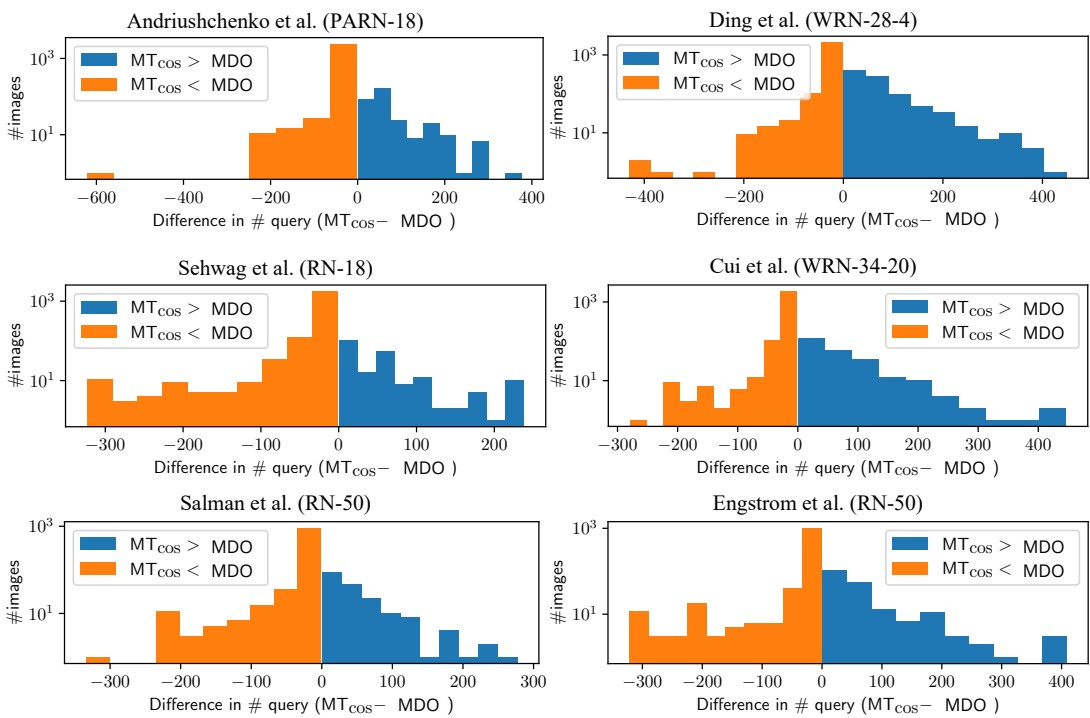

Figure 18: The difference between $MT_{cos}$ and the MDO framework in #queries to find adversarial examples for some models. The attacked models are Andriushchenko & Flammarion (2020), Sehwag et al. (2022), and Ding et al. (2020) for CIFAR-10, Cui et al. (2021) for CIFAR-100, and Salman et al. (2020) and Engstrom et al. (2019) for ImageNet.

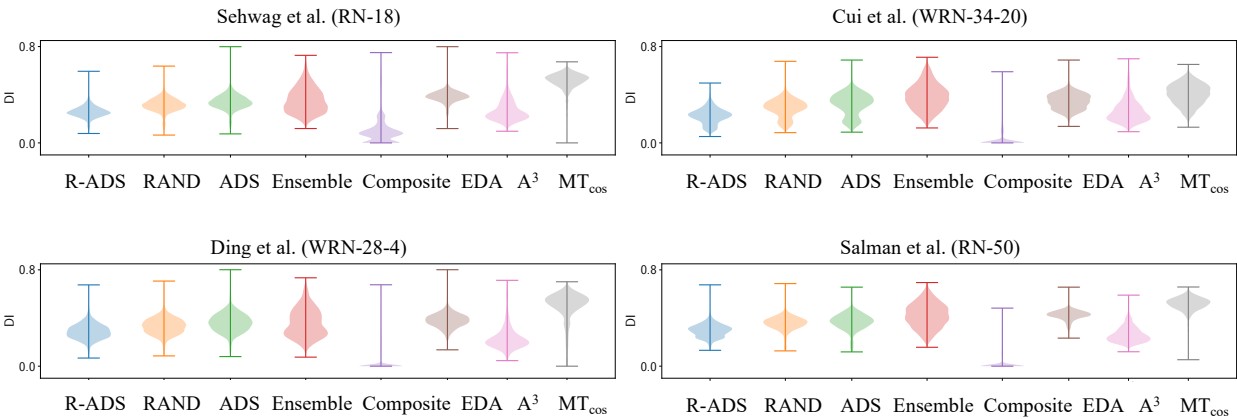

Figure 19: Violin plot of DI. The attacked models are Sehwag et al. (2022) and Ding et al. (2020) for CIFAR-10, Cui et al. (2021) for CIFAR-100, and Salman et al. (2020) for ImageNet.

direction does not. From the above, it is considered that the pair with the maximum $P_i^e$ is likely to include the ACG direction. DI measures the diversity of the best point set obtained by the search. In our use case, DI represents the dissimilarity between the best points. That is, we expect that pairs with the largest DI are more likely to enumerate dissimilar solutions. Intuitively, updates in diverse directions contribute to the

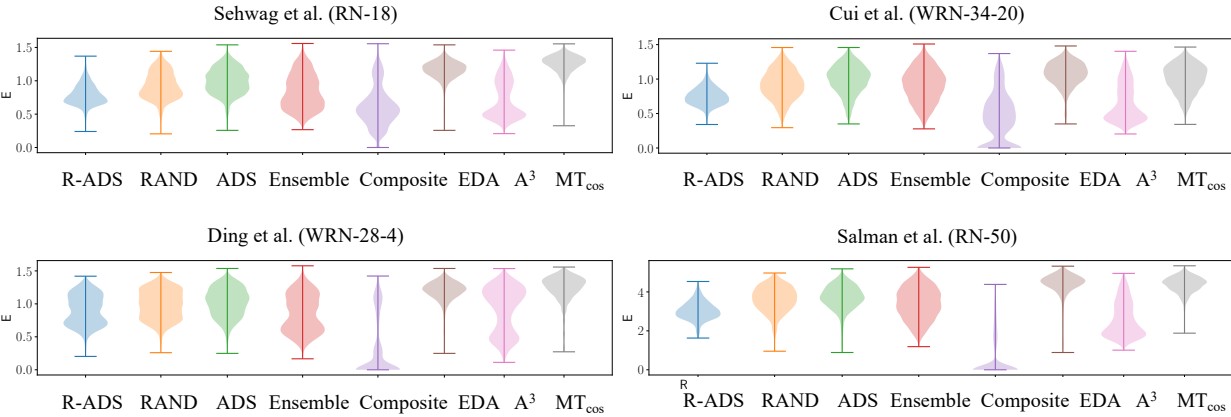

Figure 20: Violin plot of equation 22. The attacked models are Sehwag et al. (2022) and Ding et al. (2020) for CIFAR-10, Cui et al. (2021) for CIFAR-100, and Salman et al. (2020) for ImageNet.

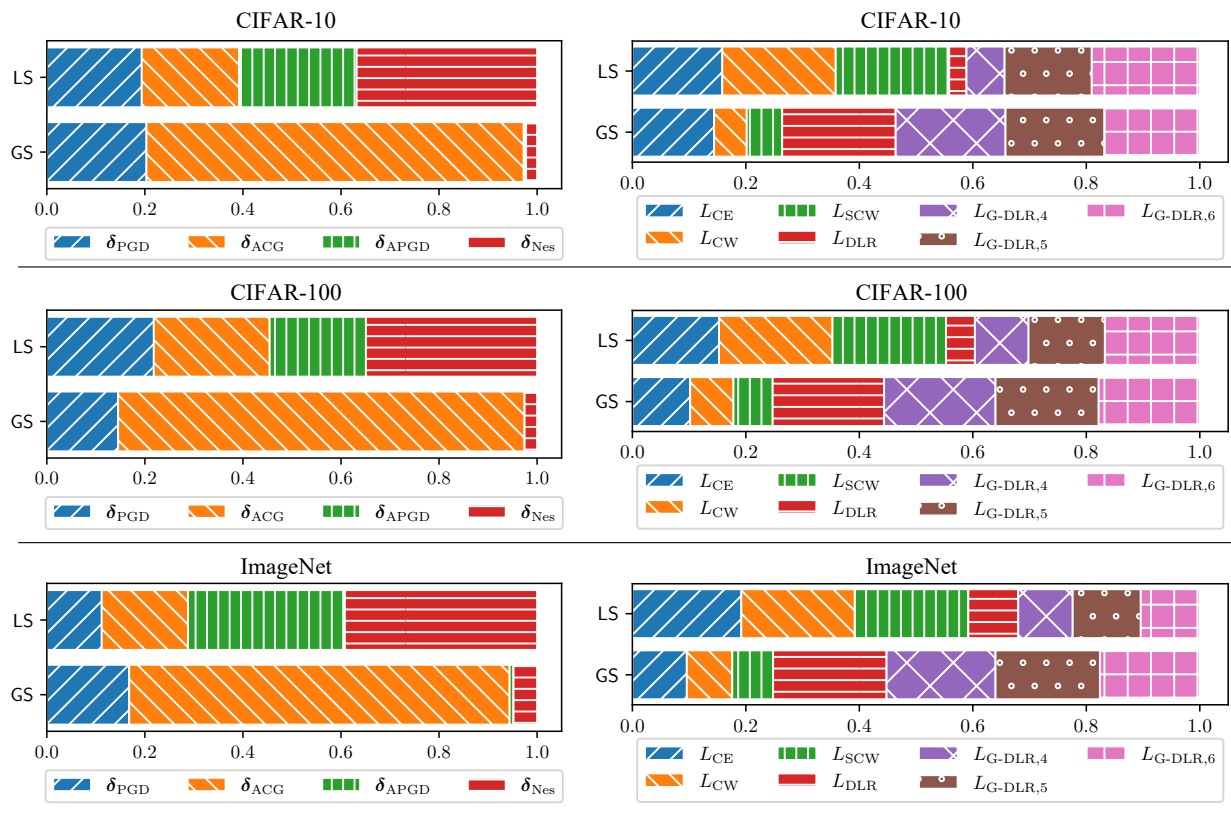

Figure 21: The average ratio of times each search direction and objective function used by EDA. LS and GS refer to the intensification phase and diversification phase, respectively.

enumeration of dissimilar solutions. Given that the search direction is gradient-dependent, the pair with the largest DI is likely to include a variety of objective functions and update formulas.

**ACG's search direction may be similar to the steepest for small step sizes.** The reason for this is as follows. According to equation 10, $\boldsymbol{s}^{(k)}$ is close to the gradient $\nabla L(g(\boldsymbol{x}^{(k)}), c)$ when $\beta^{(k)}$ is close to 0. From equation 9, $\langle \nabla L(g(\boldsymbol{x}^{(k)}), c), \boldsymbol{y}^{(k-1)} \rangle$ is the numerator of $\beta^{(k)}$. Therefore, as $\boldsymbol{y}^{(k-1)}$ approaches $\boldsymbol{0}$, $\beta^{(k)}$ also approaches 0. When the step size is small, $\|x^{(k)} - x^{(k-1)}\|$ is also small, so $\boldsymbol{y}^{(k-1)}$ is likely to be close to $\boldsymbol{0}$. As a result, the ACG's direction and the steepest direction may be similar. The experiments conducted by Yamamura et al. (2022) also support this claim.

**The difference between Nesterov's acceleration gradient direction and gradient direction.** Nesterov's accelerated gradient (NAG) method updates the search point using the gradient of the point moved from the current search point to the momentum direction. Assuming that the objective function is multimodal, the gradient at the current point is unlikely to be similar to NAG's search direction. Thus, if the objective function is multimodal, a search in the NAG's direction may find different local solutions from that in the gradient direction.

# E  Prediction Aware Sampling (PAS)

**Motivation** The hypothesis behind PAS is that starting the search with an initial point near multiple decision boundaries increases the likelihood of finding an adversarial example. When maximizing the inner product of a random vector and logit, as ODS does, the distance to the decision boundary may be farther away than the initial point. However, when moving in the direction where the predicted probability for the correct class is as small as possible, the initial point is more likely to be closer to the decision boundary than the original point. We hypothesized that the attack's success rate could be improved by starting the search at a point closer to the decision boundary.

**Prediction Aware Sampling** One promising initial point sampling is ODS, which considers diversification in the output space. However, there is room for improvement because its sampling does not consider image-specific information. Based on the idea that the randomly sampled initial point close to decision boundaries makes the attacks easier to succeed, we propose Prediction-Aware Sampling (PAS), a variant of ODS. PAS maximizes the following function in the same way as ODS to sample the initial point.

$$v(\boldsymbol{w}, g, \boldsymbol{x}) = \boldsymbol{w}^T g(\boldsymbol{x}) \times \exp\left(-g_c(\boldsymbol{x})\right), \quad \left(\boldsymbol{w} \sim U\left(-1, 1\right)^C\right) \tag{23}$$

PAS samples the initial point by repeating the following updates for $N_{\text{PAS}}$ iterations.

$$\boldsymbol{x} \leftarrow P_{\mathcal{S}}\left(\boldsymbol{x} + \eta_{\text{PAS}} \operatorname{sign}\left(\frac{\nabla_{\boldsymbol{x}} v(\boldsymbol{w}, g, \boldsymbol{x})}{\|\nabla_{\boldsymbol{x}} v(\boldsymbol{w}, g, \boldsymbol{x})\|_2}\right)\right) \tag{24}$$

Same as ODS, PAS used $N_{\text{PAS}} = 2$ and $\eta_{\text{PAS}} = \varepsilon$. Intuitively, maximizing equation 23 means moving the initial point closer to the decision boundary by reducing the prediction probability of the correct class $c$ and, at the same time, moving the logit $g(\boldsymbol{x})$ closer to the random vector $\boldsymbol{w}$.

**Experiments** To test our hypothesis, we compared the success rate for each class in targeted attacks with nine target classes, using the input point, the point sampled by ODS, and the point sampled by PAS as initial points. In our notation, we compared the attack performance of $(\mathbf{x}, \eta_{\cos}, \boldsymbol{\delta}_{\text{GD}}, L_{\text{CW}}^T), \mathbf{x} \in \{\mathbf{x}_{\text{org}}, \mathbf{x}_{\text{ODS}}, \mathbf{x}_{\text{PAS}}\}$ with 100 iterations for each target class and initial step size of $2\varepsilon$. The number of target classes $K$ was set to 9. The following five models were used in the experiments. 1. ResNet-18 (Sehwag et al., 2022), 2. WideResNet-28-10 (Gowal et al., 2020), 3. PreActResNet-18 (Rice et al., 2020), 4. WideResNet-34-10 (Sitawarin et al., 2021), 5. ResNet-50 (Engstrom et al., 2019). These numbers correspond to the "No." column in Table 17. The experimental results in Table 17 show that the attack with PAS can achieve higher attack success rates for many target classes than other initial point selections. The experimental results support our hypothesis that PAS brings the starting point closer to the decision boundary, resulting in a more successful attack. As described in Appendix G.2, the ablation results for the initial point of the EDA also indicate that the PAS contributes to the attack performance of the EDA and the MDO framework.

Table 17: Validation of PAS. The lowest robust accuracy is in bold.

| Dataset | No. | target | input | ODS | PAS |
|---------|-----|--------|-------|-----|-----|
| CIFAR-10 | 1 | 1 | 57.41 | 57.40 | **57.29** |
|          |   | 2 | 65.33 | 64.82 | **64.28** |
|          |   | 3 | 67.85 | 66.98 | **66.27** |
|          |   | 4 | 69.06 | 68.21 | **67.52** |
|          |   | 5 | 69.88 | 68.77 | **67.81** |
|          |   | 6 | 69.66 | 68.59 | **67.88** |
|          |   | 7 | 69.11 | 68.37 | **67.59** |
|          |   | 8 | 68.08 | 67.47 | **67.10** |
|          |   | 9 | 67.32 | 66.84 | **66.46** |
| CIFAR-10 | 2 | 1 | **57.52** | 57.54 | 57.57 |
|          |   | 2 | 65.93 | 65.79 | **65.33** |
|          |   | 3 | 68.88 | 68.63 | **68.15** |
|          |   | 4 | 70.62 | 70.34 | **69.33** |
|          |   | 5 | 71.41 | 70.95 | **70.00** |
|          |   | 6 | 72.37 | 71.82 | **70.71** |
|          |   | 7 | 72.57 | 71.99 | **70.80** |
|          |   | 8 | 73.09 | 72.49 | **71.31** |
|          |   | 9 | 72.86 | 72.28 | **71.08** |
| CIFAR-100 | 3 | 1 | 20.37 | 20.39 | **20.34** |
|          |   | 2 | 24.71 | 24.67 | **24.46** |
|          |   | 3 | 26.65 | 26.64 | **26.14** |
|          |   | 4 | 27.70 | 27.49 | **26.99** |
|          |   | 5 | 28.37 | 28.36 | **27.70** |
|          |   | 6 | 29.02 | 28.98 | **28.33** |
|          |   | 7 | 29.55 | 29.32 | **28.62** |
|          |   | 8 | 29.83 | 29.68 | **28.95** |
|          |   | 9 | 29.72 | 29.55 | **28.94** |
| CIFAR-100 | 4 | 1 | 27.25 | 27.22 | **27.17** |
|          |   | 2 | 31.87 | 31.74 | **31.50** |
|          |   | 3 | 33.70 | 33.73 | **33.18** |
|          |   | 4 | 34.92 | 34.72 | **34.17** |
|          |   | 5 | 36.00 | 35.91 | **35.13** |
|          |   | 6 | 36.55 | 36.50 | **35.58** |
|          |   | 7 | 36.75 | 36.56 | **35.87** |
|          |   | 8 | 37.34 | 37.20 | **36.39** |
|          |   | 9 | 37.81 | 37.54 | **36.73** |
| ImageNet | 5 | 1 | 32.66 | 32.62 | **32.42** |
|          |   | 2 | 38.04 | 37.98 | **37.68** |
|          |   | 3 | 39.64 | 39.64 | **39.22** |
|          |   | 4 | 41.50 | 41.34 | **40.74** |
|          |   | 5 | 41.68 | 41.48 | **41.00** |
|          |   | 6 | 42.82 | 42.80 | **42.20** |
|          |   | 7 | 43.10 | 42.92 | **42.46** |
|          |   | 8 | 43.22 | 43.02 | **42.52** |
|          |   | 9 | 43.20 | 43.12 | **42.58** |

Table 18: Validation of targeted attack

| Dataset | No. | EDA | EDA+ |
|---------|-----|-----|------|
| CIFAR-10 | 1 | **55.49** | **55.49** |
| CIFAR-10 | 2 | **56.75** | 56.78 |
| CIFAR-100 | 3 | **18.86** | 18.87 |
| CIFAR-100 | 4 | **24.47** | 24.48 |
| ImageNet | 5 | **29.00** | 29.02 |

Table 19: Ablation study of EDA. The default parameter is in bold.

| | | $N_s$ | | | # sampled images | | | |
|---------|-----|-------|--------|-------|-------|-------|-------|-------|
| Dataset | No. | 5 | **10** | 15 | **1%** | 3% | 5% | 7% |
| CIFAR-10 | 1 | 55.59 | 55.49 | 55.49 | 55.49 | 55.47 | 55.48 | 55.50 |
| CIFAR-10 | 2 | 56.73 | 56.76 | 56.76 | 56.76 | 56.81 | 56.81 | 56.79 |
| CIFAR-100 | 3 | 18.85 | 18.86 | 18.86 | 18.86 | 18.87 | 18.88 | 18.87 |
| CIFAR-100 | 4 | 24.46 | 24.47 | 24.48 | 24.47 | 24.49 | 24.49 | 24.47 |
| ImageNet | 5 | 29.04 | 29.04 | 29.04 | 29.04 | 29.04 | 29.08 | 29.04 |

## F   Targeted attack in EDA

**Motivation**   The motivation for using a targeted attack is to efficiently diversify the most likely prediction class of the adversarial example away from the correct class (diversification in the output space). CW loss and DLR loss are objective functions that generate adversarial examples misclassified into the class with the highest prediction probability among classes other than the correct class. In other words, they attempt to generate an adversarial example misclassified into the class whose decision boundary is closest to the current point. However, it is difficult to approach the decision boundary when the gradient is zero, even if the distance to the decision boundary is close because the gradient-based attack moves in the direction of the gradient. In addition, Yamamura et al. (2022) reported that in the steepest gradient-based attacks, the class with the highest prediction probability among classes other than the correct class hardly changes during the search. Considering these factors, diversification in the output space could be effective, especially for attacks with untargeted losses. Some existing research also supports the effectiveness of diversification in the output space (Tashiro et al., 2020; Gowal et al., 2019).

**Target selection**   Although a multi-target attack shows high performance by achieving diversification in the output space, existing methods (Gowal et al., 2019; Croce & Hein, 2020b) are computationally expensive because they assign an equal number of iterations to each target. To reduce the computational cost of the multi-target attack, we propose Target Selection (TS), which estimates the easiest target class to attack based on a small-scale search. TS estimates the easiest target class to attack based on a small-scale search to reduce the computational cost of the multi-target attack. The objective is to reduce the computational cost by focusing the number of iterations on the selected target. The procedure of TS is as follows: (1) Upper $K$ classes with large logit values of initial points are selected as target candidates. (2) Targeted attacks are performed for $N_s$ iterations for each target candidate. (3) The output is the target candidate $T$ with the highest objective function value.

**Hyperparameters**   The parameters of the targeted attack are the number of candidate targets in target selection $K = 9, 14, 20$, the number of iterations $N_s = 10$, and the number of iterations in targeted attack $N = 100$. We chose $N = 100$ because the number of iterations per targeted attack in AA is set to 100, which achieves a reasonable trade-off between computational cost and attack performance.

**Experiments**   To investigate the validity of this target selection, we compare the attack performance of EDA+, which executes a normal targeted attack after the MDO framework, with that of EDA. In this experiment, we performed $(\mathbf{x}_{\mathrm{PAS}}, \eta_{\mathrm{cos}}, \boldsymbol{\delta}_{\mathrm{GD}}, L_{\mathrm{CW}}^T)$ with 100 iterations for each target class and the initial step size of $2\varepsilon$. The following five models were used in the experiments. 1. ResNet-18 (Sehwag et al., 2022), 2. WideResNet-28-10 (Gowal et al., 2020), 3. PreActResNet-18 (Rice et al., 2020), 4. WideResNet-34-10 (Sitawarin et al., 2021), 5. ResNet-50 (Engstrom et al., 2019). These numbers correspond to the "No." column in Table 18. Although the normal targeted attack requires $K \times N = 100K$ queries, our targeted attack scheme selects a single target class and thus requires $N_s \times K + N = 10K + 100$ queries per image. Given the parameter of $K = 9, 14, 20 \geq 2$, our targeted attack scheme requires fewer queries than the normal targeted attack. Considering the results described in Table 18, our target selection may reduce runtime without significantly degrading the attack performance of EDA.

Table 20: Ablation study of EDA. The default parameter is in bold.

| Dataset | No. | initial stepsize | | | | $N_{\mathrm{ADS}}$ | | | | $n_a$ | | | |
|---|---|---|---|---|---|---|---|---|---|---|---|---|---|
| | | $\varepsilon/4$ | $\varepsilon/2$ | $\varepsilon$ | $\mathbf{2\varepsilon}$ | 3 | **4** | 5 | 10 | 3 | 4 | **5** | 6 |
| CIFAR-10 | 1 | 55.50 | 55.48 | 55.51 | 55.49 | 55.47 | 55.49 | 55.50 | 55.51 | 55.50 | 55.48 | 55.48 | 55.49 |
| CIFAR-10 | 2 | 56.78 | 56.78 | 56.78 | 56.75 | 56.78 | 56.76 | 56.77 | 56.76 | 56.77 | 56.82 | 56.76 | 56.76 |
| CIFAR-100 | 3 | 18.92 | 18.92 | 18.87 | 18.86 | 18.88 | 18.86 | 18.89 | 18.89 | 18.87 | 18.90 | 18.89 | 18.86 |
| CIFAR-100 | 4 | 24.49 | 24.50 | 24.50 | 24.47 | 24.47 | 24.47 | 24.49 | 24.48 | 24.48 | 24.49 | 24.48 | 24.47 |
| ImageNet | 5 | 29.10 | 29.08 | 29.08 | 29.00 | 29.00 | 29.04 | 29.02 | 29.04 | 29.06 | 29.04 | 29.04 | 29.02 |

Table 21: Ablation for the targeted attack. The default setting is in bold.

| Dataset | No. | $\delta_{\mathrm{GD}}$ | | | $\delta_{\mathrm{CG}}$ | | $\delta_{\mathrm{APGD}}$ |
|---|---|---|---|---|---|---|---|
| | | $\boldsymbol{L}^T_{\mathrm{CW}}$ | $L^T_{\mathrm{CE}}$ | $L^T_{\mathrm{DLR}}$ | $L^T_{\mathrm{CW}}$ | $L^T_{\mathrm{DLR}}$ | $L^T_{\mathrm{DLR}}$ |
| CIFAR-10 | 1 | 55.49 | 55.57 | 55.48 | 55.50 | 55.50 | 55.48 |
| CIFAR-10 | 2 | 56.76 | 56.89 | 56.81 | 56.81 | 56.83 | 56.79 |
| CIFAR-100 | 3 | 18.86 | 18.96 | 18.90 | 18.93 | 18.94 | 18.91 |
| CIFAR-100 | 4 | 24.47 | 24.63 | 24.50 | 24.53 | 24.53 | 24.50 |
| ImageNet | 5 | 29.00 | 29.18 | 29.02 | 29.08 | 29.06 | 29.00 |

# G  Ablation study

## G.1  Hyperparameter sensitivity of EDA

We investigated the impact of hyperparameter values on EDA's performance. In this experiment, the following five models were used. 1. ResNet-18 (Sehwag et al., 2022), 2. WideResNet-28-10 (Gowal et al., 2020), 3. PreActResNet-18 (Rice et al., 2020), 4. WideResNet-34-10 (Sitawarin et al., 2021), 5. ResNet-50 (Engstrom et al., 2019). Tables 19 and 20 show the robust accuracy obtained by EDA with each parameter value. Although the attack performance of the MDO framework is different among different hyperparameters, as described in Table 16, the EDA's performance is stable regardless of the hyperparameter setting. These experimental results imply that the attacks composed by different strategies could be robust to the hyperparameter settings. In addition, we tested the EDA's performance with several targeted attacks. As described in Table 21, the search directions based on the steepest direction performed better than the conjugate gradient-based direction. Also, the margin-based losses showed higher performance than the CE loss.

## G.2  The impact of initial point sampling on the performance of the MDO framework and EDA

Table 22 shows the robust accuracy of the MDO framework and EDA using different initial point sampling, including original input ($\mathbf{x}_{\mathrm{org}}$), Output Diversified Sampling (ODS, $\mathbf{x}_{\mathrm{ODS}}$), and Prediction Aware Sampling (PAS, $\mathbf{x}_{\mathrm{PAS}}$). the MDO framework and EDA showed slightly better performance using PAS than using the original input. The difference in robust accuracy between the MDO framework using the original input and ODS was larger than that using PAS and the original input. For EDA, the difference in robust accuracy among the three initial point sampling was smaller than that of the MDO framework. These experimental results suggest that initial point sampling does affect attack performance but that differences in the framework and the pairs of objective function and search direction have a greater impact.

## G.3  Influence of DI and $P^e$ on ADS's performance

To investigate the influence of two terms that appear in the computation of indicators used in ADS, we executed the MDO framework and EDA with the pairs of search direction and objective function selected using DI, $P^e$, and $D^e$. The model under attack is a subset of the 41 CNN-based robust models used in the main text. The allowable perturbation size, test samples, and computing environment are the same as in the experiments described in the text. The results of the experiments are shown in Table 23. DI, $P^e$, and $D^e$ in Table 23 indicate that the pairs of search direction and objective function were selected using these indicators,

Table 22: Robust accuracy of the ablation study for the MDO framework and EDA with different initial point sampling.

| CIFAR-10 | model | MDO framework | | | EDA | | |
|---|---|---|---|---|---|---|---|
| $(\varepsilon = 8/255)$ | | $x_{org}$ | $x_{ODS}$ | $x_{PAS}$ | $x_{org}$ | $x_{ODS}$ | $x_{PAS}$ |
| (Andriushchenko & Flammarion, 2020) | PARN-18 | 43.96±0.03 | 44.05±0.07 | 43.95±0.02 | 43.85±0.02 | 43.86±0.01 | 43.85±0.02 |
| (Addepalli et al., 2022) | RN-18 | 52.50±0.03 | 52.66±0.12 | 52.49±0.03 | 52.41±0.01 | 52.45±0.02 | 52.43±0.03 |
| (Sehwag et al., 2022) | RN-18 | 55.62±0.03 | 55.60±0.02 | 55.56±0.02 | 55.49±0.01 | 55.50±0.01 | 55.49±0.01 |
| (Engstrom et al., 2019) | RN-50 | 49.40±0.03 | 49.34±0.06 | 49.43±0.09 | 49.11±0.01 | 49.10±0.01 | 49.10±0.03 |
| (Carmon et al., 2019) | WRN-28-10 | 59.49±0.03 | 59.62±0.03 | 59.49±0.03 | 59.40±0.01 | 59.45±0.02 | 59.40±0.01 |
| (Gowal et al., 2020) | WRN-28-10 | 62.84±0.02 | 62.99±0.03 | 62.82±0.01 | 62.75±0.01 | 62.78±0.02 | 62.75±0.02 |
| (Hendrycks et al., 2019) | WRN-28-10 | 54.87±0.03 | 54.84±0.01 | 54.85±0.04 | 54.77±0.01 | 54.77±0.01 | 54.77±0.02 |
| (Rebuffi et al., 2021) | WRN-28-10 | 60.78±0.04 | 60.83±0.07 | 60.75±0.03 | 60.69±0.02 | 60.69±0.02 | 60.64±0.01 |
| (Sehwag et al., 2020) | WRN-28-10 | 57.11±0.03 | 57.22±0.05 | 57.11±0.02 | 57.01±0.02 | 57.07±0.03 | 57.03±0.01 |
| (Sridhar et al., 2022) | WRN-28-10 | 59.62±0.02 | 59.68±0.10 | 59.55±0.03 | 59.48±0.03 | 59.52±0.03 | 59.46±0.02 |
| (Wang et al., 2020) | WRN-28-10 | 56.48±0.06 | 56.52±0.24 | 56.29±0.02 | 56.18±0.01 | 56.20±0.04 | 56.15±0.02 |
| (Wu et al., 2020) | WRN-28-10 | 60.01±0.02 | 60.11±0.09 | 60.00±0.02 | 59.94±0.01 | 59.97±0.03 | 59.94±0.01 |
| (Ding et al., 2020) | WRN-28-4 | 43.33±0.16 | 43.69±0.13 | 43.32±0.12 | 41.70±0.04 | 41.84±0.03 | 41.74±0.06 |
| (Addepalli et al., 2022) | WRN-34-10 | 57.78±0.01 | 58.01±0.09 | 57.75±0.03 | 57.70±0.01 | 57.74±0.01 | 57.69±0.02 |
| (Sehwag et al., 2022) | WRN-34-10 | 60.27±0.02 | 60.30±0.02 | 60.27±0.05 | 60.17±0.01 | 60.18±0.02 | 60.18±0.02 |
| (Sitawarin et al., 2021) | WRN-34-10 | 50.72±0.04 | 50.69±0.03 | 50.72±0.03 | 50.59±0.02 | 50.60±0.02 | 50.59±0.01 |
| (Zhang et al., 2019) | WRN-34-10 | 44.62±0.03 | 44.65±0.04 | 44.65±0.06 | 44.51±0.02 | 44.52±0.02 | 44.51±0.02 |
| (Zhang et al., 2020) | WRN-34-10 | 53.53±0.03 | 53.56±0.03 | 53.50±0.03 | 53.43±0.01 | 53.44±0.03 | 53.42±0.02 |
| (Sridhar et al., 2022) | WRN-34-15 | 60.43±0.02 | 60.44±0.11 | 60.38±0.03 | 60.31±0.02 | 60.32±0.03 | 60.32±0.01 |
| (Gowal et al., 2020) | WRN-34-20 | 56.89±0.02 | 57.08±0.05 | 56.86±0.02 | 56.79±0.01 | 56.81±0.01 | 56.79±0.03 |
| (Pang et al., 2020) | WRN-34-20 | 53.85±0.03 | 53.80±0.03 | 53.82±0.03 | 53.64±0.00 | 53.67±0.01 | 53.66±0.01 |
| (Rice et al., 2020) | WRN-34-20 | 53.47±0.02 | 53.56±0.09 | 53.46±0.04 | 53.33±0.01 | 53.34±0.01 | 53.34±0.01 |
| (Gowal et al., 2020) | WRN-70-16 | 57.21±0.02 | 57.28±0.05 | 57.18±0.02 | 57.12±0.00 | 57.14±0.01 | 57.12±0.01 |
| (Gowal et al., 2020) | WRN-70-16 | 65.92±0.04 | 65.91±0.05 | 65.87±0.01 | 65.81±0.03 | 65.81±0.02 | 65.83±0.01 |
| (Rebuffi et al., 2021) | WRN-70-16 | 64.31±0.04 | 64.36±0.06 | 64.30±0.05 | 64.20±0.01 | 64.20±0.02 | 64.20±0.03 |
| CIFAR-100 | model | the MDO framework | | | EDA | | |
| $(\varepsilon = 8/255)$ | | $x_{org}$ | $x_{ODS}$ | $x_{PAS}$ | $x_{org}$ | $x_{ODS}$ | $x_{PAS}$ |
| (Rice et al., 2020) | PARN-18 | 18.97±0.01 | 19.04±0.09 | 18.98±0.01 | 18.87±0.01 | 18.86±0.01 | 18.88±0.01 |
| (Hendrycks et al., 2019) | WRN-28-10 | 28.46±0.02 | 28.60±0.13 | 28.41±0.02 | 28.28±0.01 | 28.28±0.02 | 28.27±0.02 |
| (Rebuffi et al., 2021) | WRN-28-10 | 32.22±0.03 | 32.45±0.23 | 32.07±0.04 | 31.97±0.01 | 31.99±0.01 | 31.94±0.03 |
| (Addepalli et al., 2022) | WRN-34-10 | 31.90±0.03 | 32.01±0.16 | 31.87±0.02 | 31.78±0.01 | 31.78±0.01 | 31.78±0.01 |
| (Cui et al., 2021) | WRN-34-10 | 29.00±0.04 | 29.14±0.23 | 28.96±0.05 | 28.85±0.02 | 28.84±0.02 | 28.83±0.02 |
| (Sitawarin et al., 2021) | WRN-34-10 | 24.61±0.04 | 24.67±0.07 | 24.66±0.03 | 24.50±0.01 | 24.48±0.01 | 24.50±0.01 |
| (Wu et al., 2020) | WRN-34-10 | 28.95±0.04 | 29.17±0.18 | 28.89±0.04 | 28.76±0.01 | 28.77±0.01 | 28.76±0.01 |
| (Cui et al., 2021) | WRN-34-20 | 30.06±0.02 | 30.11±0.10 | 30.00±0.02 | 29.86±0.01 | 29.86±0.01 | 29.85±0.01 |
| (Gowal et al., 2020) | WRN-70-16 | 30.13±0.03 | 30.29±0.21 | 30.07±0.01 | 29.97±0.01 | 29.99±0.01 | 29.96±0.01 |
| (Gowal et al., 2020) | WRN-70-16 | 37.06±0.04 | 37.17±0.15 | 36.96±0.04 | 36.81±0.01 | 36.83±0.01 | 36.81±0.01 |
| (Rebuffi et al., 2021) | WRN-70-16 | 34.69±0.04 | 34.81±0.09 | 34.64±0.03 | 34.57±0.02 | 34.58±0.02 | 34.55±0.01 |
| ImageNet | model | the MDO framework | | | EDA | | |
| $(\varepsilon = 4/255)$ | | $x_{org}$ | $x_{ODS}$ | $x_{PAS}$ | $x_{org}$ | $x_{ODS}$ | $x_{PAS}$ |
| (Salman et al., 2020) | RN-18 | 25.31±0.05 | 25.48±0.21 | 25.25±0.02 | 25.12±0.02 | 25.10±0.01 | 25.11±0.02 |
| (Engstrom et al., 2019) | RN-50 | 29.29±0.06 | 29.77±0.48 | 29.27±0.07 | 29.01±0.01 | 29.00±0.02 | 29.01±0.01 |
| (Salman et al., 2020) | RN-50 | 34.67±0.09 | 35.13±0.30 | 34.73±0.07 | 34.54±0.03 | 34.54±0.03 | 34.52±0.02 |
| (Wong et al., 2020) | RN-50 | 26.41±0.12 | 26.71±0.34 | 26.32±0.13 | 26.13±0.12 | 26.14±0.08 | 26.12±0.10 |
| (Salman et al., 2020) | WRN-50-2 | 38.42±0.04 | 39.36±0.10 | 38.31±0.06 | 38.04±0.02 | 38.03±0.03 | 38.03±0.02 |

respectively. From the experimental results, the highest attack performance is expected when $P^e$ is used to select the pairs. However, focusing on the MDO framework against CIFAR-10 and CIFAR-100 models, DI showed the highest number of models that performed better than the 2nd-best. Thus, the MDO framework is expected to show higher performance when the pairs are selected with DI than with $P^e$ for a wide range of models. The attack performance of the MDO framework with $D^e$ is often intermediate between that with DI and $P^e$. Thus, we used $D^e$ to select the pairs in the text, considering performance and generalization. The reason for the better performance in selecting the pairs that maximize $P^e$ in EDA can be attributed to its different nature from targeted attacks. Targeted attacks maximize the prediction probability of the target class. Thus, the class with the highest prediction probability, excluding the correct prediction, is unlikely to change during the search. In contrast, the pairs that maximize $P^e$ have a high probability that the class with the highest prediction probability, excluding the correct prediction, will change during the search. The pairs

Table 23: Influence of DI and $P^e$ on ADS's performance. The lowest and second lowest robust accuracy of each attack is in **bold** and underlined, respectively.

| defense | model | MDO framework | | | EDA | | |
|---|---|---|---|---|---|---|---|
| CIFAR-10 | | DI | $P^e$ | $D^e$ | DI | $P^e$ | $D^e$ |
| Andriushchenko & Flammarion (2020) | PARN-18 | **43.90** | 44.00 | 43.95 | **43.85** | **43.85** | 43.86 |
| Addepalli et al. (2022) | RN-18 | 52.52 | **52.45** | 52.50 | 52.42 | **52.41** | 52.43 |
| Sehwag et al. (2022) | RN-18 | 55.61 | 55.59 | **55.58** | 55.50 | 55.50 | **55.49** |
| Engstrom et al. (2019) | RN-50 | 49.50 | **49.31** | 49.52 | 49.11 | **49.09** | 49.12 |
| Carmon et al. (2019) | WRN-28-10 | 59.49 | 59.51 | **59.46** | 59.42 | 59.42 | **59.38** |
| Sehwag et al. (2020) | WRN-28-10 | **57.10** | **57.10** | 57.14 | 57.03 | **57.01** | 57.06 |
| Sridhar et al. (2022) | WRN-28-10 | **59.50** | 59.54 | 59.59 | **59.45** | 59.46 | 59.47 |
| Wu et al. (2020) | WRN-28-10 | 60.02 | 60.04 | **59.99** | 59.96 | 59.96 | **59.95** |
| Addepalli et al. (2022) | WRN-34-10 | 57.73 | **57.72** | **57.72** | 57.71 | **57.66** | 57.68 |
| Sitawarin et al. (2021) | WRN-34-10 | 50.75 | 50.85 | **50.70** | 50.62 | 50.60 | **50.58** |
| Pang et al. (2020) | WRN-34-20 | 53.81 | **53.74** | 53.81 | **53.65** | **53.65** | **53.65** |
| Gowal et al. (2020) | WRN-70-16 | 65.89 | **65.80** | 65.85 | 65.83 | **65.78** | 65.81 |
| # bold | | 3 | 6 | 5 | 3 | 7 | 5 |
| # underlined | | 6 | 2 | 5 | 5 | 4 | 3 |
| CIFAR-100 | | | | | | | |
| Rice et al. (2020) | PARN-18 | **18.91** | 19.01 | 18.97 | **18.88** | **18.88** | 18.89 |
| Addepalli et al. (2022) | WRN-34-10 | **31.81** | 31.85 | 31.86 | **31.77** | **31.77** | 31.78 |
| Cui et al. (2021) | WRN-34-10 | **28.93** | 28.96 | 28.99 | 28.85 | 28.82 | **28.81** |
| Sitawarin et al. (2021) | WRN-34-10 | 24.63 | **24.59** | 24.65 | 24.49 | **24.48** | 24.49 |
| Wu et al. (2020) | WRN-34-10 | **28.87** | 28.90 | 28.88 | 28.78 | 28.78 | **28.77** |
| Gowal et al. (2020) | WRN-70-16 | 37.03 | **36.94** | 36.96 | 36.84 | **36.79** | 36.80 |
| Rebuffi et al. (2021) | WRN-70-16 | 34.65 | **34.64** | 34.65 | 34.57 | **34.54** | 34.55 |
| # bold | | 4 | 3 | 0 | 2 | 5 | 2 |
| # underlined | | 2 | 2 | 4 | 2 | 2 | 5 |
| ImageNet | | | | | | | |
| Salman et al. (2020) | RN-18 | 25.20 | **25.16** | 25.22 | 25.12 | **25.06** | 25.10 |
| Engstrom et al. (2019) | RN-50 | 29.20 | **29.10** | 29.20 | **28.98** | 29.00 | 29.00 |
| Salman et al. (2020) | RN-50 | 34.64 | **34.62** | 34.84 | 34.52 | **34.50** | 34.56 |
| Wong et al. (2020) | RN-50 | 26.34 | **26.10** | 26.22 | 26.10 | **25.96** | 26.02 |
| Salman et al. (2020) | WRN-50-2 | 38.32 | 38.30 | **38.28** | 38.04 | **38.00** | 38.02 |
| # bold | | 0 | 4 | 2 | 1 | 4 | 0 |
| # underlined | | 3 | 1 | 1 | 1 | 1 | 4 |

that maximize $P^e$ are expected to be more complementary to the targeted attack than those that maximize DI.

### G.4 Performance evaluation with different perturbation bounds

This experiment evaluates the performance of EDA at different perturbation sizes with $\varepsilon = 4/255$ and $16/255$ for CIFAR-10/100 and $\varepsilon = 2/255$ and $8/255$ for ImageNet. We used AutoAttack (AA) and Adaptive Auto Attack ($A^3$) as baseline methods. AA is a standard technique for robustness evaluation, and $A^3$ showed superior performance in terms of execution time and attack performance in the experiments presented in the main text. The experimental setup is the same as in the main text, except for the perturbation size. Table 24 shows the robust accuracy and execution time. The performance difference between the different attacks is not that large for small $\varepsilon$, even for models trained on ImageNet. When $\varepsilon$ is large, $A^3$ shows the highest attack performance. However, the computation time of $A^3$ tends to increase significantly compared

Table 24: Performance evaluation of EDA with different perturbation bound $\varepsilon$.

**CIFAR-10**

| Defence | Model | $\varepsilon = 4/255$ AA acc | AA sec | $A^3$ acc | $A^3$ sec | EDA acc | EDA sec | $\varepsilon = 16/255$ AA acc | AA sec | $A^3$ acc | $A^3$ sec | EDA acc | EDA sec |
|---|---|---|---|---|---|---|---|---|---|---|---|---|---|
| Andriushchenko & Flammarion (2020) | PARN-18 | 63.24 | 4,066 | 63.24 | 633 | 63.24 | 613 | 10.45 | 786 | 10.30 | 470 | 10.18 | 231 |
| Addepalli et al. (2022) | RN-18 | - | - | 71.08 | 694 | 71.08 | 705 | - | - | 17.59 | 758 | 17.53 | 293 |
| Sehwag et al. (2022) | RN-18 | 71.98 | 4,616 | 71.96 | 705 | 71.96 | 647 | 20.55 | 1,412 | 20.07 | 868 | 20.11 | 293 |
| Engstrom et al. (2019) | RN-50 | - | - | 70.97 | 1,967 | 70.98 | 1,822 | - | - | 12.23 | 1,566 | 12.20 | 587 |
| Carmon et al. (2019) | WRN-28-10 | 77.59 | 38,346 | 77.58 | 6,244 | 77.59 | 4,042 | 20.35 | 10,906 | 19.96 | 5,995 | 20.00 | 1,429 |
| Gowal et al. (2020) | WRN-28-10 | 78.63 | 45,270 | 78.60 | 7,045 | 78.60 | 5,547 | 24.95 | 15,441 | 24.49 | 2,291 | 24.56 | 2,292 |
| Rebuffi et al. (2021) | WRN-28-10 | - | - | 75.81 | 6,789 | 75.80 | 5,235 | - | - | 24.96 | 2,697 | 25.01 | 2,250 |
| Sehwag et al. (2020) | WRN-28-10 | 75.80 | 37,244 | 75.76 | 5,945 | 75.78 | 4,030 | 18.76 | 9,959 | 18.59 | 1,716 | 18.55 | 1,403 |
| Sridhar et al. (2022) | WRN-28-10 | 77.77 | 38,391 | 77.74 | 6,209 | 77.74 | 4,180 | 20.50 | 10,944 | 20.10 | 6,036 | 20.18 | 1,468 |
| Wang et al. (2020) | WRN-28-10 | 74.52 | 36,749 | 74.51 | 5,892 | 74.51 | 4,150 | 17.89 | 9,685 | 17.48 | 5,353 | 17.77 | 1,336 |
| Wu et al. (2020) | WRN-28-10 | - | - | 76.38 | 6,041 | 76.38 | 4,131 | - | - | 23.59 | 2,299 | 23.50 | 1,635 |
| Ding et al. (2020) | WRN-28-4 | 63.39 | 6,471 | 63.36 | 946 | 63.33 | 864 | 12.15 | 1,540 | 12.27 | 870 | 13.72 | 368 |
| Addepalli et al. (2022) | WRN-34-10 | 75.26 | 46,846 | 75.25 | 7,460 | 75.26 | 5,104 | 20.40 | 13,667 | 19.91 | 2,221 | 19.89 | 1,798 |
| Sehwag et al. (2022) | WRN-34-10 | - | - | 75.75 | 7,581 | 75.77 | 4,992 | - | - | 24.47 | 9,489 | 24.66 | 2,029 |
| Sitawarin et al. (2021) | WRN-34-10 | 71.54 | 44,598 | 71.52 | 7,160 | 71.52 | 4,672 | 13.69 | 9,583 | 13.31 | 5,161 | 13.33 | 1,334 |
| Zhang et al. (2019) | WRN-34-10 | 69.15 | 43,120 | 69.11 | 6,930 | 69.13 | 4,499 | 8.38 | 6,229 | 8.08 | 13,233 | 8.18 | 968 |
| Zhang et al. (2020) | WRN-34-10 | 71.04 | 44,221 | 71.01 | 7,067 | 71.01 | 4,956 | 21.06 | 14,088 | 20.74 | 2,249 | 20.68 | 1,799 |
| Sridhar et al. (2022) | WRN-34-15 | 75.34 | 76,827 | 75.33 | 11,393 | 75.34 | 9,502 | 24.75 | 26,999 | 24.48 | 3,986 | 24.61 | 3,807 |
| Gowal et al. (2020) | WRN-34-20 | 73.32 | 162,529 | 73.32 | 21,954 | 73.33 | 17,833 | 22.96 | 53,608 | 22.55 | 8,119 | 22.57 | 6,545 |
| Rice et al. (2020) | WRN-34-20 | - | - | 72.06 | 18,871 | 72.06 | 14,227 | - | - | 15.33 | 15,510 | 15.44 | 4,185 |
| Gowal et al. (2020) | WRN-70-16 | 73.23 | 240,017 | 73.22 | 30,382 | 73.21 | 26,431 | 24.04 | 84,819 | 23.82 | 11,810 | 23.71 | 10,315 |

**CIFAR-100**

| Defence | Model | $\varepsilon = 4/255$ AA acc | AA sec | $A^3$ acc | $A^3$ sec | EDA acc | EDA sec | $\varepsilon = 16/255$ AA acc | AA sec | $A^3$ acc | $A^3$ sec | EDA acc | EDA sec |
|---|---|---|---|---|---|---|---|---|---|---|---|---|---|
| Rice et al. (2020) | PARN-18 | 33.37 | 2,372 | 33.36 | 403 | 33.37 | 586 | 4.45 | 399 | 4.38 | 1,029 | 4.39 | 218 |
| Rebuffi et al. (2021) | WRN-28-10 | 45.88 | 26,571 | 45.82 | 4,173 | 45.83 | 3,724 | 12.47 | 7,995 | 12.23 | 1,615 | 12.17 | 1,237 |
| Addepalli et al. (2022) | WRN-34-10 | 47.99 | 30,028 | 47.98 | 4,848 | 47.98 | 3,968 | 10.79 | 7,610 | 10.67 | 1,671 | 10.55 | 1,211 |
| Cui et al. (2021) | WRN-34-10 | 43.27 | 26,879 | 43.26 | 4,393 | 43.25 | 4,526 | 11.35 | 7,763 | 11.11 | 1,705 | 11.12 | 1,297 |
| Sitawarin et al. (2021) | WRN-34-10 | 41.37 | 28,193 | 41.37 | 4,223 | 41.37 | 3,067 | 6.38 | 5,087 | 6.14 | 19,993 | 6.23 | 807 |
| Rebuffi et al. (2021) | WRN-70-16 | 48.26 | 158,196 | 48.23 | 20,161 | 48.23 | 21,621 | 14.16 | 50,339 | 13.97 | 9,017 | 13.79 | 6,990 |

**ImageNet**

| Defence | Model | $\varepsilon = 2/255$ AA acc | AA sec | $A^3$ acc | $A^3$ sec | EDA acc | EDA sec | $\varepsilon = 8/255$ AA acc | AA sec | $A^3$ acc | $A^3$ sec | EDA acc | EDA sec |
|---|---|---|---|---|---|---|---|---|---|---|---|---|---|
| Salman et al. (2020) | RN-18 | - | - | 38.44 | 1,581 | 34.70 | 2,184 | - | - | 8.16 | 3,990 | 8.12 | 1,153 |
| Engstrom et al. (2019) | RN-50 | 45.64 | 17,000 | 45.64 | 4,284 | 45.62 | 3,874 | 7.88 | 3,389 | 7.44 | 11,717 | 7.72 | 1,578 |
| Salman et al. (2020) | RN-50 | - | - | 46.02 | 4,255 | 46.02 | 4,192 | - | - | 12.20 | 2,016 | 12.10 | 1,965 |
| Wong et al. (2020) | RN-50 | - | - | 35.22 | 5,810 | 35.16 | 5,657 | - | - | 7.66 | 20,459 | 7.80 | 2,290 |
| Salman et al. (2020) | WRN-50-2 | - | - | 50.32 | 8,017 | 50.28 | 6,788 | - | - | 12.56 | 3,506 | 12.60 | 2,377 |

Table 25: Robust accuracy of randomized defenses. The lowest robust accuracy is in bold.

| CIFAR-10 ($\varepsilon = 8/255$) | | | AA | | $A^3$ | | EDA | |
|---|---|---|---|---|---|---|---|---|
| Defense | Model | clean | acc | sec | acc | sec | acc | sec |
| RNA | RN-18 | 84.48 | 62.13 | 5,094 | 65.99 | 1,342 | **53.87** | 743 |
| RNA | WRN-32 | 86.49 | 64.53 | 30,560 | 69.70 | 16,087 | **53.43** | 4,764 |
| DWQ | PARN-18 | 82.11 | 53.81 | 6,469 | 56.63 | 3,161 | **48.00** | 897 |
| DWQ | WRN-34 | 81.56 | 58.05 | 46,058 | 60.21 | 7,043 | **50.32** | 5,856 |
| SVHN ($\varepsilon = 8/255$) | | | | | | | | |
| DWQ | PARN-18 | 81.97 | 31.90 | 4,117 | 38.80 | 1,802 | **28.53** | 676 |
| DWQ | WRN-34 | 88.65 | 37.20 | 29,969 | 43.87 | 5,438 | **35.23** | 4,027 |
| CIFAR100 ($\varepsilon = 8/255$) | | | | | | | | |
| RNA | RN-18 | 56.75 | 42.78 | 2,378 | 47.55 | 1,921 | **30.67** | 426 |
| RNA | WRN-32 | 60.19 | 41.94 | 15,296 | 45.98 | 17,575 | **30.71** | 1,851 |
| DWQ | PARN-18 | 55.96 | 32.61 | 3,559 | 33.72 | 1,491 | **26.51** | 607 |
| DWQ | WRN-34 | 55.94 | 36.77 | 25,539 | 37.82 | 4,658 | **29.64** | 3,311 |
| ImageNet ($\varepsilon = 4/255$) | | | | | | | | |
| DWQ | RN-50 (free) | 49.44 | 35.62 | 2,386 | 45.08 | 109 | **24.18** | 829 |
| DWQ | RN-50 (fgsmrs) | 62.18 | 39.08 | 21,994 | 40.72 | 4,854 | **33.72** | 3,773 |

to other methods. The above results indicate that robustness evaluation can be effectively achieved by using these methods in different ways depending on the acceptable computational resources.

## H    EDA's performance against randomized defenses

We investigated the efficacy of EDA against two randomized defenses, including Double-Win Quant (DWQ) (Fu et al., 2021) and Random Normalization Aggregation (RNA) (Dong et al., 2022). For the models trained on CIFAR-10/100 and SVHN (Netzer et al., 2011), we used $\varepsilon = 8/255$ and 10,000 test images. For the ImageNet, we used $\varepsilon = 4/255$ and 5,000 images, the same as the RobustBench. We report the results of a single run with a fixed random seed for reproducibility. The compared methods are AutoAttack (AA) and Adaptive Auto Attack ($A^3$). $A^3$ showed sufficiently fast and strong performance among recently proposed attacks against CNN-based robust models. For consistency with AA, we calculated the robust accuracy using generated adversarial examples when the attack terminated and compared these values because the model with randomized defense may produce different outputs with each inference. Official implementations of AA and EDA calculate robust accuracy in this way, but the official implementation of $A^3$ calculates differently. Therefore, we slightly modified the implementation of $A^3$ to calculate robust accuracy in the same way. The parameters of the compared attacks were the same as in the main text. Table 25 shows the robust accuracy and runtime of the standard version of AA, $A^3$, and EDA. From Table 25, EDA showed higher attack performance in less computation time. Although these are the results of a single run, EDA is expected to be effective for randomized defenses.

