# OpenReview forum: "Appropriate Balance of Diversification and Intensification Improves Performance and Efficiency of Adversarial Attacks"
_TMLR — Accepted by TMLR_

### Review · Reviewer_nGX2 · 2024-03-12

**Summary Of Contributions:**

This paper proposes an adversarial attack that improves the attack success rate of existing methods. It observes that different attacks tend to cluster in certain regions (i.e., local minima) in the search space. This paper proposes searching for different local solutions using various attack methods and hyper-parameters. Specifically, it divides the adversarial example generation into two phases: a diversification phase and an intensification phase. In the diversification phase, the paper uses a large optimization step size (i.e., twice the perturbation bound) and applies different existing attack methods. In the intensification phase, it selects the perturbation with the largest loss from the last phase as the initialization and uses a smaller step size. The evaluation is conducted on three image datasets with 41 models and 21 different defenses. The experimental results show that the proposed attack can improve the attack performance to some extent.

**Audience:**

Yes

**Claims And Evidence:**

No

**Requested Changes:**

See the above.

**Strengths And Weaknesses:**

Strengths

1. It is an interesting observation that different attacks and hyper-parameters result in optimization clustering in certain regions.

2. The evaluation is comprehensive, incorporating multiple existing baseline attacks and various defense techniques.


Weaknesses

1. The results on robust accuracy are marginal. As shown in the table, in most cases, the proposed attack can only reduce the robust accuracy by less than 0.2% compared to existing methods such as A^3. Similar observations are reported in Table 2. This improvement, when compared to existing attacks, is limited, and it is also challenging to determine whether the slight improvement results from the selection of random seeds and images for the attack.

2. The paper is difficult to follow. One of the main algorithms is presented in Algorithm 2, consisting of 26 lines of pseudo code. However, the text lacks a detailed description of the entire procedure. Specifically, in lines 8 and 19, it merely says to "run the attack following Algorithm 4," but Algorithm 4 is presented in the appendix. This setup makes it challenging for readers to grasp this part of the content. Ideally, the main text should be self-contained, eliminating the need to consult the appendix to understand the material.

3. According to Table 1, the number of queries needed for the proposed attack is significantly lower than that for A^3. An explanation of why it requires fewer queries would be beneficial. Additionally, the submission lacks a study on different perturbation bounds. Evaluating the attack across various bounds would more effectively demonstrate the performance of the proposed attack in comparison to existing methods.

---

> ### Author Response · Authors · 2024-04-09
>
> ### Response to weakness 1
> First of all, the improvement in attack performance by the proposed method (e.g. 0.11% with Salman et al. (No. 7)) is considered large enough, taking into account developments from recent studies, such as Composite Adversarial Attacks (CAA) and AutoAE over Auto Attack (e.g. 0.05% with Salman et al. (No. 7)), and Adaptive Auto Attack (A$^3$) over CAA (e.g. 0.05% with Salman et al. (No. 7)). These past improvements are almost the same as ours.
>
> The performance improvement from the proposed method is not the result of random seed because we report averages and standard deviations over five runs for 41 CNN-based models as the complete results to show that the improvement in attack performance is sufficiently large. We revise Section 4.1 to clarify this point.
> Additionally, the experiments on CIFAR-10/100 were conducted with 10,000 images, which is the complete test set. Thus, we have no room to choose test samples. For ImageNet, we conduct additional experiments on 50,000 images and confirm the superiority of the proposed method as in the case of 5,000 images. The experimental results are shown below.
>
> Table 1. Robust accuracy on the whole validation set of ImageNet
> | Defense | Clean accuracy | A$^3$ | EDA |
> | --- | --- | --- | --- |
> AT | 62.41 | 28.92 | 28.78 |
> FastAT | 55.45 | 26.05 | 25.91 |
> Salman_R18 | 52.50 | 24.98 | 24.91|
> Salman_R50 | 63.87 | 34.80 | 34.69 |
>
> Table 2.  Runtime on the whole validation set of ImageNet in seconds.
> |Defense | A$^3$ | EDA|
> | --- | --- | --- |
> AT | 30,953 | 29,836
> FastAT | 46,928 | 46,085
> Salman_R18 | 11,453 | 16,811
> SalmanR50 | 34,869 | 33,307
>
> ### Response to weakness 2
> Thank you for your suggestion for our presentation. We move Algorithm 4 and its description to the text. We also add the high-level idea of the proposed methods as requested by reviewer XVrq.
>
> ### Response to weakness 3
> Efficient Diversified Attack (EDA) requires fewer queries than A$^3$ because of the complementarity of multi-targeted and MDO strategies (Section 4.2) and a higher degree of diversification of EDA (Section 4.1). We modify Section 4.1 to make the discussion clearer.
> We also add experimental results for multiple perturbation bounds. The results show that A$^3$ performs better but at the expense of computation time for a larger perturbation bound. EDA performs better than AA in much less time than AA and A$^3$ for a larger perturbation bound. All methods show similar performance with a small perturbation bound.

---

### Review · Reviewer_WpYh · 2024-03-12

**Summary Of Contributions:**

This paper proposes an efficient and effective adversarial attack method called EDA. There are two critical stages in EDA: (1) The Diversification stage tries to use different objectives and directions to find a data point that has the largest C&W loss; (2) The Intensification stage updates the data point via APGD. The empirical results seem to validate the effectiveness of the proposed method.

**Audience:**

Yes

**Broader Impact Concerns:**

None.

**Claims And Evidence:**

Yes

**Requested Changes:**

Please see the weaknesses above.

**Strengths And Weaknesses:**

Strengths:

1. The method is well-motivated. Considering both diversity and intensity is important for improving the attack performance.

2. The experiments are conducted on comprehensive datasets and models. The authors also report the standard deviation to validate the significance of performance gain.

Weaknesses:

1. The novelty is limited though novelty is not an important evaluation metric in TMLR. I understand that the authors first find a good starting point and then use APGD (the existing work) to find the adversarial data. The starting point is chosen based on different loss objectives which are existing works as well.

2. Not many new insights are delivered from this paper though the proposed method works empirically. Prior works have tried to consider improving both diversity and intensity to boost adversarial attacks. Therefore, I found limited contributions to inspiring future works. It would be great if the authors could provide any.

---

> ### Author Response · Authors · 2024-04-09
>
> ### Response to weakness 1
> As noted in the comments, the diversification phase focuses on finding appropriate starting points. However, the intensification phase uses other update directions selected by the ADS, in addition to APGD. APGD execution, the final step of Efficient Diversified Attack (EDA) is independent of the diversification phase because EDA executes the MDO Framework and the multi-targeted attack independently.
> We revise Section 3.4 to make this point clearer.
>
> Also, whereas the starting point is selected in the diversification phase using objective functions that include existing methods, the effective combination is not trivial. Unlike existing methods such as CAA and AutoAE, which are based on direct indicators such as objective function values and attack success rates, ADS uses an indirect indicator: the degree of diversification. This is a challenging topic, and we are the first to propose the selection method using indirect indicators. Experimental results show that this approach is effective.
>
> ### Response to weakness 2
> The experimental results in Section 3.1, which other reviewers note as a strength, may lead to a better understanding of optimization-based adversarial attacks and adversarial training. For example, adversarial training with attack methods that belong to clusters different from those formed by PGD, which are usually used for adversarial training, may yield different results than in conventional adversarial training.
> We add such future research possibilities to the conclusion.

---

### Review · Reviewer_XVrq · 2024-04-05

**Summary Of Contributions:**

This paper proposes Multi-directions/objectives (MDO) strategy and Efficient Diversified Attack (EDA) to improve adversarial attacks. Specifically, MDO uses multiple search directions and objective functions, whereas EDA is faster and more efficient by combining MDO with MT attack.

**Audience:**

Yes

**Broader Impact Concerns:**

The broader impact is explicitly discussed in the paper.

**Claims And Evidence:**

Yes

**Requested Changes:**

- The paper needs a significant improving in presentation and writing. The mathematical notions need to be simplified to improve comprehensibility.
- The motivations and high-level ideas of ADS, MDO, and EDA need to be presented and discussed.
- Adversarial examples need to be visualized. Moreover, the adversarial examples need to be used in adversarial training to report the robust accuracies. This helps to demonstrate the merits of the generated adversarial examples.
- Algorithms 2 and 3 need to be represented with more detailed explanations of the rationale of each step. Currently, they look heuristic and messy.

**Strengths And Weaknesses:**

## Strengths
- The experimental results in Section 3.1 Motivation are quite interesting and make sense.

## Weaknesses
- This paper is very heuristics. The writing can be improved by simplifying mathematical notions.
- The Automated Diversified Selection (ADS) is heuristic to me. Specifically, we need to search through many combinations of $n_a$ attacks to choose the optimal one with the smallest $D^e$. Moreover, the choosing of DI for measuring the diversity of adversarial examples is not well-justified. Even, DI has not described in the paper yet.
- The proposed attack requires a testing set in advance, hence hindering its application. It would be better if we can attack single point solely.
- Algorithm 3 is almost not understandable. It is unclear where MT is used in this EDA algorithm.
- Although the proposed algorithms are complicated, they only slightly improve $MT_{cos}$ in Table 1. Also, it is unclear why the proposed algorithms are more scalable than the baselines because they look complicated, i.e., need to do at least two NAS and for each attack in the list of NAS, they use this attack to attack a point in the testing set.

---

> ### Author Response · Authors · 2024-04-09
>
> ### Requested changes 1
> Thank you for your suggestion. First, we simplify variables that have superscripts or subscripts. We are also considering reducing the number of variables defined.
> ### Requested changes 2 and 4
> Thank you for your suggestion. We describe the motivation and overview of each algorithm, including its design philosophy, in more detail.
> ### Requested changes 3
> Thank you for your suggestion. This is an interesting perspective. We will add the visualization of adversarial examples to our paper. We are in the process of experimenting with adversarial training using EDA.

---

### Author Response · Authors · 2024-04-09

To all the reviewers, we would like to thank you very much for using your valuable time to review our paper.  We reply to each reviewer’s comments in a separate thread. We will update the PDF and inform you of the changes to our paper when additional experiments are completed.

---

> ### Author Response · Authors · 2024-04-16
> **Revised pdf is uploaded**
>
> ### Changes
> All changes are highlighted in blue. The page numbers, section numbers, figures, and algorithm numbers, etc. in the following comments correspond to the revised pdf.
> #### reviewer nGX2
> **(weakness 1)** We modified Section 4.1 (p.10) to clarify that we conducted experiments with several random seeds.
>
> **(weakness 2)** We added pseudocode of general white-box attacks and its explanation to Section 2.2 (p.3-4). We also added the definition of Diversity Index (DI) to Section 2.2 (p.4-5).
>
> **(weakness 3)** We modified Section 4.1 (p.11) to explain the reason for EDA's lower computation cost. We also added the experimental results for different perturbation bounds ($\varepsilon=4/255$ and $16/255$ for CIFAR-10/100, and $\varepsilon=2/255$ and $8/255$ for ImageNet) to Section 4.1 (p.12, Figure 9). The raw results are described in Appendix G.4 as Table 22 (p.40).
>
> #### reviewer WpYh
> **(weakness 1)** We modified Section 3.4 (p.8-9) to clarify the procedure of EDA.
>
> **(weakness 2)** We added future research directions to Conclusion (p.15-16)
>
> #### reviewer XVrq
> **(requested change 1)** X**, X*, and $(x_{i})^{*}_{a, r}$ were eliminated and unified into $X^a_i$ or union of $X^a_i$ (Section 3.1, p.5). $X^a_i$ is defined constructively in Algorithm 1 (Section 2.2, p.3). We also modified the pseudocode in Sections 3.2-3.4 and Figure 5 to reflect the above changes in mathematical notation.
>
> **(requested changes 2 and 4)** To clarify the motivations and high-level idea of this study, we modified the Introduction (p.1-3). We also added motivations, detailed explanations, and overviews of the proposed methods to Sections 3.2-3.4 (p.6-9)
>
> **(requested change 3)** We visualized the adversarial examples as Figure 8 (p.12) and added several findings (p.11-12).
>
> **About adversarial training using EDA (requested by reviewer XVrq)**
> We would like to consider adversarial training with EDA as future work for the reasons below. Firstly, we found that adversarial training of ResNet-18 using EDA requires several hours per epoch, despite EDA's lower computational cost compared to other state-of-the-art attacks. Adjusting the learning rate and other hyperparameters, as well as validating the training across multiple architectures during this rebuttal period, would be challenging because training typically requires hundreds of epochs to converge.
> Secondly, the comparative methods, including AA, CAA, A$^3$, AutoAE, and EDA, are primarily designed for robustness evaluation. Thus, some methods require further refinement for adversarial training. Considering these reasons, adversarial training appears to be somewhat outside the scope of this study.

---

### Decision · Action_Editor_ZymZ · 2024-05-13

**Recommendation:** Accept with minor revision

**Comment:**

In their official recommendation, two reviewers lean toward accept and one reviewer leans toward reject this work. I read the paper in detail and also find that some claims in this work are not very well supported.

**1. Design Choices in the Algorithms:** The proposed multi-direction/objectives (MDO) strategy and eficient diversified attack (EDA) are the main contributions of this work. The motivation proposed in section 3.1 is intuitive and great, but the readers could struggle to fully understand the details and all the steps of the proposed algorithms. During the rebuttal, the authors put some effort into improving the methodology subsections (3.2 - 3.4), but there is still room for improvement.

There are many design choices in the proposed algorithms that make them complicated. The added descriptions in the revised paper are helpful, but clear evidence and further analyses are still needed to fully support these design choices (e.g., what are their concrete effects/impact for diversification and/or intensification). Neat ablation/validation and/or intuitive illustration (as in subsection 3.1) could also be very helpful to truly motivate each proposed design choice. On the other hand, are all the proposed design choices necessary for better diversification or intensification? If some designs only have a minor (or even no) impact, simplifying the algorithms could also significantly improve the exposition of this work.

**2. Empirical Supports for Efficient Diversified Attack (EDA):** A key claim of the submission is that "(The proposed EDA method) is a faster and stronger attack using MDO and MT strategies." However, according to the new experimental results for multiple perturbation bounds, EDA is outperformed by A^3 for a larger perturbation bound, and achieves similar performance with other methods (AA and A^3) for a smaller perturbation bound. Therefore, the claimed stronger attack is not well supported.

For the computational overhead, it is promising to see EDA requires a smaller number of queries than AA and A^3 in Table 1, and also a faster runtime in Table 22. However, the support and explanation for this efficiency is not enough. In the response, the authors simply attribute the fewer queries to "the complementarity of multi-targeted and MDO strategies (Section 4.2) and a higher degree of diversification of EDA (Section 4.1)". A deeper analysis and discussion could be required, especially since EDA is outperformed by A^3 for a larger perturbation bound (e.g., will the smaller queries be due to premature convergence?).

Another possible way to tackle this concern is to properly adjust the claimed contribution on "(The proposed EDA method) is a faster and stronger attack using MDO and MT strategies." to make it precisely reflect the performance of EDA. Adding a limitation section to clearly and explicitly discuss the pros and cons of MDO/EDA could also be very helpful for the readers to truly understand the proposed method.

This paper is an interesting work that studies an important topic (diversification v. intensification) for adversarial attacks, and I also appreciate the efforts the authors have made for a comprehensive comparison with other methods as well as the added motivation/description in the revised paper. Therefore I recommend **accept this paper but with a "minor" (somewhat like a major) revision**. In the revised paper, I expect the authors can properly address the concerns raised above.

**Audience:**

All reviewers believe some individuals in TMLR's audience could be interested in the findings of this paper.

**Claims And Evidence:**

This work investigates the challenge of diversification (finding multiple different local optima) and itensification (fast convergence to local optima) for white-box adversarial attacks. The main contributions are 1) a multi-direction/objectives (MDO) strategy with automated diversified selection (ADS) to efficiently find diverse local optimal solutions, and 2) a fast diversified attack (EDA) based on MDO and multi-target (MT) strategy.

The reviewers raised different weaknesses and concerns on this submission, and the authors have provided a reasonable response with a revised paper. After rebuttal, one reviewer still believe some claims in this paper are not well supported by clear evidence.

---

> ### Author Response · Authors · 2024-05-27
> **Camera ready revision is uploaded.**
>
> Thank you for your comments and suggestions. Your comments are extremely helpful in improving the paper. Our response to your comments and corresponding changes to the paper are listed below.
>
> Comment 1:
> ---
> The first comment concerns the clarity of the procedure for the proposed method and the influence of each component.
>
> To improve the clarity regarding the procedure of the proposed algorithm, we consider illustrating the entire procedure of the MDO framework and EDA in the corresponding subsections, i.e., Section 3.3 and Section 3.4, respectively.
> In addition, an explanation of why DI-based indicators are used in the ADS and why ensemble and composite are used in the MDO framework is added to the corresponding subsections.
>
> Furthermore, several ablation studies have been conducted to clarify the role of the algorithmic components. Specifically, we examine the effect of limiting the number of combinations of objective functions and search directions in ADS (line 4 of algorithm 2) and the effect of limiting the images to be attacked in the intensification phase of the MDO framework (lines 21-23 of algorithm 3).
> We also add a more detailed explanation of the role of MT strategy in EDA from a diversification perspective.
>
> Comment 2:
> ---
> The second comment is that the key claim of this paper is not well supported by the experiments.
> Even in the pre-revision paper, EDA is a powerful and fast attack method under standard epsilon settings in RobustBench, and this claim is well supported by the experimental results.
> Therefore, we have changed the claim that EDA is a strong and fast attack method by adding the condition "under standard epsilon settings in RobustBench" so that the claim appropriately reflects EDA's performance.
>
> In addition, the additional experiments in the rebuttal showed that A$^3$ had the best attack performance and the longest computation time under large epsilon.
> To test whether the relatively degraded attack performance of EDA under large epsilon is due to premature convergence of EDA's search, we perform additional experiments by increasing EDA's computation time; if performance does not improve with increasing EDA's computation time, EDA's search is likely to converge prematurely.
>
> Additional experiments show that attack performance equal to or better than that of A$^3$ can be obtained even when EDA's runtime is increased to the extent that EDA's runtime does not exceed A$^3$'s runtime.
> Therefore, as long as the parameters related to the number of iterations of EDA are set appropriately, EDA is expected to achieve higher attack performance with a shorter execution time than A$^3$, regardless of the value of epsilon.
>
> With the above changes, we believe the revised paper addresses the concerns.

---

> > ### Comment · Action_Editor_ZymZ · 2024-05-27
> >
> > Dear Authors,
> >
> > Thank you for the thorough revision. I've gone through the updated paper and believe the raised concerns have been well addressed.
> >
> > Since this is the camera-ready revision, please **change those "changes highlighted in blue" into black**, then I can officially verify the camera-ready version of this work.
> >
> > Best Regards and Congratulations!
> >
> > AE

---

> > > ### Author Response · Authors · 2024-05-27
> > >
> > > Dear AE,
> > >
> > > Again, thank you very much for the time and energy reviewers and AE expended.
> > >
> > > We have changed those "changes highlighted in blue" into black.
> > >
> > > Best regards,
> > >
> > > Authors